# Acquisition of epithelial plasticity in human chronic liver disease

Christopher Gribben[1,2,14], Vasileios Galanakis[1,2,3,14], Alexander Calderwood[1], Eleanor C. Williams[1], Ruben Chazarra-Gil[1], Miguel Larraz[1], Carla Frau[4], Tobias Puengel[5,6], Adrien Guillot[5], Foad J. Rouhani[7], Krishnaa Mahbubani[8], Edmund Godfrey[9], Susan E. Davies[10], Emmanouil Athanasiadis[11,12], Kourosh Saeb-Parsy[8], Frank Tacke[5], Michael Allison[2,3✉], Irina Mohorianu[1✉] & Ludovic Vallier[1,2,4,13✉]

For many adult human organs, tissue regeneration during chronic disease remains a controversial subject. Regenerative processes are easily observed in animal models, and their underlying mechanisms are becoming well characterized[1–4], but technical challenges and ethical aspects are limiting the validation of these results in humans. We decided to address this difficulty with respect to the liver. This organ displays the remarkable ability to regenerate after acute injury, although liver regeneration in the context of recurring injury remains to be fully demonstrated. Here we performed single-nucleus RNA sequencing (snRNA-seq) on 47 liver biopsies from patients with different stages of metabolic dysfunction-associated steatotic liver disease to establish a cellular map of the liver during disease progression. We then combined these single-cell-level data with advanced 3D imaging to reveal profound changes in the liver architecture. Hepatocytes lose their zonation and considerable reorganization of the biliary tree takes place. More importantly, our study uncovers transdifferentiation events that occur between hepatocytes and cholangiocytes without the presence of adult stem cells or developmental progenitor activation. Detailed analyses and functional validations using cholangiocyte organoids confirm the importance of the PI3K–AKT–mTOR pathway in this process, thereby connecting this acquisition of plasticity to insulin signalling. Together, our data indicate that chronic injury creates an environment that induces cellular plasticity in human organs, and understanding the underlying mechanisms of this process could open new therapeutic avenues in the management of chronic diseases.

The ability of adult organs to regenerate has been well documented in animal models, and functional studies combined with lineage-tracing experiments have shown that different injuries induce divergent regenerative processes[1–4]. However, it is difficult to demonstrate the existence of such events in human organs for technical and ethical reasons. The liver is a particularly interesting organ in this context. The main functional cell types in the hepatic epithelium are the hepatocytes, which are known for their metabolic roles, and the cholangiocytes, which line the biliary tree and transport bile acids. The process by which these cells are replaced after injury depends on the insult encountered. Cell proliferation occurs during acute liver injury[5–7], but this capacity to proliferate is abolished in chronic diseases[8,9]. Animal studies have revealed three alternative mechanisms[10,11]: stem cells or progenitors can be activated and then differentiate into epithelial cells[12–15]; cholangiocytes may transdifferentiate into hepatocytes, or vice versa[1,16–23]; or hepatocytes and cholangiocytes could reverse to a developmental progenitor to restore the corresponding cell compartment[24–26]. Signs of these mechanisms have been observed in humans, but the nature of the regenerative processes that occur during chronic liver disease remain to be fully understood[27–29]. To address this question, we combined single-nucleus analyses, 3D imaging and functional experiments to study both cell behaviour and the regenerative processes that occur during the progression of metabolic dysfunction-associated steatotic liver disease (MASLD), a chronic liver disease that affects a growing population of people worldwide[30].

[1]Wellcome-MRC Cambridge Stem Cell Institute, University of Cambridge, Cambridge, UK. [2]Open Targets, Wellcome Genome Campus, Hinxton, UK. [3]Liver Unit, Department of Medicine, Cambridge NIHR Biomedical Research Centre, Cambridge University Hospitals NHS Foundation Trust, Cambridge, UK. [4]Berlin Institute of Health Centre for Regenerative Therapies, Berlin, Germany. [5]Department of Hepatology and Gastroenterology, Charité Universitätsmedizin Berlin, Berlin, Germany. [6]Berlin Institute of Health, Berlin, Germany. [7]Francis Crick Institute, London, UK. [8]Department of Surgery, University of Cambridge, Cambridge, UK. [9]Department of Radiology, Addenbrooke's Hospital, Cambridge, UK. [10]Department of Histopathology, Cambridge University Hospitals NHS Foundation Trust, Cambridge, UK. [11]Greek Genome Centre, Biomedical Research Foundation of the Academy of Athens, Athens, Greece. [12]Medical Image and Signal Processing Laboratory, Department of Biomedical Engineering, University of West Attica, Athens, Greece. [13]Max Planck Institute for Molecular Genetics, Berlin, Germany. [14]These authors contributed equally: Christopher Gribben, Vasileios Galanakis. ✉e-mail: michael.allison6@nhs.net; iim22@cam.ac.uk; ludovic.vallier@bih-charite.de

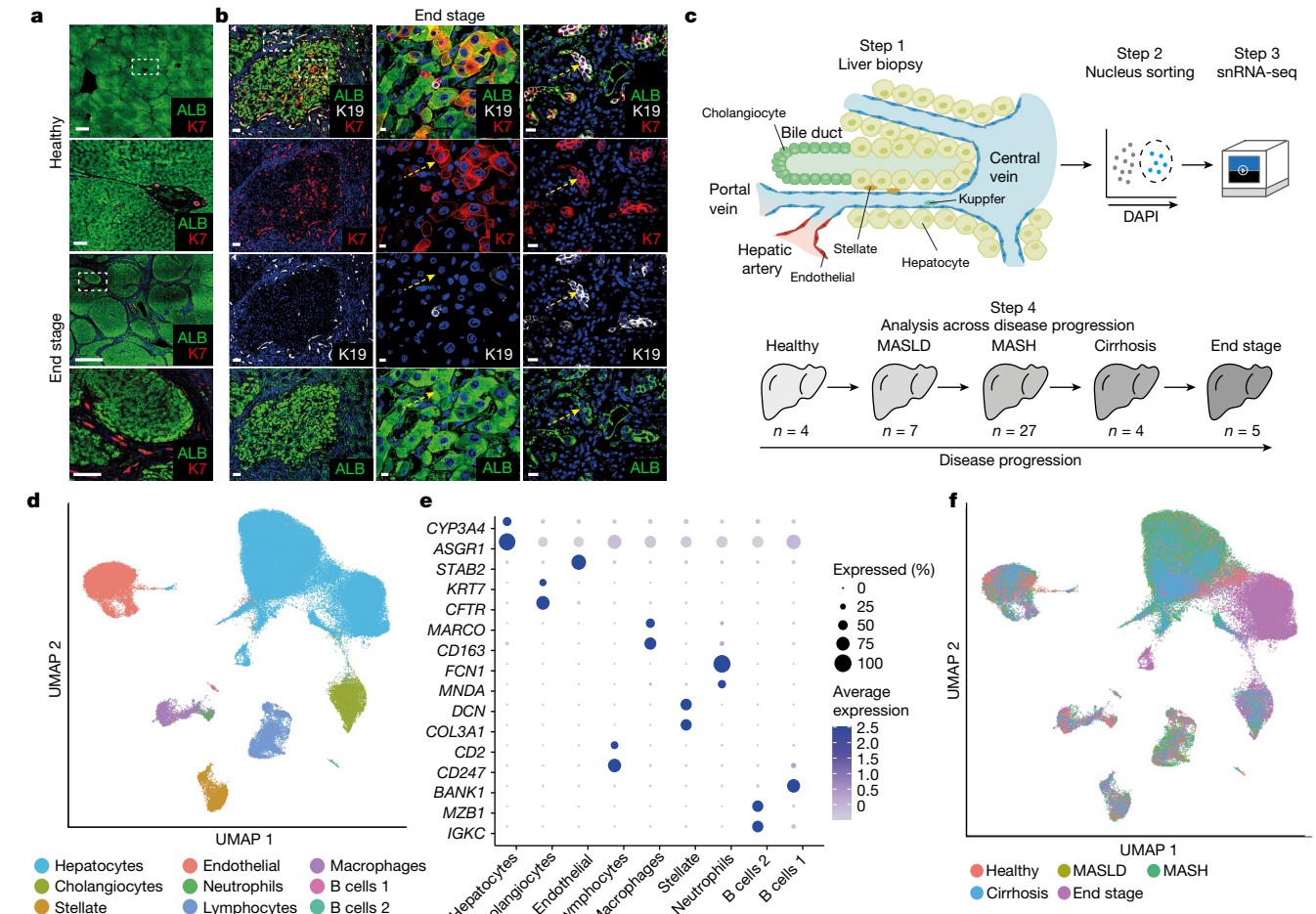

**Fig. 1 | Using snRNA-seq of MASLD progression to analyse cholangiocyte and hepatocyte plasticity. a**, Immunofluorescence staining for K7 and ALB in liver sections from healthy donors and those with end stage disease, with high magnification of the areas in the dashed boxes underneath. Scale bars: 1,000 μm for low magnifications and 100 μm for high magnifications; $n = 3$ healthy and $n = 3$ end-stage-disease tissue samples. **b**, Immunofluorescence staining of end-stage MASLD tissue sections. High magnification of the dashed boxes shows examples of cells that are double-positive for K19 or K7 and for ALB in the hepatocyte nodule and in the surrounding ductal structures. An example is indicated by the yellow arrows; $n = 3$ tissue samples. Scale bars: left, 100 μm; middle, 10 μm; right, 15 μm. **c**, Schematic of the snRNA-seq experimental workflow; $n$ shows the number of samples at each stage. **d**, Overall UMAP showing cell annotation from all disease stages after quality control. **e**, Bubble plot of the expression of cell-type markers. **f**, Overall UMAP shown by disease stage.

## snRNA-seq captures liver cells across MASLD

MASLD is a progressive disease that starts with the accumulation of fat in hepatocytes. Over time, this accumulation can result in cell death, leading to inflammation, fibrosis, cirrhosis and liver failure or liver cancer[31]. We first assessed whether livers affected by progressive MASLD display evidence of regenerative processes. For that, we performed immunostaining to compare tissue sections of healthy liver with those of biopsies from people at different stages of disease progression (Fig. 1a,b and Extended Data Fig. 1a). Major changes were evident, especially in livers from people with end-stage disease, with the expected appearance of regenerative nodules containing hepatocytes (indicated by the hepatocyte marker ALB) surrounded by large collagen depositions[32] (Extended Data Fig. 1b). Immunostaining for the cholangiocyte markers keratin 7 (K7, also known as KRT7) and keratin 19 (K19, also known as KRT19) showed a strong increase in ductal structures around these nodules (Fig. 1a), a process known as the ductular reaction[33], which is commonly seen in acute and chronic liver disease[34,35]. These experiments also revealed cells co-expressing K7 and the hepatocyte markers ALB or HepPar1 (Fig. 1b and Extended Data Fig. 1c). These may represent intermediate hepatocytes, which have been observed histologically in human MASLD[28]. However, we also observed cells co-expressing ALB, K7 and K19 that seem to be present specifically in end-stage liver (Fig. 1b and Extended Data Fig. 1d), indicating the presence of cells combining hepatocyte and cholangiocyte phenotypes. Importantly, such biphenotypic cells have been associated with the regenerative process[32,36,37], so their appearance could be indicative of epithelial regeneration in end-stage MASLD.

To further examine the events leading to the emergence of biphenotypic cells and their role in disease, we decided to study MASLD progression at the single-cell level. To this end, we collected liver biopsies from 47 people across the different stages of MASLD progression defined by histology as healthy, MASLD, metabolic dysfunction-associated steatohepatitis (MASH), cirrhosis and end-stage disease (Fig. 1c and Supplementary Tables 1 and 2). Half of the biopsy was allocated for diagnostic and staging work, and the other half was rapidly frozen to be processed at a later stage (Fig. 1c). However, we quickly abandoned using cells isolated from fresh biopsy because many hepatocytes and cholangiocytes were lost using this method, as shown by previous studies[38,39]. To bypass this limitation, we developed a protocol for nucleus isolation involving tissue lysis and fluorescence-activated

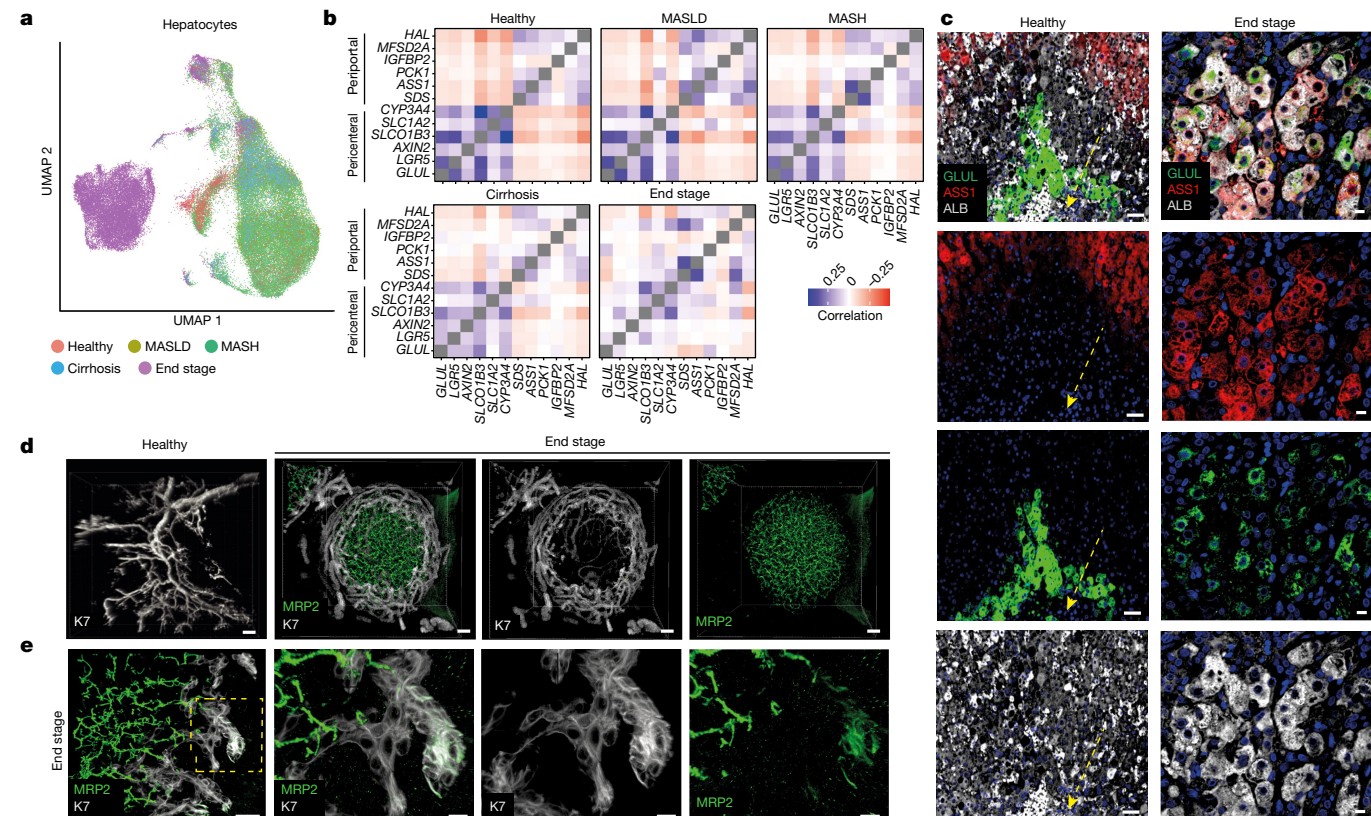

**Fig. 2 | Major changes in hepatocyte zonation and biliary-tree remodelling in end-stage MASLD. a,** UMAP of hepatocytes annotated by disease stage. **b,** Correlation analysis examining expression of pericentral and periportal hepatocyte markers across disease progression. **c,** Immunofluorescence staining for pericentral marker GLUL, periportal marker ASS1 and pan-hepatocyte marker ALB in healthy and end-stage MASLD tissue sections. The yellow dashed arrow indicates the central vein; $n = 3$ healthy and $n = 3$ end-stage tissue samples. Scale bars: 50 μm (healthy) and 10 μm (end stage).

**d,** 3D projections of cleared healthy and end-stage MASLD liver samples. Staining with K7 for cholangiocytes and MRP2 for hepatocytes; $n = 3$ healthy and $n = 3$ end-stage tissue samples. Scale bars: 100 μm (healthy), 50 μm (end stage). **e,** The area in the yellow box (left) is shown in higher magnification in the other three boxes, highlighting an example of a MRP2–K19 co-positive cell at the end of a duct; $n = 3$ healthy and $n = 3$ end-stage tissue samples. Scale bars: 30 μm (left), 10 μm (the other panels). See also Supplementary Videos 3 and 4.

cell sorting (FACS), which allowed the purification of high-quality nuclei even from fibrotic tissues. Using this protocol, just under 100,000 nuclei were isolated after quality control, which excluded cells expressing stress markers such as mitochondrial and ribosomal proteins (Fig. 1d and Extended Data Fig. 2a,b). Further analyses confirmed that our method captured all the expected liver cell types from all disease stages at similar proportions to the native tissue (Fig. 1d–f, Extended Data Fig. 3a–c and Supplementary Table 3). Accordingly, our collection was enriched in hepatocytes ($n = 69,426$) and cholangiocytes ($n = 5,412$). Notably, cell-type-specific clusters mostly overlap independently of the disease stage, except for hepatocytes, which display clear transcriptional changes following disease progression, even after a batch correction of technical effects using Harmony[40] (Fig. 1f and Extended Data Fig. 3d,e). Thus, hepatocytes seem to be the cell type most affected by the disease. Finally, our single-nucleus analyses also revealed the presence of cells co-expressing hepatocyte and cholangiocyte markers. We observed cells that appeared to bridge hepatocyte and cholangiocyte clusters and that co-expressed specific markers for both cell types (Extended Data Fig. 3f). Quality control was performed to confirm that these cells were not the result of doublets or RNA contamination. Together, these experiments show that our single-nucleus isolation protocol is compatible with single-cell-level transcriptomic analyses of liver biopsies and confirm the presence of biphenotypic cells, which have previously been associated with regenerative processes in the liver of people with progressive MASLD.

## MASLD remodels the liver microenvironment

Before investigating the origin of biphenotypic cells in more detail, we decided to probe the transcriptomic changes occurring in each cell type. All cell types exhibited differentially expressed genes across disease progression with strong separation of cells in end-stage disease for cholangiocytes, stellate cells and endothelial cells observed in the UMAP (uniform manifold approximation and projection) space, indicating that disease progression affects all the liver cells (Extended Data Fig. 4a–f). However, hepatocyte populations displayed the strongest transcriptional change in end-stage disease (Fig. 2a), and gene-set enrichment analyses (GSEA) showed a diversity of pathways upregulated during disease progression. Of particular interest, we observed major adjustments in pathways related to the microenvironment, such as hypoxia-inducible factor I signalling and gluconeogenesis, indicative of changes in liver zonation (Extended Data Fig. 5a). In the healthy liver, hepatocytes located in different zones of the liver lobules diverged in their expression of functional markers. For example, hepatocytes closer to the central vein (pericentral) express the WNT signalling genes *LGR5* and *AXIN2*, whereas hepatocytes closer to the portal triad (periportal) express the metabolic enzymes HAL and ASS1 (ref. 41). Accordingly, healthy hepatocytes can be clearly separated using correlation analyses for known zonation markers (Fig. 2b). However, this distinction breaks down during disease progression, with end-stage hepatocytes co-expressing pericentral and periportal markers (Fig. 2b and Extended Data Fig. 5b). These observations were validated by immunostaining

and 3D fast light-microscopic analysis of antibody-stained whole organs (FLASH) imaging for pericentral marker GLUL, periportal marker ASS1 and the pan-hepatocyte marker ALB in optically cleared tissue. Cells aberrantly co-expressing these markers were observed across regenerative nodules in end-stage livers (Fig. 2c, Extended Data Fig. 5c and Supplementary Videos 1 and 2), indicating a loss of zonation at the transcriptional and protein level in hepatocytes. These results reinforce previous studies[42,43], but by showing that hepatocytes acquire progressively the capacity to co-express zonation markers that are mutually exclusive in healthy liver, they also indicate that disease progression strongly modifies the liver microenvironment, resulting in the loss of functional zonation.

The organization of the biliary epithelium can also be strongly affected during disease progression by the ductular reaction[44,45]. Accordingly, cholangiocytes also display a strong disease signature (Extended Data Fig. 4d) characterized by an increase in the ductular-reaction markers *NCAM1* and *TNFRSF12A*. In parallel, we observed increased numbers of bile ducts during MASLD progression (Fig. 1a) to an extent indicating that this process has a major effect on liver architecture. To confirm this hypothesis, we imaged the biliary tree in 3D using FLASH technology on healthy and end-stage tissue. As expected, K7 staining of healthy tissue revealed a network of ducts that formed a branching tree-like structure (Fig. 2d). By contrast, end-stage samples exhibited complex basket-like structures surrounding the hepatocyte nodules. (Fig. 2d and Supplementary Videos 3 and 4). Such structures indicate a profound remodelling of the biliary tree to an extent not previously suspected. Furthermore, FLASH imaging was also performed to define the location of the biphenotypic cells in the diseased biliary tree. This revealed that cells co-expressing K7 and hepatocyte marker MRP2 tended to be located towards the ends of the small ducts (Fig. 2e and Supplementary Video 5), which undergo major transformation during disease progression by becoming bulkier and containing multiple cells in the end stage. Interestingly, the same region has previously been associated with hepatic stem cells[46] (Extended Data Fig. 5d). Taken together, these results indicate that the appearance of biphenotypic cells could be associated with a major reorganization of the biliary tree and the liver microenvironment during disease progression.

## Hepatocyte and cholangiocyte plasticity

Having established the presence of biphenotypic cells and their association in part with the abnormal organization of the biliary tree during disease, we next focused on defining their origin by performing detailed subclustering of hepatocytes and cholangiocytes. These analyses revealed two cholangiocyte subpopulations, namely *MUC1*-expressing cholangiocytes from the larger ducts and small cholangiocytes that expressed *BCL2* (Extended Data Fig. 6a–d). The population of small cholangiocytes was also associated with the ductular-reaction markers *NCAM1* and *TNFRSF12A* (ref. 29; Extended Data Fig. 6e–g), indicating that our sampling did capture ductular-reaction structures. These ductular-reaction cells were more common in end-stage disease, in line with our observations of the tissue sections (Extended Data Figs. 1a,d and 6j). Furthermore, subclustering of cholangiocyte populations identified biphenotypic cells expressing multiple hepatocyte markers (Fig. 3a,b) in clusters 5, 9 and 1 (Fig. 3c,d). Interestingly, the same analyses performed on hepatocytes also revealed that cluster 9 includes cells expressing multiple cholangiocyte markers. Thus, both cell types could be able to generate biphenotypic cells. Hepatocyte cluster 9 contains cells from different disease stages, so we decided to further subcluster this population to identify a more-biphenotypic phenotype. Notably, hepatocytes expressing the highest level of cholangiocyte markers were more common in end-stage disease (Fig. 3e), indicating that cell plasticity could occur with disease progression (Fig. 1b). These cells tended to express cholangiocyte markers for small but not large

cholangiocytes (Extended Data Fig. 7a) and may indicate that they are more likely to be found in small ducts, in line with our observations from 3D staining (Fig. 2d). Similar analyses performed on cholangiocytes showed that cholangiocyte cluster 1 cells were more prominent in end-stage disease, whereas clusters 5 and 9 were also found in earlier stages (Extended Data Fig. 7b–d). Biphenotypic cells were found to express comparable levels of cholangiocyte and hepatocyte markers to the main cholangiocyte and hepatocyte populations (Extended Data Fig. 7e). These results indicate that biphenotypic cells could appear earlier in disease than was initially suggested by our immunostaining analyses (Extended Data Fig. 1a). These early cells could represent intermediate cells described previously[28] that display limited plasticity. The full phenotype, and thus the capacity to generate cells with a biphenotypic transcriptome, seems to be acquired only towards the end stage of progression.

We then decided to define the origin of these biphenotypic cells. In the biphenotypic population, no cells were found to co-express the adult stem-cell markers *LGR5* and *TROP2* (also known as *TACSTD2*) (Extended Data Fig. 7f–i). We also hypothesized that stem cells, by definition, should be able to self-renew. However, proliferative markers such as *MKI67* were rarely co-expressed with *LGR5* ($n = 1$ of 46 *LGR5*-positive cells), and no proliferative cells expressed *TROP2* (Extended Data Fig. 7f–i). Notably, quiescence-marker expression increased during disease progression in hepatocytes (Extended Data Fig. 7j), confirming that regeneration by proliferation is limited in chronic injury. Together, these results indicate that biphenotypic cells are unlikely to originate from a stem-cell population. We next tried to determine whether a dedifferentiation or redifferentiation process could be occurring in the biphenotypic cells. For that, we examined the expression of the fetal liver markers *AFP* and *SPINK1* (ref. 47). No cells co-expressing *AFP* and *SPINK1* were found, and although rare *AFP*+ cells were observed, none were proliferative (Extended Data Fig. 7k,l). Thus, biphenotypic cells do not seem to originate from a stem-cell population or from a dedifferentiated or developmental progenitor. Together, these results indicate that biphenotypic cells appear during disease progression, whereas transdifferentiation is prominent in end-stage disease. These data do not rule out a role for the ductular reaction and/or intermediate hepatocytes in this acquisition of plasticity, and that these cells may act as precursors to biphenotypic cells, which increase over time during chronic injury.

## Identification of plasticity factors

We next investigated the mechanisms that increase plasticity by focusing on end-stage cells because they display the highest level of marker co-expression. We generated a UMAP including only hepatocytes and cholangiocytes from end-stage disease (Fig. 3f) and then localized the biphenotypic cells from a subcluster of hepatocyte cluster 9 and a subcluster of cholangiocyte cluster 1. As expected, the selected cells bridge cholangiocytes and hepatocytes, confirming their transdifferentiating state (Fig. 3g). To address the directionality of this transdifferentiation, we calculated the RNA velocity for these cells. Cholangiocyte-like hepatocytes indicated bidirectionality, whereas hepatocyte-like cholangiocytes showed a predominant direction from cholangiocytes to hepatocytes (Fig. 3h,i). Thus, transdifferentiation seems to occur in both directions. We then inferred the pseudotime to identify genes expressed specifically during transdifferentiation (Fig. 3j). This analysis revealed numerous genes that were upregulated in the biphenotypic population (Fig. 3k), and we selected *SOX4*, *KRT23*, *KLF4* and *NCAM1* for further validation. Immunostaining revealed SOX4+ nuclei in end-stage cholangiocytes and hepatocytes, whereas SOX4 was not observed in healthy liver (Fig. 3l and Extended Data Fig. 8a). Similarly, K23+ cholangiocytes were observed in end-stage disease, with some cells co-expressing K19, ALB and HepPar1 (Extended Data Fig. 8b). By contrast, K23 was not found in healthy cholangiocytes.

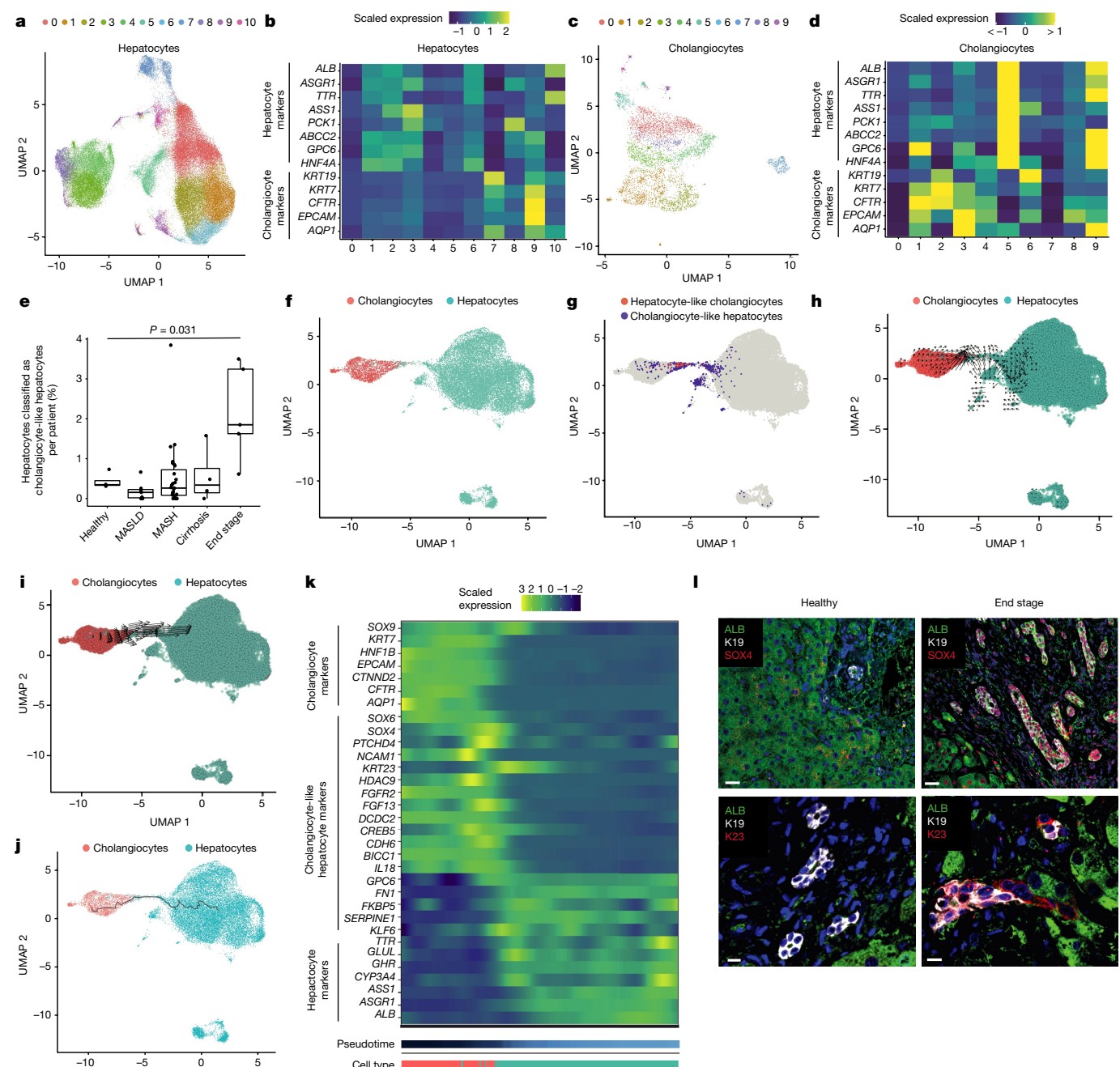

**Fig. 3 | Using snRNA-seq identifies cholangiocyte and hepatocyte plasticity. a**, Subclustering of hepatocytes. **b**, Relative expression of hepatocyte and cholangiocyte markers across hepatocyte subclusters. **c**, Subclustering of cholangiocytes. **d**, Relative expression of hepatocyte and cholangiocyte markers across cholangiocyte subclusters. **e**, Proportion of hepatocytes classified as cholangiocyte-like hepatocytes (identified by subclustering hepatocyte cluster 9 in **b**) that are expressing cholangiocyte markers, by disease stage. Statistical significance was calculated using two-sided Welch's *t*-test (*n* = 47 biologically independent donors: healthy, 4; NAFLD, 7; NASH, 27; cirrhosis, 4; end stage, 5). The *P* value was 0.03058 (significant under a 0.05 threshold). The mid-point, minimum and maximum of the boxplot summary correspond to the median, first and third quartiles. The extent of the whiskers corresponds to the largest and smallest values no further than 1.5 IQR from the inter-quartile range. **f**, UMAP of cholangiocytes and hepatocytes from end-stage MASLD disease only. **g**, Cholangiocyte-like hepatocytes and hepatocyte-like cholangiocytes (identified by subclustering hepatocyte cluster 9 in **b** and subclustering cholangiocyte cluster 1 in **d**, respectively) that express hepatocyte markers are plotted to show their location on the UMAP. **h**, RNA velocity using cholangiocyte-like hepatocytes. **i**, RNA velocity using hepatocyte-like cholangiocytes. **j**, Pseudotime trajectory across the connected region of the two cell types. **k**, Heat map of DEGs (differentially expressed genes) across the trajectory. **l**, Immunofluorescence staining of the cholangiocyte-like hepatocyte markers SOX4 and K23 alongside ALB and K19 in healthy and end-stage sections; *n* = 3 healthy and *n* = 3 end-stage tissue samples. Scale bars: 30 μm in the SOX4 images (top); 10 μm in the K23 images (bottom).

Cells co-expressing SOX4 and K23 were also observed (Extended Data Fig. 8a). Similar staining patterns were found for KLF6 and NCAM1, with clear increases in end-stage disease (Extended Data Fig. 8c–e). Notably, analysis of proliferation in the biphenotypic population identified some cells co-positive for *MKI67* and *SOX4* (*n* = 4 of 16 proliferative cells) or *KRT23* (*n* = 2 of 16 proliferative cells), indicating that transdiff-erentiation may be associated with cell division (Extended Data Fig. 8f). Together, these observations demonstrate that our single-nucleus

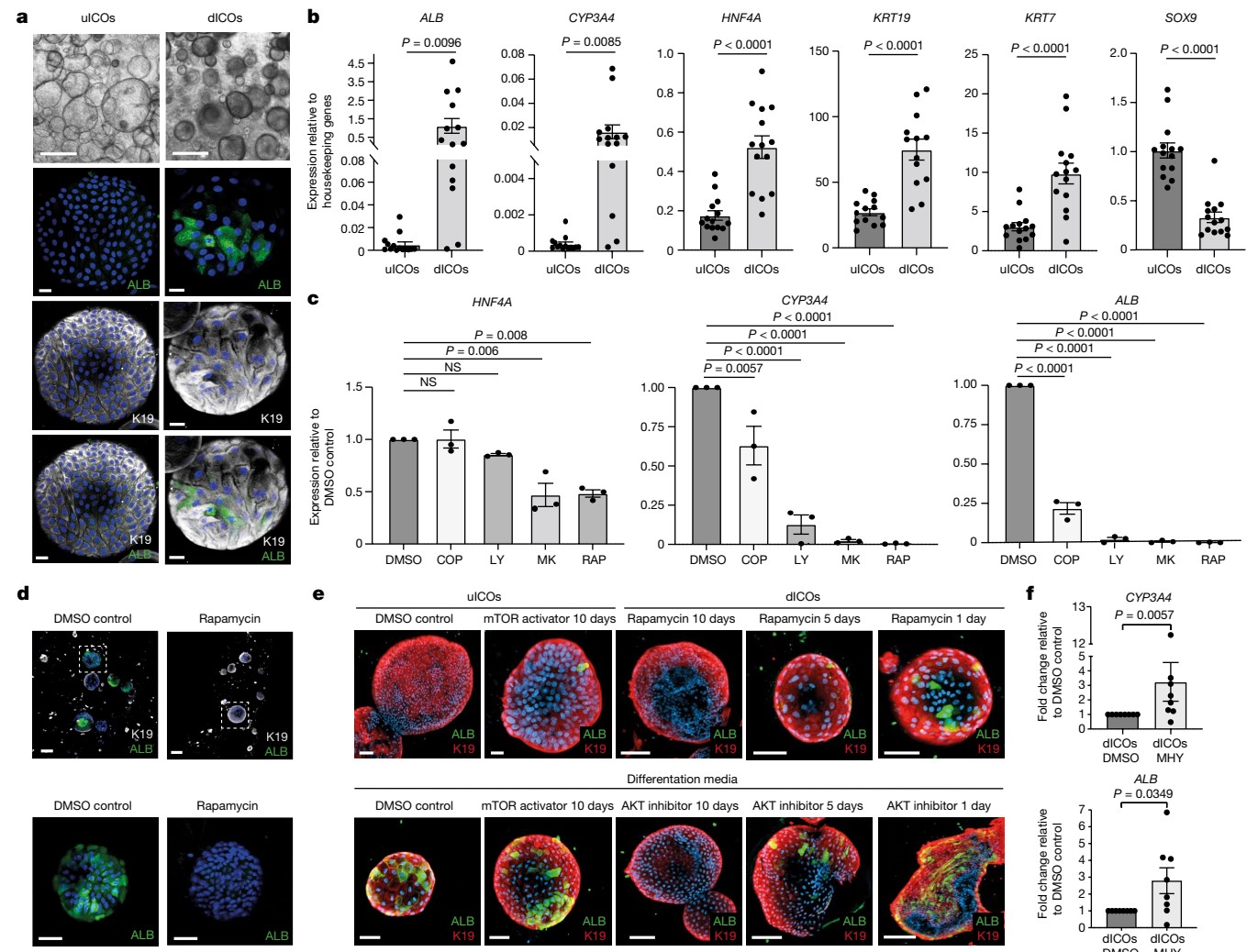

**Fig. 4 | The PI3K–AKT–mTOR pathway is a key regulator of cholangiocyte-to-hepatocyte plasticity. a**, Bright-field images and immunofluorescence staining of organoids treated with cholangiocyte organoid medium (uICO) or with differentiation medium (dICO) for ALB and K19; *n* = 6 patient-derived organoid lines. Scale bars: 500 μm, bright-field; 20 μm, immunofluorescence. **b**, mRNA expression of hepatocyte markers (*ALB*, *CYP3A4* and *HNF4A*) and cholangiocyte markers (*KRT19*, *KRT7* and *SOX9*) in uICOs and dICOs; *n* = 14 biologically independent experiments (unpaired two-tailed *t*-test; errors bars indicate s.e.m.). **c**, mRNA expression of hepatocyte markers in organoids differentiated in the presence of DMSO, copanlisib (a PI3K inhibitor), LY294002 (a PI3K inhibitor), MK2206 (an AKT inhibitor) or rapamycin (an mTOR inhibitor); *n* = 3 biologically independent experiments (*P* values indicated, ordinary one-way ANOVA, adjusted for multiple comparisons; errors bars show mean ± s.d.). **d**, Immunofluorescence staining for K19 and ALB in organoids differentiated in the presence of DMSO or the mTOR inhibitor rapamycin (top); bottom image shows high magnification of the area in the white box; *n* = 3 patient-derived organoid lines. Scale bars: 100 μm, top; 50 μm, bottom. **e**, Immunofluorescence staining for K19 and ALB in uICOs and dICOs treated with DMSO, an mTOR activator (MHY1485), an AKT inhibitor (MK2206) or an mTOR inhibitor (rapamycin) for the number of days indicated; *n* = 3 patient organoid lines. Scale bars: top, left to right: 100 μm, 40 μm, 100 μm, 70 μm, 70 μm; bottom, left to right: 40 μm, 80 μm, 100 μm, 100 μm, 100 μm. **f**, mRNA expression of the hepatocyte markers *ALB* and *CYP3A4* in dICOs treated for 10 days with DMSO or an mTOR activator (MHY); *n* = 8 biologically independent experiments (*P* values indicated, two-tailed *t*-test; error bars indicate s.e.m.).

analysis has identified factors that mark transdifferentiating cells in end-stage MASLD and could be relevant for monitoring disease progression.

## PI3K–AKT signalling regulates plasticity

Interestingly, GSEA of the genes specific to biphenotypic cells indicated an enrichment in Gene Ontology terms for processes such as cell differentiation, and in KEGG terms including tight junction and PI3K–AKT signalling (Extended Data Fig. 9a). This pathway has been associated with obesity and metabolic syndrome[48], both of which are tightly linked to MASLD[49]. To further investigate the functional importance of the PI3K–AKT pathway in the molecular mechanisms regulating cholangiocyte and hepatocyte plasticity, we decided to take advantage of intra-hepatic cholangiocyte organoids (ICOs). These cells can be grown for an extended period of time in vitro and maintain their biliary identity[50] and their capacity to differentiate into cells expressing hepatocyte markers[26]. We first generated ICOs from end-stage MASLD livers (Supplementary Table 4). These cells expressed K19, confirming their identity (Fig. 4a, left). MASLD ICOs were then differentiated towards cells expressing hepatocyte markers, as previously described[26]. As expected, the resulting organoids contained cells positive for ALB (Fig. 4a, right), and quantitative PCR (qPCR) showed that they display increased expression of the genes *CYP3A4*, *HNF4A* and *ALB* (Fig. 4b). However, only some of the cells in an organoid become ALB+ (Fig. 4a), confirming previous observations that this process is heterogenous[26].

Notably, ALB$^+$ cells were also K19$^+$, indicating a biphenotypic identity. Expression of cholangiocyte markers K7 and K19 was also found to increase, but expression of the cholangiocyte transcription factor gene *SOX9* decreased (Fig. 4b), indicating that the biliary nature of the cells was mainly maintained. Notably, we also performed differentiation using ICOs derived from healthy and end-stage MASLD livers in parallel and found no difference in their capacity for differentiation (Extended Data Fig. 9b), indicating that our culture conditions can induce cellular plasticity without disease environment. More importantly, qPCR analyses showed that several genes associated with biphenotypic cells in vivo also increased during ICO differentiation, including *SOX4* and *KRT23* (Extended Data Fig. 9c). Thus, ICOs differentiated in vitro provide a model for the transdifferentiation events observed in vivo (Fig. 3).

We next used ICOs to validate the importance of PI3K–AKT signalling. ICOs differentiated in the presence of the mTOR inhibitor rapamycin, the PI3K inhibitors LY294002 and copanlisib, and the AKT inhibitor MK2206 displayed a strong reduction in hepatocyte marker expression (Fig. 4c,d). Furthermore, differentiation of ICOs in the presence of the mTOR activator MHY1485 enhanced differentiation (Fig. 4e,f), indicating that this pathway can increase the expression of hepatocyte markers in cholangiocytes. Finally, inhibition of mTOR, PI3K or AKT blocked differentiation when applied at the start of the differentiation (10 days of treatment) but had less or no effect when applied from the half-way point (5 days of treatment) or just for the final 24 h, respectively (Fig. 4e and Extended Data Fig. 9d). Together these data indicate that the mTOR–PI3K–AKT pathway could be necessary for cholangiocytes to differentiate into biphenotypic cells, but not for the survival of these cells. Importantly, the PI3K–AKT–mTOR pathway is activated by insulin[51], and insulin resistance is commonly associated with MASLD progression[30]. We measured the serum insulin levels of patients across the disease stages and observed a sharp increase in all stages compared with controls, with levels highest at the stage of cirrhosis (Extended Data Fig. 9e). Taken together, these findings indicate that increased circulating insulin during disease progression could have a key role through the PI3K–AKT–mTOR pathway in inducing plasticity in the hepatic epithelium. However, our single-cell analyses also indicated that the acquisition of plasticity is progressive and occurs only after large changes in the liver microenvironment. Thus, we hypothesized that the PI3K–AKT–mTOR pathway may be one of various pathways involved and decided to test for other pathways in vitro. We first identified *FGF13* as being upregulated in biphenotypic cells (Fig. 3k), and we found that differentiation of ICOs in the presence of FGF13 caused a limited increase in hepatocyte marker expression (Extended Data Fig. 10a). We also performed differentiation in the presence of the proinflammatory cytokine TWEAK and fatty acids, because both play a role in MASLD progression[52,53], and found no change in hepatocyte marker expression (Extended Data Fig. 10b,c). Finally, we observed increased expression in YAP-signalling genes in cholangiocytes and hepatocytes from end-stage livers. We therefore performed differentiation in the presence of a YAP activator and observed a strong decrease in the expression of hepatocyte marker genes (Extended Data Fig. 10d,e), indicating that the YAP–TAZ pathway could limit cholangiocyte plasticity but promote the ductular reaction, as shown in mouse studies[54,55]. Finally, we performed differentiation of ICOs in a matrix containing an increased amount of collagen to mimic more closely the cirrhotic liver environment. Strikingly, this change in the composition of the extracellular matrix caused organoid branching and the appearance of ALB$^+$ cells in tubular K19$^+$ structures (Extended Data Fig. 10f). Thus, changes in the composition of the extracellular matrix may instruct tubulogenesis, which resembles the ductular reaction, but without substantially improving transdifferentiation. Taken together, these data suggest that the acquisition of plasticity could involve complex interplays between different signalling pathways, including the YAP and PI3K–AKT–mTOR pathways.

## Discussion

Our single-cell analyses provide an advanced resource to study the factors driving disease progression. The use of snRNA-seq, as opposed to scRNA-seq, allows the unbiased capture of hepatocytes and cholangiocytes without the over-representation of immune cells, which has been reported in scRNA-seq studies. Comparison of the two approaches for human liver indicates that snRNA-seq also enhances the detection rate of rare populations[56]. These benefits, plus the suitability of snRNA-seq for processing frozen biopsies, made this approach suitable for our study and aims. The information contained in this dataset certainly goes beyond mechanisms of regeneration, and subsequent analyses will probably reveal new cellular activity involving more cell types. However, we decided to focus on the regenerative process because this aspect is difficult to investigate in human tissue and could have profound implications for organs targeted by progressive disorders. By combining snRNA-seq with advanced imaging of tissue, we showed that cellular plasticity between cholangiocytes and hepatocytes increases with disease progression to culminate during end-stage disease. This finding supports the results from animal studies, which have reported cholangiocyte-to-hepatocyte plasticity[1,16–19]. Furthermore, our analysis builds on histological observations of intermediate hepatocytes (K7-expressing hepatocyte-like cells) in MASLD[28] by showing that transdifferentiating cells, and thus truly biphenotypic cells, are found mainly in end-stage liver. These biphenotypic cells are different from the hepatobiliary hybrid progenitors previously identified[57], which are present only in healthy tissue. Furthermore, we could not find evidence of liver stem cells or dedifferentiation processes. However, single-cell data resolution can be a limitation and we are unable to totally exclude the existence of a rare population of adult stem cells in the liver. Such cells could hypothetically be activated by other types of injury. The resolution of our dataset was sufficient to capture cells representing the ductular reaction and intermediate hepatocytes during the early stage of the disease. Although lacking the expression of plasticity factors, these cells share a transcriptional signature with the biphenotypic cells identified in our study because they express markers of both hepatocytes and cholangiocytes. Thus, our data do not exclude the possibility that the ductular reaction or intermediate hepatocytes could represent early precursors necessary for the production of biphenotypic cells in end-stage liver. More importantly, our analyses revealed that transdifferentiation might not be a real event in regeneration. Indeed, transdifferentiating cells were observed mainly in end-stage livers, which display little function, represent a damaged environment and have a high incidence of liver cancer. Thus, although we cannot rule out a regenerative effort, the acquisition of plasticity represents a disease process, rather than a repair mechanism. This hypothesis is reinforced by the major changes occurring in the niche surrounding hepatocytes and cholangiocytes, as evidenced by the abnormal zonation, the loss of cellular identity and the aberrant remodelling of the biliary tree, all of which are difficult to associate with a healthy regenerative process. Interestingly, expression of *SOX4*, *KLF6* and *KRT23* has been associated not only with liver steatosis[58,59] and biliary remodelling[60,61], but also with hepatocellular carcinoma[62,63]. Finally, our data indicate a role for the PI3K–AKT–mTOR pathway in regulating cholangiocyte-to-hepatocyte transdifferentiation. This pathway has previously been implicated in the conversion of biliary epithelial cells to hepatocytes in zebrafish[64], which may suggest that this mechanism is conserved between species. The involvement of the insulin signalling pathway in the regulation of plasticity also highlights a potential role for insulin resistance, which is commonly associated with an increased risk of cancer. Thus, the plasticity observed in the liver could reflect a broader mechanism occurring in several organs of MASLD patients with type 2 diabetes. Future work investigating the interplay of insulin resistance and cellular plasticity may address this important question. Our data also indicate that the PI3K–AKT–mTOR pathway is probably one of various pathways involved

in plasticity. Thus, the acquisition of cellular plasticity in human epithelium is likely to be a disease mechanism involving multiple signals and modifications of the microenvironment over a prolonged period of time. Consequently, a deeper understanding of the signals controlling the appearance of plasticity could pave the way for the development of efficient and safe therapeutic strategies against chronic liver diseases.

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

## Methods

### Ethics

Biopsy collection and processing of human samples were carried out under ethics approval by Addenbrookes Hospital REC 18/WM/0397. The study met all the UK criteria for the responsible use of human tissue. Every donor whose samples were used was offered the patient information sheet and provided informed consent. Healthy deceased transplant organ tissue and explants were taken under ethics approval by the National Research Ethics Service Committee East of England - Cambridge South (REC number REC 15/EE/152).

### Tissue collection and freezing

Liver biopsies were done with ultrasound guidance using a 16 g end cut needle (Biopince). Two ultrasound-guided needle core liver biopsies of approximately 2 cm were obtained. Half of the second biopsy (1 cm) was placed in a cryo-vial and frozen immediately using liquid nitrogen. For healthy donors and explant tissue, a cube of approximately 1 cm$^3$ was cut and frozen as above. For two healthy donors (Hl1 and HL3) and all end-stage patients, samples were taken from each of the three liver lobes (left, right and caudate), so these individuals contributed three samples to the dataset. Samples were then stored at −80 °C. Details of patient demographics and disease staging are included in Supplementary Tables 1 and 2.

### Nucleus isolation

Frozen samples were transferred to a Dounce homogenizer and lysed in 1 ml lysis buffer (IGEPAL 0.1%, NaCl 10 mM, Tris-HCL pH 7.5 10 mM, MgCl$_2$ 3 mM in nuclease-free water supplemented with 0.2 U μl$^{-1}$ RNasin plus). Lysis was done by performing five strokes with part A and 10–15 strokes with part B on ice, with 2 min incubation on ice between using parts A and B. After a further 2 min on ice, the sample was mixed using a P1000 by pipetting up and down ten times before a further 1 min on ice. The sample was then passed through a pre-wet 40 μm cell strainer, transferred to a 1.5 ml low-bind microfuge tube and centrifuged at 500$g$ for 5 min at 4 °C. The pellet was resuspended in 1 ml wash buffer (Ultrapure BSA 1% in tissue-culture grade supplemented with 0.2 U μl$^{-1}$ RNasin plus) and centrifuged at 500$g$ for 5 min at 4 °C. The pellet was resuspended in 400 μl wash buffer and transferred to a tube for FACS and kept on ice, and the sample was treated with 3 μM DAPI. FACS sorting was performed on an Influx or Aria Fusion cell sorter. Nuclei were defined by strict FSC (forward scatter) and SSC (side scatter) gating to remove debris and intact cells (larger events on the FSC). A strict singlet gate was applied and nuclei were sorted in high-purity mode with the sorter precooled. Then 20,000 DAPI-positive nuclei were sorted into a 1.5 ml microfuge tube containing 500 μl wash buffer and the tube was topped up and centrifuged at 500$g$ for 5 min at 4 °C. The pellet was resuspended in 43 μl wash buffer and kept on ice until loading on the 10x chromium. As part of the protocol optimization, a series of lysis buffers and incubation times were tested and lysis was examined using Trypan blue and a cell counter, with efficient lysis showing more than 95% lysed cells before sorting. After sorting, nuclei were examined to ensure a single nuclei suspension of intact nuclei (nuclear membrane intact with minimal blebbing).

### Single-nucleus RNA-seq

Single-nucleus RNA-seq libraries were prepared using the following: Chromium Single Cell 3′ Library and Gel Bead Kit v.3.1, Chromium Chip G Kit and Chromium Single Cell 3′ Reagent Kits v.3.1 User Guide (manual part CG000316 Rev A; 10x Genomics). One sample was run per lane of the 10x chip. For each sample, 16,000 nuclei were loaded on the Chromium instrument with the expectation of collecting gel–bead emulsions containing cell nuclei. RNA from the barcoded nuclei for each sample was subsequently reverse-transcribed in a C1000 Touch Thermal cycler (Bio-Rad) and all subsequent steps to generate single-nuclei

libraries were done according to the manufacturer's protocol with 19 PCR cycles in the cDNA amplification step. cDNA quality and quantity were measured using Agilent TapeStation 4200 (High Sensitivity 5000 ScreenTape) after which 25% of the material was used to prepare the gene-expression library. Library quality was confirmed with Agilent TapeStation 4200 (High Sensitivity D1000 ScreenTape to evaluate library sizes) and Qubit 4.0 Fluorometer (ThermoFisher Qubit dsDNA HS Assay Kit to evaluate the double-stranded DNA quantity). Each sample was normalized and pooled in equal molar concentrations. To confirm the concentration of the pool we performed qPCR using a KAPA Library Quantification Kit on QuantStudio 6 Flex before sequencing. The pool was sequenced on an Illumina NovaSeq6000 sequencer with the following parameters: 28 base pairs (bp), read 1; 10 bp, i5 index; 10 bp, i7 index; 90 bp, read 2.

### FLASH imaging

FLASH was performed as described[65]. Samples were fixed overnight in 4% PFA at 4 °C. The sample was transferred to PBS and sliced using a vibratome to generate slices 500 μm thick. Depigmentation was performed by incubating samples in DMSO and H$_2$O$_2$ in PBS in a 1:1:4 (by volume) ratio overnight. The next day, samples were washed briefly in PBS and transferred to an antigen-retrieval solution. To prepare the antigen-retrieval solution, urea was dissolved in 200 mM boric acid to 250 g l$^{-1}$. Zwittergent was then dissolved in the urea–borate solution to 80 g l$^{-1}$. Samples were incubated in 1 ml of the solution in a 2 ml microcentrifuge tube at room temperature for 1 h, then left overnight at 54 °C with gentle mixing on a thermo-mixer. The next day, samples were washed in PBT (0.2% Triton X-100 in PBS) three times for 1 h at room temperature before being moved to blocking buffer (1% BSA, 5% DMSO, 10% FCS and 0.2% Triton X-100) in PBS and incubated overnight at room temperature. Primary antibodies were then incubated in blocking buffer (dilution 1:100) for at least 2 nights at room temperature on a nutator. Samples were washed in PBT three times for 1 h per wash before fluorophore-conjugated secondary antibodies were added for two nights (dilution of 1:200) at room temperature on a nutator. Samples were then washed in PBS three times for 30 min per wash and passed through a dehydration series of 30%, 50%, 75% and then 2 × 100% methanol for at least 30 min in each solution, protected from light. Dehydrated samples were then gradually cleared by submerging in methyl salicylate diluted in methanol at 25%, 50%, 75% and 2 × 100% methyl salicylate for at least 30 min each in a glass dish protected from light. Cleared samples were then mounted on a glass slide in 100% methyl salicylate. Samples were imaged using an upright LSM 880 microscope, using 10× and 20× water immersion lenses.

### Immunofluorescence staining of tissue slides

For all tissue-staining experiments, multiple tissue sections from at least four different patients of the relevant disease stage were analysed. Slides were dewaxed in HistoClear twice for 5 min before being washed in 100% ethanol for 5 min. Slides were then passed through a rehydration series for 5 min of 95%, 90%, 80% and 50% ethanol, then distilled water. Heat-mediated antigen retrieval was performed using 10 mM citrate (pH 6.2). The buffer was pre-warmed in a microwave until gently bubbling, before slides were submerged and heated for 15 min in the microwave on 50% power to maintain gentle bubbling. Slides were then cooled and washed twice briefly in PBS. Slides were incubated in blocking solution containing 1% (w/v) BSA, 5% (v/v) donkey serum and 0.1% (v/v) Triton X-100 for 30 min at room temperature in a humidified chamber. Primary antibodies were then diluted in blocking solution (all at 1:100 dilution except for anti-SOX4, which was used at 1:50) and incubated overnight at 4 °C in a humidified chamber. The next day, slides were washed three times for 15 min in PBS before fluorophore-conjugated secondary antibodies (1:500 dilution) plus DAPI were applied for 1 h at room temperature in the humidified chamber. Then, slides were washed three times for 15 min in PBS. Slides were

mounted in one drop of DAKO fluorescent mounting medium. Slides were imaged using a Zeiss inverted 710 confocal microscope.

## Organoid derivation

Tissue was stored at 4 °C in basal medium (Advanced DMEM/F12, 1% Glutamax, 1% HEPES, 1% penicillin-streptomycin) after retrieval and derivation was attempted within 24 h of tissue storage. Tissue was minced with a scalpel or scissors to small pieces of less than 1 mm³ in basal medium. The minced tissue was transferred to a 50 ml conical tube with enough digestion medium (collagenase D 2.5 mg ml⁻¹ and DNAse I 0.1 mg ml⁻¹ in HBSS) to fully cover it and placed in a water-bath at 37 °C for 70 min with pipetting to mix every 10 min. Cold wash medium (DMEM, 1% Glutamax, 1% FBS, 1% pen-strep) was added to stop the digestion and the sample was centrifuged at 400$g$ for 4 min. The pellet was resuspended in 5 ml wash medium and centrifuged again as before. The resulting pellet was then resuspended in growth-factor-reduced Matrigel and plated in 50-µl domes on a 24-well plate. The plate was incubated at 37 °C for 15 min before 500 µl isolation medium was added (Advanced DMEM/F12, 1% Glutamax, 1% HEPES, 1% pen-strep, 1% B27 without vitamin A, 1% N2 supplement, 10% conditional RSPO medium, 30% WNT-conditioned medium, 25 ng ml⁻¹ Noggin, 100 ng ml⁻¹ FGF10, 25 ng ml⁻¹ HGF, 50 ng ml⁻¹ EGF, 10 mM nicotinamide 0.4 M, 10 nM gastrin, 1 mM N-acetyl cysteine, 10 µM FSK, 5 µM A8301, Noggin, 10 µM Y27632). Details of patient demographics are included in Supplementary Table 4.

## Organoid culturing

After organoid derivation, the medium was changed from isolation medium to expansion medium (isolation medium without Y27632, Noggin and WNT-conditioned medium). Organoids were typically passaged every 7–10 days and the medium was changed every 2–3 days. For splitting organoids, the medium was replaced with 500 µl Cell Recovery solution (Corning). The Matrigel dome was scraped and collected using a P1000 and incubated on ice for 20 min. This was then spun at 400$g$ for 4 min and the pellet was resuspended in basal medium using a P1000 to break up the organoids, before being centrifuged as before. The pellet was resuspended in an appropriate volume of Matrigel and plated in 50-µl domes in a 24-well plate. The plate was incubated at 37 °C for 15 min before 500 µl expansion medium was added.

## Differentiation of organoids

Cholangiocyte organoids were split into expansion medium with the addition of 25 ng ml⁻¹ BMP7 for 5 days (medium renewed every 2–3 days). Organoids were then passaged as above and plated into differentiation medium for additional 10 days and renewed every 2–3 days (Advanced DMEM/F12, 1% Glutamax, 1% HEPES, 1% pen-strep, 1% B27 without vitamin A, 1% N2 supplement, 25 ng ml⁻¹ HGF, 50 ng ml⁻¹ EGF, 10 nM Gastrin, 1mM N-acetyl cysteine, 0.5 µM A8301, 100 ng ml⁻¹ FGF19, 10 µM DAPT, 3 µM dexamethasone, 25 ng ml⁻¹ BMP7).

## In vitro treatments

Cholangiocyte organoids were treated with expansion medium and BMP7 for 5 days. They were then passaged directly into differentiation medium (as above) with the addition of the small molecule of interest per condition (10 µM LY294002, 20 nM copanlisib, 1 µM MK-2206, 100 nM rapamycin, 10 µM MHY1485) for a total of 10 days. The medium was renewed every 2–3 days. For the time-course experiment, inhibitors were applied only at the time point indicated in the figure. For experiments in which organoids were cultured in increased collagen, the cell pellet was resuspended in a 50:50 mix of Matrigel and collagen I with NaOH added to neutralize the collagen before resuspending the cells. Organoids were then cultured as described above.

## Immunofluorescence staining of organoids

Organoids that were planned for immunofluorescence staining were plated after splitting in a µ-Slide 8 Well High Glass Bottom (Ibidi) for better imaging quality. For staining on 3D organoid cultures, cells were washed with PBS once and then incubated with 4% PFA–PBS for 20 min at room temperature. After incubation, cells were washed three times with PBS and stored in PBS at 4 °C for up to a month. For intracellular epitopes, organoids were permeabilized using a solution of 10% donkey serum in PBS plus 0.3% Triton X-100 for at least 3 h. Cells were incubated with the primary antibody (1:100 dilution) in 1% donkey serum plus 0.1% Triton X-100 at 4 °C overnight. Cells were washed with PBS three times at room temperature for 1 h per wash. Then, cells were incubated with secondary antibody diluted 1:1000 in 1% donkey serum plus 0.1% Triton X-100 at 4 °C overnight. Cells were washed with PBS three times at room temperature for 1 h per wash. Cells were stained with Hoechst dye at 1:10,000 dilution in PBS for 30 min and washed twice. Cells were stored in PBS at 4 °C for up to a month. A Zeiss LSM 710 confocal microscope was used for imaging.

## Collagen and haematoxylin-and-eosin staining

Collagen (Picro Sirius Red) staining and haematoxylin-and-eosin staining was done by the Department of Pathology at Addenbrookes Hospital in Cambridge according to their local protocol.

## Statistical analysis of qPCR

Unpaired $t$-tests were used to perform statistical analysis on the qPCR, comparing uICOs and dICOs. One-way ANOVA adjusted for multiple comparisons was used to analyse in vitro treatments of organoids and patient insulin serum level. $P$ values are indicated in the figure legends.

## Computational methods

**Sample quantification.** The samples were mapped and the expression levels summarized using 10x Genomics CellRanger v.5.0.0 (ref. 66) against version GRCh38.p13 of the *H. sapiens* genome. To accommodate the characteristics of single-nucleus data, that is, a higher proportion of reads mapped to introns, the option '--include-introns' was enabled.

**Quality control.** Seurat (v.4.0.3)[67] objects were created considering genes expressed in more than three cells, and cells with more than 200 features expressed. Barcodes (nuclei) were excluded that had less than 1,000 or less than 800 features, or for which more than 10% of counts mapped to mitochondrial or ribosomal genes. To remove potential doublets, nuclei with more than 50,000 counts were also removed; the nCount, nFeature, %MT and %RP distributions per patient were visualized. After filtering, mitochondrial and ribosomal protein-coding genes were removed from the dataset, resulting in a dataset of 99,809 cells and 31,257 features across 47 samples.

**Preprocessing.** The preprocessing of raw count matrices was performed using Seurat (v.4.0.3)[67]. Gene-expression values were normalized for library size using sctransform[68]. Principal component analysis was carried out using the top 3,000 highly variable genes. Neighbours were identified using the first 50 principal components and clustering was done using the Louvain algorithm with the 20 nearest neighbours per cell. UMAP projections were calculated using 'RunUMAP(n.neighbors = 20, min.dist = 0.3)'. The clustering parameters used were identified by evaluating the resulting cluster stability using ClustAssess[69]

**Annotation of cells.** Expression of cell-type marker genes (Supplementary Table 3 and Extended Data Fig. 3e) was used to assign cell-type labels.

**Data integration.** For hepatocyte and cholangiocyte cells, some sample-specific segregation was observed within each disease stage. To alleviate potential batch effects, the data were integrated using Harmony[40] with default parameters except $\theta$ (the diversity clustering penalty parameter), which was minimized such that within each disease stage, all recovered clusters included cells from each patient.

**Differential expression analysis.** Genes differentially expressed between cell groups were identified using the Seurat FindMarkers function. Differentially expressed genes were called on: abs($\log_2$FC) > 0.5, Benjamini–Hochberg corrected $P$ < 0.05 and a minimum of 25% of cells expressing the gene in the higher-expression group. GSEA of differentially expressed genes was carried out using gprofiler2 (v.0.2.0)[70] using all genes detected in the compared cell groups as the background set. Enrichment was tested on the standard Gene Ontology terms, KEGG and Reactome pathway databases, and the microRNA and TF regulatory features. The Benjamini–Hochberg correction for multiple testing was applied to GSEA $P$ values.

Cells labelled as positive for one or multiple genes are those with SCtransform-normalized expression greater than 0, per gene. The proportion of biphenotypic cells across each disease stage was compared using Welch's $t$-test. Loss of zonation through disease was assessed by comparing the correlation between pairs of periportal and pericentral markers, which were then contrasted using Welch's $t$-test.

**RNA velocity.** Velocyto (v.0.17.17), and velocyto.R (v.0.6) were used to estimate RNA velocity on the basis of the prevalence of spliced and unspliced mRNA[71]. Velocyto run10× was run using GRCh38.p13 annotation and repeat mask. The dataset was randomly downsampled to 20,000 cells, and cell distance was calculated as 1 minus correlation in the first 50 principal components. RNA velocity was estimated using 'gene.relative.velocity.estimates(deltaT = 1, kCells = 20, cell.dist = D, fit.quantile = 0.2)'.

### Reporting summary

Further information on research design is available in the Nature Portfolio Reporting Summary linked to this article.

### Data availability

The sequencing data and raw expression matrix are available on the Gene Expression Omnibus, series entry GSE202379. R Shiny apps illustrating the analysis are at https://www.mohorianulab.org/shiny/vallier/LiverPlasticity_GribbenGalanakis2024/. All other data are available from the corresponding author(s) upon reasonable request. Source data are provided with this paper.

### Code availability

Scripts for all bioinformatics analyses carried out are available at https://github.com/Core-Bioinformatics/MASLD-NASH.

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

**Acknowledgements** This research was supported by the Cambridge NIHR BRC Cell Phenotyping Hub, the Cambridge Stem Cell Institute imaging facility, genomics facility and core bioinformatics group and the Cambridge CRUK Institute genomics facility, with thanks to K. Kania for snRNA-seq and sequencing support. We thank the Addenbrookes Hospital radiology ultrasound department for support with biopsy collection and the pathology department for histology slide preparation and staining. This project was funded by Open Targets grant OTAR2051, the Wellcome Leap (ref: HOPE), an Einstein Foundation professorship, a BIH core grant, the ERC (New-Chol 741707), the Wellcome Trust (203151/Z/16/Z), the UKRI Medical Research Council (MC_PC_17230) and an MRC-DTP iCASE PhD studentship award, jointly funded by AstraZeneca (G117817). This research was funded in whole, or in part, by the Wellcome Trust (203151/Z/16/Z, 203151/A/16/Z) and the UKRI Medical Research Council (MC_PC_17230).

**Author contributions** C.G. did experimental design, single-nucleus method optimization, single-nucleus isolations, staining and imaging of histology sections and 3D tissue, organoid experiments, data interpretation and manuscript preparation. V.G. did experimental design, biopsy preservation, collection of end-stage samples, staining and imaging of histology sections and 3D tissue, organoid experiments, data interpretation and manuscript preparation. A.C. and E.C.W performed the bioinformatics analysis of the single-nucleus sequencing data. R.C.-G. performed earlier-stage bioinformatics analysis (single-nucleus sequencing data). M.L. performed bioinformatics analysis using the patient dataset for revisions. C.F. performed staining and imaging of histology sections. T.P. and A.G. provided mouse tissue and performed tissue sectioning. F.J.R. collected end-stage liver samples. K.M. collected healthy donor samples. E.G. performed ultrasound-guided collection of biopsies. S.E.D. conducted histopathology analysis. E.A. assisted in interpretation of bioinformatic analysis. K.S.-P. collected donor samples. F.T. oversaw revision experiments involving mouse tissue and interpreted data. M.A. conceived the study, oversaw the MASLD clinic and assisted with experimental design and interpretation of the data. I.M. contributed to the experimental design, supervised all bioinformatics analysis and interpreted the outputs. L.V. conceived the study, performed experimental design, interpreted data and prepared the manuscript.

**Competing interests** The authors declare no competing interests.

**Additional information**
**Correspondence and requests for materials** should be addressed to Michael Allison, Irina Mohorianu or Ludovic Vallier.

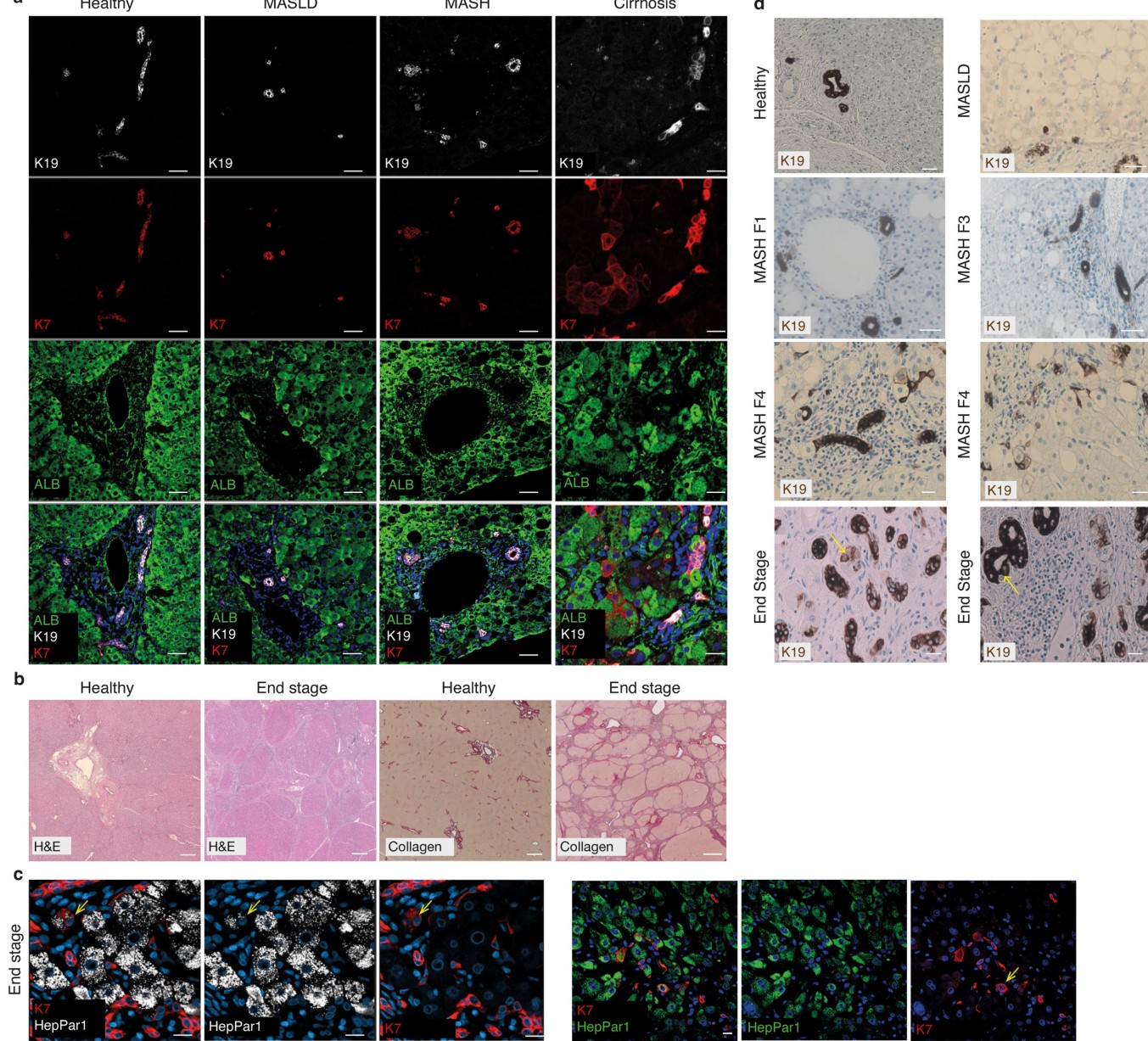

**Extended Data Fig. 1 | Biphenotypic cells are observed in late stages of the disease progression. a)** Immunofluorescent staining for ALB, K19 and K7 on tissue sections from healthy, MASLD, MASH and Cirrhosis staging. Scale bars = 50 um for Healthy MASLD and MASH panels and 20um for cirrhosis panels **b)** H&E and collagen staining of healthy and end stage MASLD tissue sections. Scale bars = 500 um Healthy H&E, 1000 um End stage H&E, 2000 um for Healthy and End stage collagen staining. **c)** Staining for KRT7 and HepPar1 in end stage liver. An example of a double positive cell is indicated (yellow arrow).

Scale bars = 20 um. **d)** Immunohistochemistry staining for K19 on tissue section of the indicated disease stage. In end stage (bottom panels), a cell with hepatocyte morphology expressing low levels of K19 is indicated in the left panel (yellow arrow) and a cell negative for K19 within a duct with hepatocyte morphology is indicated in the right panel (yellow arrow). Scale bars = 50 um (4 upper panels) and 20 um (4 lower panels). n = 3 patient samples from each indicated disease stage in (a-d).

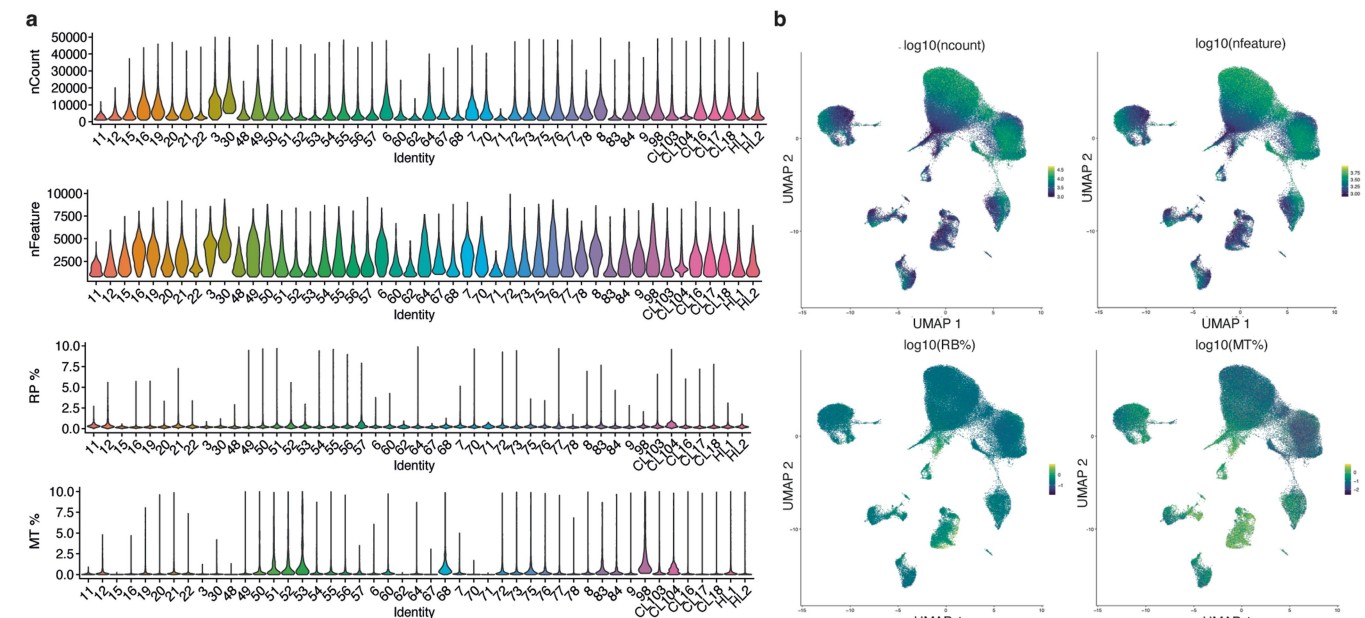

**Extended Data Fig. 2 | QC showing that snRNAseq protocol generates high quality data. a**) Violin plots summarising the number of UMIs detected per cell (nCount), number of genes per cell (nFeature), proportions of reads incident to ribosomal genes (rp%) and mitochondrial genes (mt%) per patient. **b**) gradient of ncount, nfeatures, mt%, rp%, all presented on log$_{10}$ scale displayed on the expression-driven UMAP, for all samples.

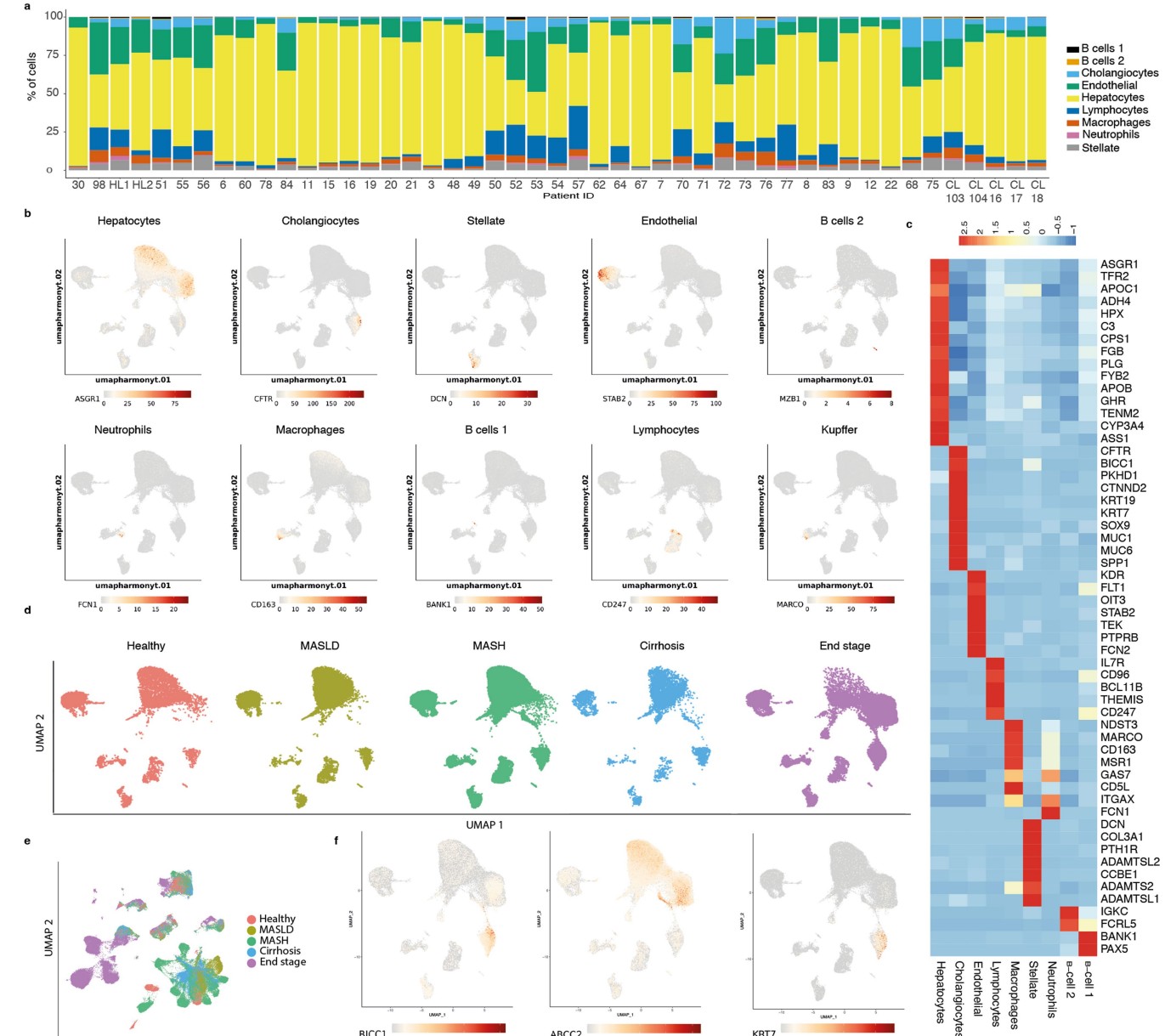

**Extended Data Fig. 3 | snRNAseq captures all liver cell types across disease progression. a**) Proportions of cells, captured per patient, assigned to each cell type. **b**) Expression UMAPs of cell type markers corresponding to Fig. 1e shown on overall UMAP. **c**) Heatmap of relative expression of various markers used in the annotation of the cell types. **d**) Overall UMAP facetted by disease stage. **e**) Uncorrected overall UMAP show by disease stage. **f**) Expression UMAPs for cholangiocyte markers *KRT7* and *BICC1*, and hepatocyte marker ABCC2 (MRP2) on overall umap.

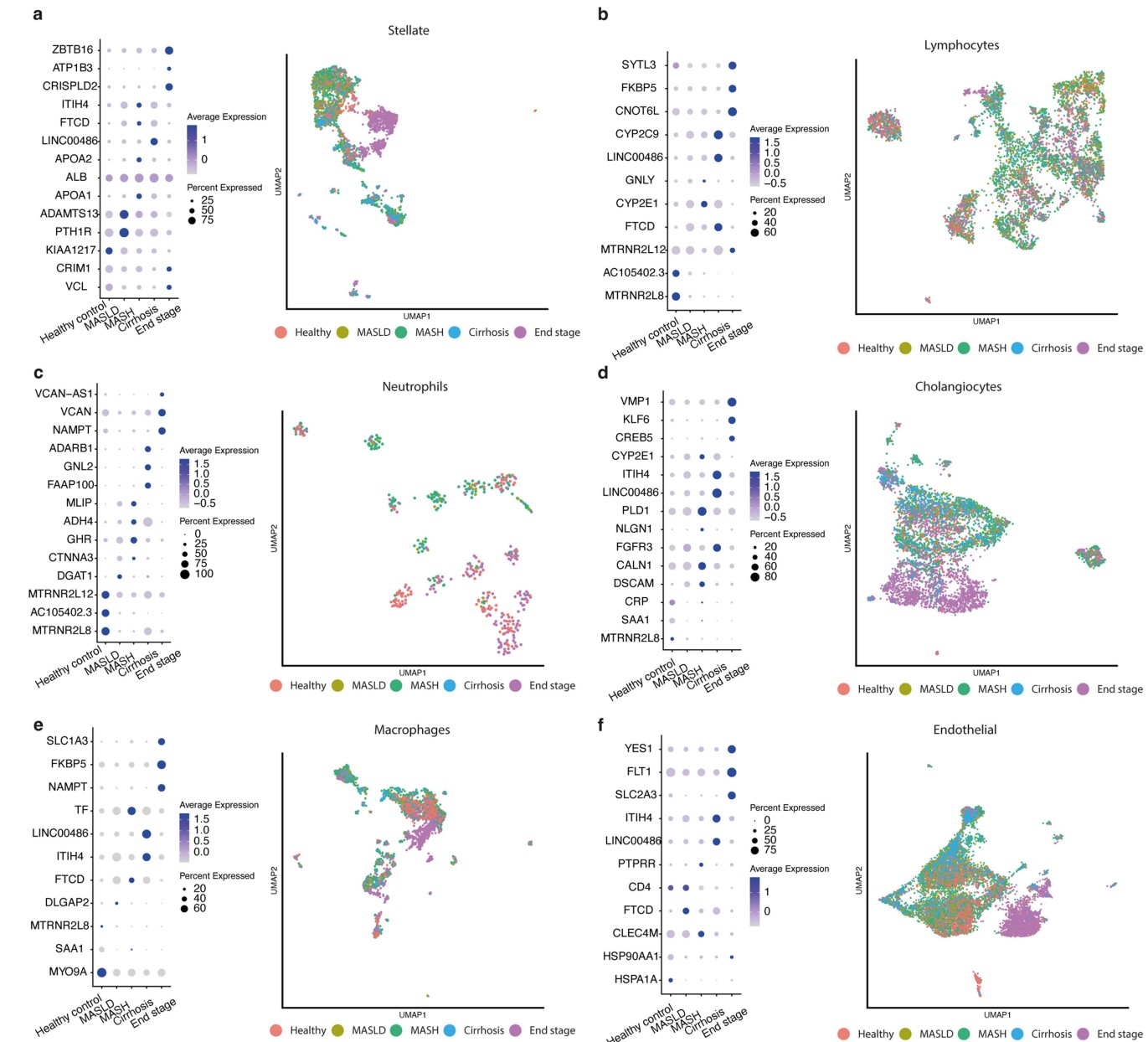

**Extended Data Fig. 4 | Hepatic cell types display different disease signature.** Bubble plot of examples of significantly differential gene expression across disease stages with corresponding UMAPs for **a**) Stellate cells **b**) Lymphocytes **c**) Neutrophils **d**) Cholangiocytes **e**) Macrophages **f**) Endothelial.

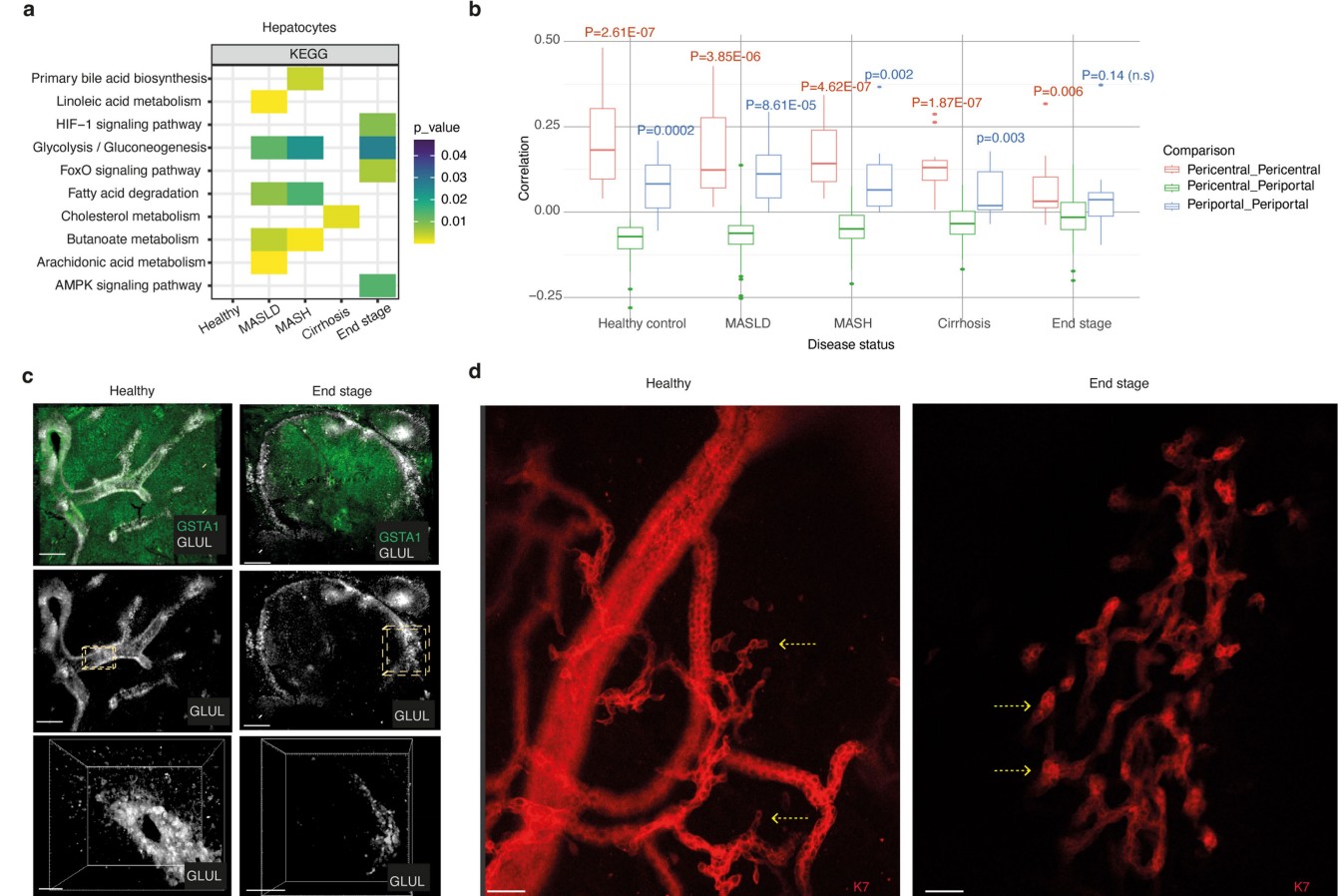

**Extended Data Fig. 5 | Hepatocytes and cholangiocytes are strongly affected by disease progression. a**) GSEA analysis of hepatocytes across disease stages. Examples of significantly enriched terms are shown for each disease stage. Benjamini-Hochberg corrected values shown. **b**) Statistical analysis corresponding to Fig. 3b. Expression-based correlations for pericentral and periportal genes, compared within groups (pericentral vs pericentral and periportal vs periportal) or between groups (pericentral vs periportal) across disease stages. P-values corresponding to comparisons of distributions for within-group and between-group correlations are indicated. Statistical significance was calculated using two-sided Welch's t-test. Per disease stage n = 66 pairwise correlations between unique pairs of genes were compared (Pericentral_Pericentral:15, Pericentral_Periportal: 36, Periportal_Periportal:15). Mid-point, minimum and maximum of the boxplot summary correspond to the

median, first and third quartiles. The extent of the whiskers correspond to the largest/smallest value no further than 1.5*IQR from the inter-quartile range. Points beyond this range are defined as outliers and are plotted individually. **c**) FLASH imaging of cleared healthy and end-stage liver tissue, with staining for pan-hepatocyte marker GSTA1 and pericentral hepatocyte marker GLUL. In healthy the high magnification (yellow box and right panel) highlights a region of the central vein with a view displayed through the lumen of the vessel. In end-stage the high magnification examines one side of a hepatocyte nodule. See also supplementary videos 1 and 2. Scale bars = 400 um low magnifications and 200 um high magnifications (right panels). **d**) FLASH imaging of highlighting ductal endings in healthy and end stage samples. Yellow arrows indicate single cell endings in healthy and bulkier endings in end stage samples. Scale bars = 50 um. n = 3 healthy and 3 end stage patient tissue samples (c-d).

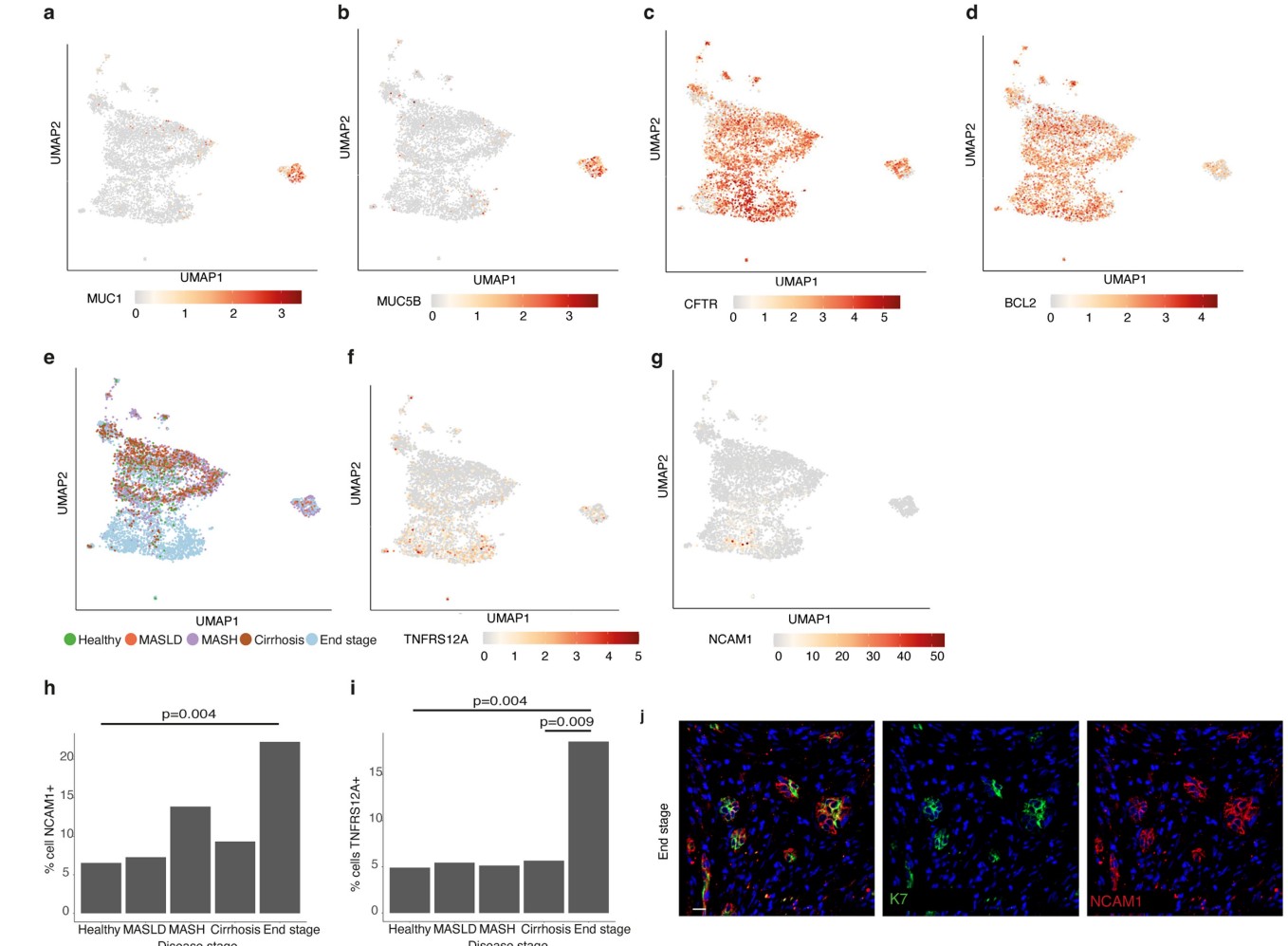

**Extended Data Fig. 6 | snRNAseq confirms cholangiocyte diversity and ductal reaction in late-stage disease. a-b**) Cholangiocyte UMAP with overlaid gradient of expression of large cholangiocyte markers *MUC5B* and *MUC1* **c-d**) Cholangiocyte UMAP with overlaid gradient of expression cholangiocyte marker *CFTR* and small cholangiocyte marker *BCL2*. **e**) UMAP indicating disease stage of cells. **f-g**) Cholangiocyte UMAP with overlaid gradient of expression of ductal reaction markers *TNFRS12A* and *NCAM1*. **h-i**) Quantification of the proportion of cholangiocytes expressing ductal reaction markers *TNFRS12A* and *NCAM1* across disease stages. p-values indicated (two-sided Fisher exact test). **j**) Immunostaining of K7 and NCAM1 in end stage tissue sections. Scale bar = 15 um.

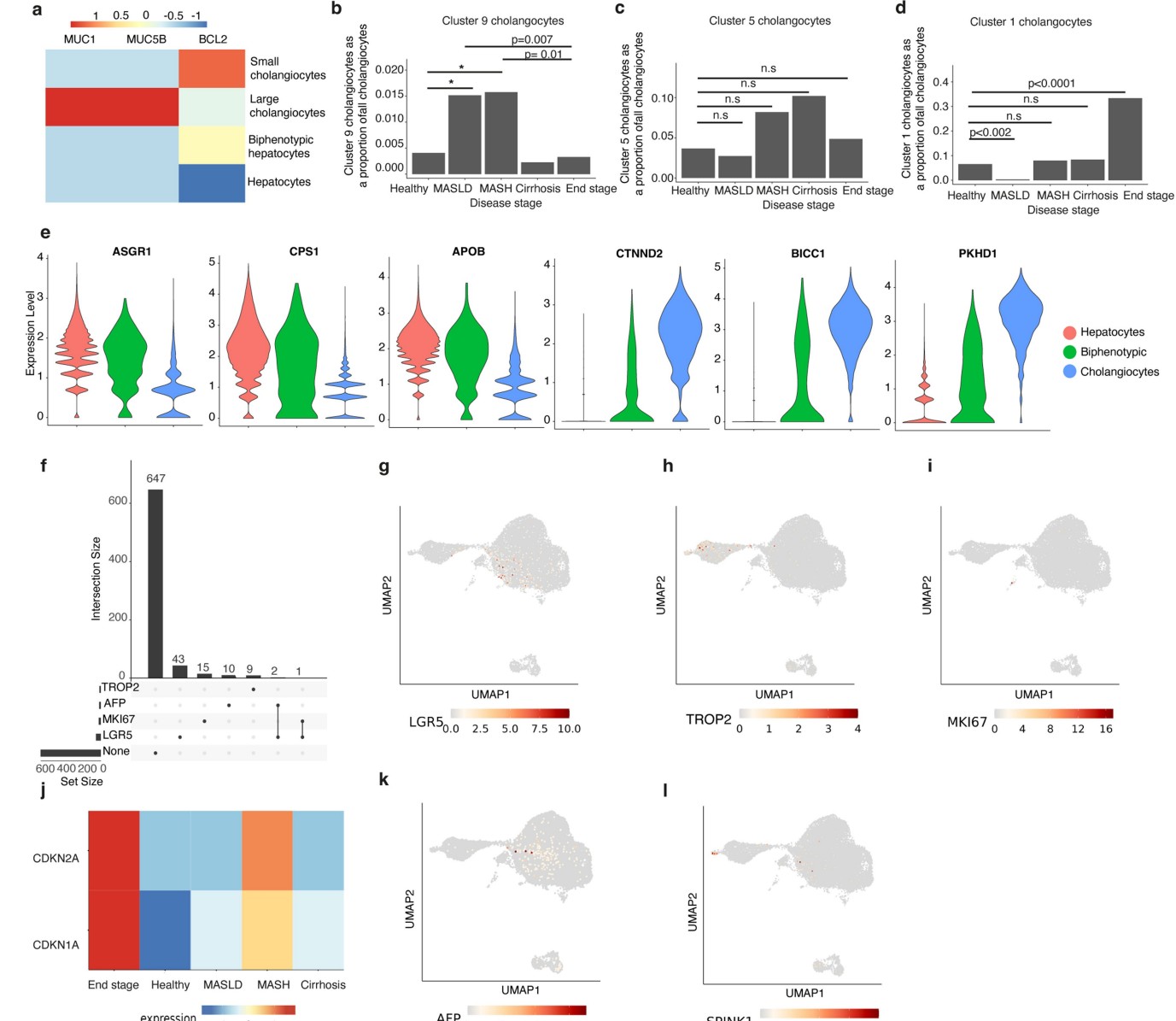

**Extended Data Fig. 7 | Characterisation of biphenotypic cells suggests an absence of an adult stem or foetal progenitor population. a**) Heatmap of relative expression of large cholangiocyte markers *MUC1* and *MUC5b* and small cholangiocyte marker *BCL2* across the indicated cell types. **b**) Cluster 9 cholangiocytes identified in Fig. 3d plotted as a proportion of cholangiocytes from each disease stage. **c**) Cluster 5 cholangiocytes identified in Fig. 3d plotted as a proportion of cholangiocytes from each disease stage. **d**) Cluster 1 cholangiocytes identified in Fig. 3d plotted as a proportion of cholangiocytes from each disease stage. P-values indicated. (Binomial Generalized Linear Mixed-Effects Model (BOBYQA optimiser, maxfun = 2e5) with patient ID

as a random effect). **e**) Violin plots of expression of indicated hepatocyte and cholangiocyte markers comparing the hepatocyte, cholangiocyte and biphenotypic populations. **f**) Upset plot displaying the number of biphenotypic hepatocytes co-expressing the indicated stem/ progenitor cell genes. **g-i**) End stage hepatocyte and cholangiocyte UMAP with overlaid gradient of expression for indicated stem cell markers. **j**) Heatmap of relative expression of senescence markers *CDKN2A* and *CDKN1A* in bi- phenotypic hepatocytes across disease progression. **k-i**) End stage hepatocyte and cholangiocyte UMAP with overlaid gradient of expression for indicated liver progenitor cell markers.

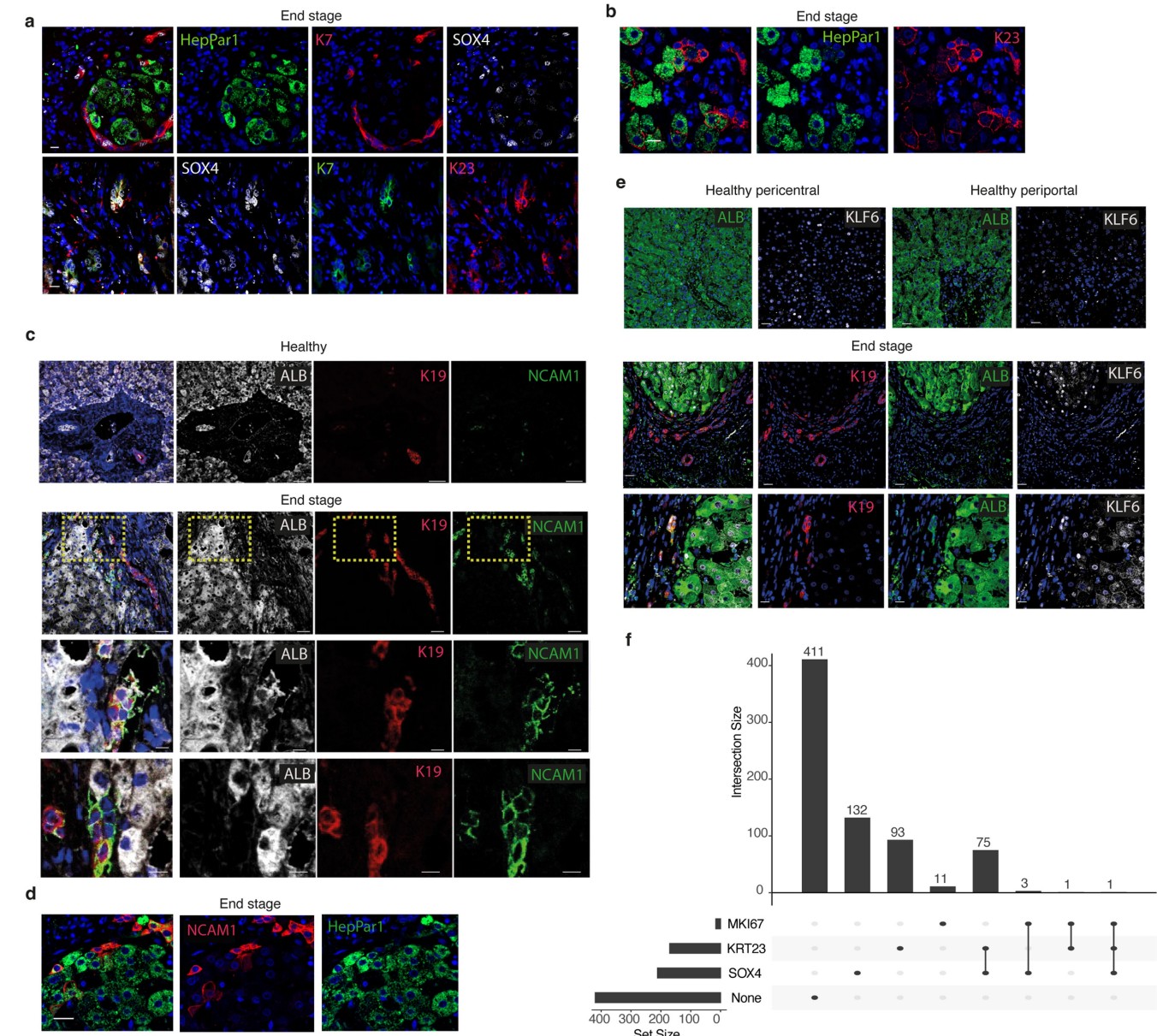

**Extended Data Fig. 8 | Plasticity markers are expressed in end stage liver.**
**a**) Immunofluorescence staining for HepPar1, K7 and SOX4 (upper panel) K23, K7 and SOX4 (lower panel) in end stage tissue sections. Scale bars = 15 um (lower panel) and 10um (upper panel). **b**) Immunofluorescence staining for HepPar1 and K23 in end stage tissue sections. Scale bar = 15 um. **c**) Immunofluorescence staining for ALB, K19 and NCAM1 in healthy and end stage tissue sections. Yellow box indicates the region shown in higher magnification. Scale

bars = 50 um upper 2 panels and 10 um lower 2 panels. **d**) Immunofluorescence staining for HepPar1 and K23 in end stage tissue sections. Scale bar = 20 um. **e**) Immunofluorescence staining for ALB, K19 and KLF6 in healthy and end stage tissue sections. Scale bars = 30 um upper 2 panels and 10 um lower panel. n = 3 healthy and 3 end stage patient tissue samples (a-e). **f**) Upset plot displaying the number of biphenotypic hepatocytes co-expressing the indicated plasticity and proliferative genes.

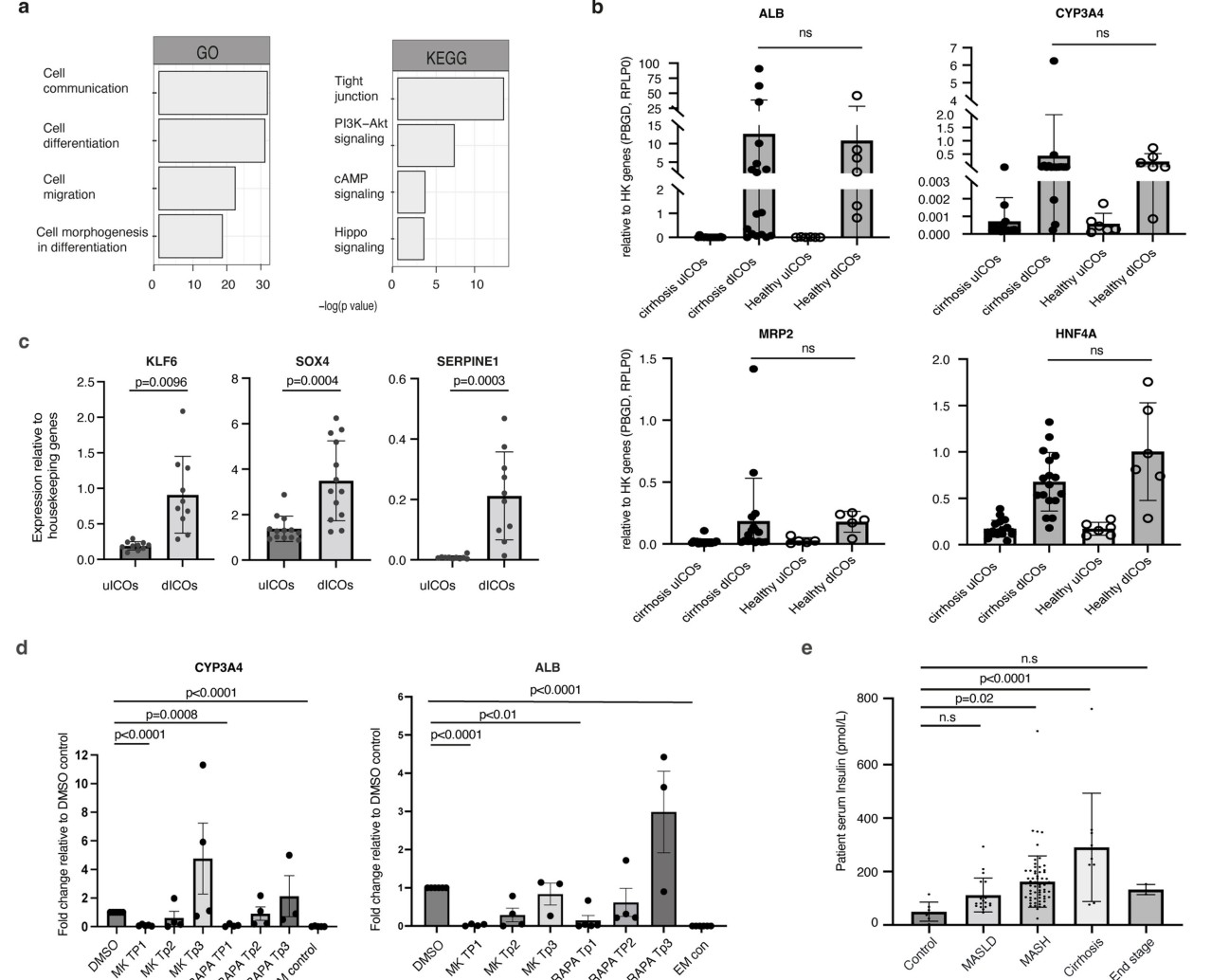

**Extended Data Fig. 9 | Intrahepatic cholangiocyte organoids (ICOs) differentiation provides a model to study cellular plasticity. a)** GSEA analysis of cholangiocyte-like-hepatocytes and hepatocyte-like-cholangiocytes from in vivo data (Fig. 3) combined. Top significantly enriched terms are shown. **b)** qPCR of hepatocyte marker expression in uICOs and dICOs derived from cirrhotic (end stage) livers or healthy donor livers. n = 17 biologically independent experiments for cirrhosis and n = 6 for healthy. Errors bars indicate mean with SD. **c)** qPCR of *KLF6*, *SOX4* and *SERPINE1*, which were identified in Fig. 3k as markers of biphenotypic cells, in uICOs and dICOs. n = 10 biologically independent experiments. P-values indicated, two-tailed unpaired t-test. Errors bars indicate SEM. **d)** qPCR for hepatocyte markers dICOs treated with either DMSO, MK2206 (AKT inhibitor) or rapamycin (mTOR inhibitor) for indicated

time. Untreated uICOs included as a control. For *CYP3A4* expression n = 6 biologically independent experiments for DMSO, 5 for MK TP1, 4 for MK TP2 and TP3, RAPA TP1 and TP2, 3 for RAPA TP3, 5 for EM control. For *ALB* expression n = 6 biologically independent experiments for DMSO, 4 for MK TP1 and TP3, 3 for MK TP3, 5 for RAPA TP1, 4 for RAPA TP2, 3 for RAPA TP3, 6 for EM control. P-values are indicated, ordinary one-way ANOVA adjusted for multiple comparisons. Error bars indicate mean with SE. **e)** Serum insulin levels of patients diagnosed from different MASLD stages. n = 7 biologically independent patients (control), 19 (MASLD), 63 (MASH), 9 (cirrhosis), 3 (end stage). P-values indicated, ordinary one-way ANOVA adjusted for multiple comparisons. Errors bars indicate mean with SD.

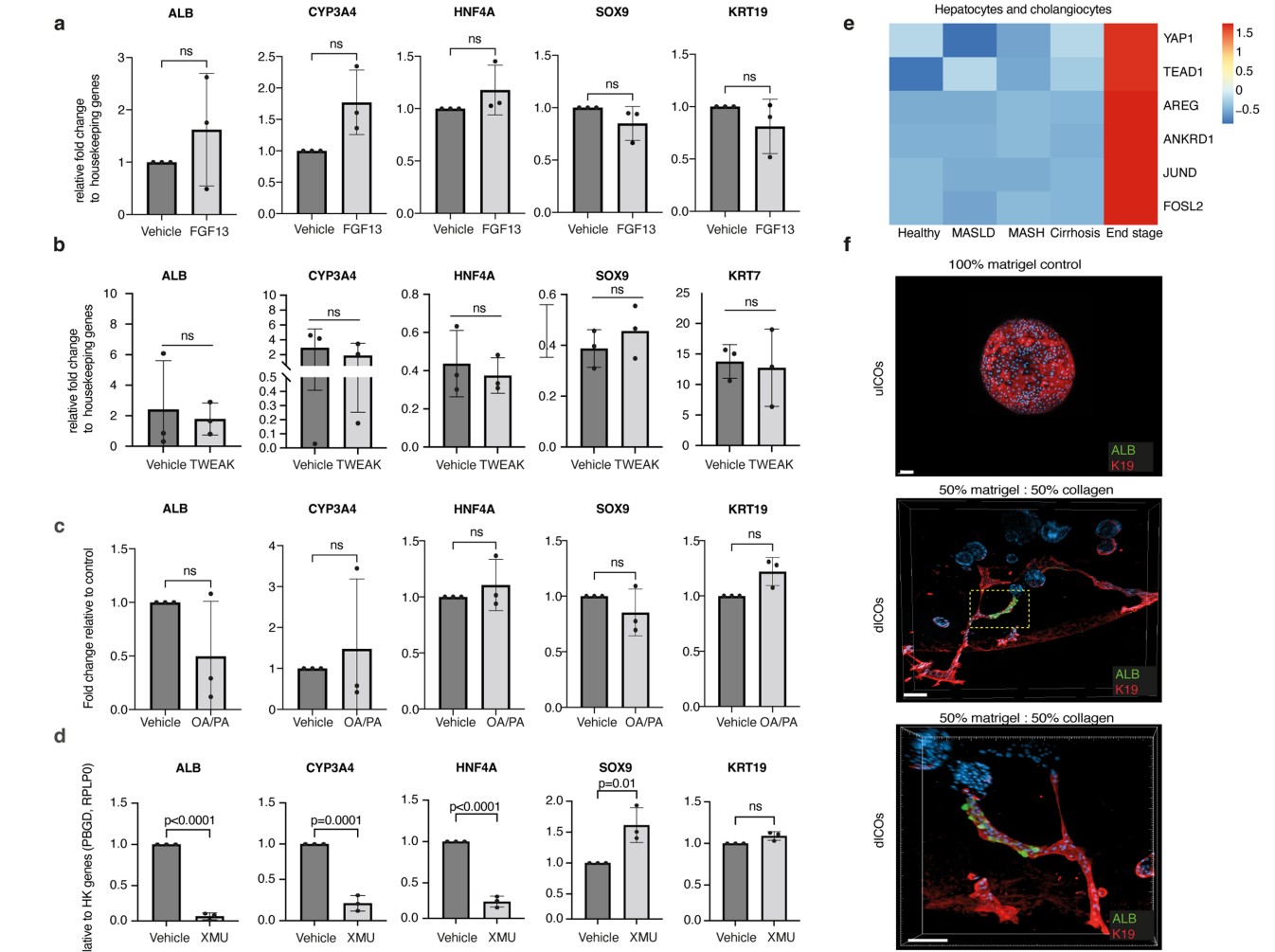

**Extended Data Fig. 10 | Extracellular matrix composition alters ICOs branching and differentiation. a-d**) qPCR for hepatocyte and cholangiocyte marker expression in dICOs treated with wither vehicle or indicated treatment for the duration of the differentiation. n = 3 biologically independent experiments, P-values are indicated unpaired t-test. Errors bars indicate mean with SD. **e**) Heatmap showing relative expression of YAP signalling genes across disease stages in hepatocytes and cholangiocytes combined. **f**) Examples of dICOs differentiated in a mixture of 50:50 collagen I: Matrigel (lower 2 panels). Immunofluorescence staining for ALB and K19. White box indicates region shown in high magnification (lower panel). uICOs grown in 100% Matrigel included as a control (upper panel). n = 3 patient organoid lines. Scale bars = 100 um upper, 200 um middle and 150 um lower panel.

# Reporting Summary

## Statistics

For all statistical analyses, confirm that the following items are present in the figure legend, table legend, main text, or Methods section.

| n/a | Confirmed | |
|---|---|---|
| ☐ | ☒ | The exact sample size (*n*) for each experimental group/condition, given as a discrete number and unit of measurement |
| ☐ | ☒ | A statement on whether measurements were taken from distinct samples or whether the same sample was measured repeatedly |
| ☐ | ☒ | The statistical test(s) used AND whether they are one- or two-sided<br>*Only common tests should be described solely by name; describe more complex techniques in the Methods section.* |
| ☒ | ☐ | A description of all covariates tested |
| ☐ | ☒ | A description of any assumptions or corrections, such as tests of normality and adjustment for multiple comparisons |
| ☒ | ☐ | A full description of the statistical parameters including central tendency (e.g. means) or other basic estimates (e.g. regression coefficient) AND variation (e.g. standard deviation) or associated estimates of uncertainty (e.g. confidence intervals) |
| ☐ | ☒ | For null hypothesis testing, the test statistic (e.g. *F*, *t*, *r*) with confidence intervals, effect sizes, degrees of freedom and *P* value noted<br>*Give P values as exact values whenever suitable.* |
| ☒ | ☐ | For Bayesian analysis, information on the choice of priors and Markov chain Monte Carlo settings |
| ☒ | ☐ | For hierarchical and complex designs, identification of the appropriate level for tests and full reporting of outcomes |
| ☒ | ☐ | Estimates of effect sizes (e.g. Cohen's *d*, Pearson's *r*), indicating how they were calculated |

*Our web collection on statistics for biologists contains articles on many of the points above.*

## Software and code

Policy information about availability of computer code

| Data collection | Zen 2011 SP7 on Zeiss 710, 880 or 980 confocal microscope. QuantStudio 5 384 well block (Thermo Fisher). |
|---|---|
| Data analysis | For snRNAseq, pre-processing of raw count matrices was performed using Seurat (v4.0.3) [doi:10.1016/j.cell.2021.04.048]. Gene expression values were normalised for library size using sctransform version 0.3 (doi: 10.1186/s13059-019-1874-1). The clustering parameters used were identified by evaluating the resulting cluster stability using ClustAssess doi:(10.1101/2022.01.31.478592). Data was integrated using Harmony version 0.1.1. doi: 10.1038/s41592-019-0619-0). Velocyto (v0.17.17), and velocyto.R (v0.6) were used to estimate RNA velocity based on prevalence of spliced and unspliced mRNA [doi: 10.1038/s41586-018-0414-6]. Gene set enrichment analysis (GSEA) was done using gprofiler2 (version 0.2.0). For snRNAseq Scripts for all bioinformatics analyses carried out are made available at https://github.com/Core-Bioinformatics/NAFLD-NASH.<br>Prism 9 was used for creating graphs and statistical analyses. Images were analysed using Imaris 9.7.1. |

For manuscripts utilizing custom algorithms or software that are central to the research but not yet described in published literature, software must be made available to editors and reviewers. We strongly encourage code deposition in a community repository (e.g. GitHub). See the Nature Portfolio guidelines for submitting code & software for further information.

## Data

Policy information about availability of data

All manuscripts must include a data availability statement. This statement should provide the following information, where applicable:

- Accession codes, unique identifiers, or web links for publicly available datasets
- A description of any restrictions on data availability
- For clinical datasets or third party data, please ensure that the statement adheres to our policy

The sequencing data and raw expression matrix is available on the Gene Expression Omnibus (GEO) series entry GSE202379.
R Shiny apps illustrating the analysis can be found at https://bioinf.stemcells.cam.ac.uk/shiny/vallier/LiverPlasticity_GribbenGalanakis2023/

## Human research participants

Policy information about studies involving human research participants and Sex and Gender in Research.

| | |
|---|---|
| Reporting on sex and gender | 30 males and 17 females were included in the dataset. Sex and age of patients are included Extended Data Table 1-2. |
| Population characteristics | Samples from 47 patients were included in the snRNAseq data and organoids were derived from 10 patients. Population information is provided in extended tables 1-3. |
| Recruitment | Patients were referred to Addenbrookes Hospital liver unit for liver biopsy due to suspected non-alcoholic fatty liver disease. Screening criteria: 18 years old or above, suspected NAFLD on referral, alcohol intake of less than 14 units / week. |
| Ethics oversight | Biopsy collection and processing of human samples was carried out under ethics approved by Addenbrookes hospital REC 18/WM/0397. The study met all criteria for responsible use of human tissue that is used in the UK. All patients were offered the patient information sheet and provided informed consent. Healthy deceased transplant organ donor tissue and explants were taken under ethics approved by NRES Committee East of England - Cambridge South (REC number REC 15/EE/152). All patients provided informed consent. |

Note that full information on the approval of the study protocol must also be provided in the manuscript.

# Field-specific reporting

Please select the one below that is the best fit for your research. If you are not sure, read the appropriate sections before making your selection.

☒ Life sciences          ☐ Behavioural & social sciences          ☐ Ecological, evolutionary & environmental sciences

For a reference copy of the document with all sections, see nature.com/documents/nr-reporting-summary-flat.pdf

# Life sciences study design

All studies must disclose on these points even when the disclosure is negative.

| | |
|---|---|
| Sample size | No statistical analysis were performed to predetermine sample size. For snRNAseq all biopsies which were collected and successfully processed for snRNAseq based on bioinformatics QC were included in the dataset. All were included to allow for representation of all disease stages. For organoid experiments at least an n of 3 (where n is a different organoid line) were used. |
| Data exclusions | No data was excluded |
| Replication | For organoid experiments at least an n of 3 (where n is a different organoid line) were used. Details of n are given in figure legends. |
| Randomization | Patients were categorized by disease stage based on histology. For organoid work, cells were plated in different wells of a 24 well plate and were randomly allocated to experimental groups (control and treatments). |
| Blinding | Blinding was not possible for sample processing due to the logistics of patient sample collection and tissue processing. |

# Reporting for specific materials, systems and methods

We require information from authors about some types of materials, experimental systems and methods used in many studies. Here, indicate whether each material, system or method listed is relevant to your study. If you are not sure if a list item applies to your research, read the appropriate section before selecting a response.

## Materials & experimental systems

| n/a | Involved in the study |
|---|---|
| ☐ | ☒ Antibodies |
| ☐ | ☒ Eukaryotic cell lines |
| ☒ | ☐ Palaeontology and archaeology |
| ☒ | ☐ Animals and other organisms |
| ☒ | ☐ Clinical data |
| ☒ | ☐ Dual use research of concern |

## Methods

| n/a | Involved in the study |
|---|---|
| ☒ | ☐ ChIP-seq |
| ☒ | ☐ Flow cytometry |
| ☒ | ☐ MRI-based neuroimaging |

# Antibodies

| Antibodies used | Albumin (Bethyl A80-229A)<br>K19 (abcam ab7754)<br>K7 (abcam ab68459)<br>MRP2 (abcam ab3373)<br>GLUL (abcam ab125724)<br>ASS1 (Cambridge Bioscience HPA020896)<br>SOX4 (ab86809)<br>K23 (Cambridge Bioscience HPA016959)<br>GSTA1  (abcam ab53940)<br>HepPar1 (Agilent Dako clone 0CH1E5)<br>KLF6 (Sigma HPA069585)<br>NCAM1 (Sigma HPA039835)<br>Alexa fluor 488 donkey anti-goat (Invitrogen A11055)<br>Alexa fluor 568 donkey anti-rabbit (Invitrogen A10042)<br>Alexa fluor 647 donkey anti-mouse (Invitrogen A31571).<br>FlexAble CoraLite® Plus 647 Antibody Labeling Kit for Rabbit IgG<br>FlexAble CoraLite® Plus 647 Antibody Labeling Kit for Mouse IgG1<br>FlexAble CoraLite® Plus 555 Antibody Labeling Kit for Rabbit IgG<br>FlexAble CoraLite® Plus 555 Antibody Labeling Kit for Mouse IgG1<br>FlexAble CoraLite® Plus 488 Antibody Labeling Kit for Mouse IgG1<br>FlexAble CoraLite® Plus 488 Antibody Labeling Kit for Rabbit IgG |
|---|---|
| Validation | Albumin (Bethyl A80-229A) https://www.thermofisher.com/antibody/product/Human-Albumin-Antibody-Polyclonal/A80-229A<br>K19 (abcam ab7754) https://www.abcam.com/cytokeratin-19-antibody-a53-ba2-cytoskeleton-marker-ab7754.html<br>K7 (abcam ab68459) https://www.abcam.com/cytokeratin-7-antibody-epr1619y-cytoskeleton-marker-ab68459.html<br>MRP2 (abcam ab3373) https://www.abcam.com/mrp2-antibody-m2-iii-6-ab3373.html<br>GLUL (abcam ab125724) https://www.abcam.com/glutamine-synthetase-antibody-6glutamine-synthetase-ab125724.html<br>ASS1 (Cambridge Bioscience HPA020896) https://www.bioscience.co.uk/product~683967. Validated as part of the Human Protein Atlas.<br>SOX4 (ab86809) https://www.abcam.com/sox4-antibody-ab86809.html.<br>K23 (Cambridge Bioscience HPA016959) https://www.bioscience.co.uk/product~682837. Validated as part of the Human Protein Atlas.<br>GSTA1  (abcam ab53940) https://www.abcam.com/gsta1-antibody-ab53940.html<br>HepPar1 (Agilent Dako clone 0CH1E5) https://www.agilent.com/en/product/immunohistochemistry/antibodies-controls/primary-antibodies/hepatocyte-%28dako-omnis%29-76237<br>KLF6 (Sigma HPA069585) https://www.sigmaaldrich.com/GB/en/product/sigma/hpa069585<br>NCAM1 (Sigma HPA039835) https://www.sigmaaldrich.com/GB/en/product/sigma/hpa039835<br>Alexa fluor 488 donkey anti-goat (Invitrogen A11055) https://www.thermofisher.com/antibody/product/Donkey-anti-Goat-IgG-H-L-Cross-Adsorbed-Secondary-Antibody-Polyclonal/A-11055<br>Alexa fluor 568 donkey anti-rabbit (Invitrogen A10042) https://www.thermofisher.com/antibody/product/Donkey-anti-Rabbit-IgG-H-L-Highly-Cross-Adsorbed-Secondary-Antibody-Polyclonal/A10042<br>Alexa fluor 647 donkey anti-mouse (Invitrogen A31571). https://www.thermofisher.com/antibody/product/Donkey-anti-Mouse-IgG-H-L-Highly-Cross-Adsorbed-Secondary-Antibody-Polyclonal/A-31571<br>https://www.ptglab.com/products/FlexAble-CoraLite-Plus-647-Antibody-Labeling-Kit-for-Rabbit-IgG-KFA003.htm#:~:text=Product%20Information-,FlexAble%20CoraLite®%20Plus%20647%20Antibody%20Labeling%20Kit%20for%20Rabbit,primary%20antibodies%20from%20any%20supplier.<br>https://www.ptglab.com/products/FlexAble-CoraLite-Plus-647-Antibody-Labeling-Kit-for-Mouse-IgG1-KFA023.htm#:~:text=Product%20Information-,FlexAble%20CoraLite®%20Plus%20647%20Antibody%20Labeling%20Kit%20for%20Mouse,primary%20antibodies%20from%20any%20supplier.<br>https://www.ptglab.com/products/FlexAble-CoraLite-Plus-555-Antibody-Labeling-Kit-for-Rabbit-IgG-KFA002.htm<br>https://www.fishersci.co.uk/shop/products/flexable-coralite-plus-550-antibody-labeling-kit-mouse-igg1-3/p-7227370<br>https://www.ptglab.com/products/FlexAble-CoraLite-488-Antibody-Labeling-Kit-for-Rabbit-IgG-KFA001.htm<br>https://www.ptglab.com/products/FlexAble-CoraLite-488-Antibody-Labeling-Kit-for-Rabbit-IgG-KFA001.htm |

# Eukaryotic cell lines

Policy information about cell lines and Sex and Gender in Research

Cell line source(s)

12 organoid lines were derived in this study (from 8 males and 4 females). Age and sex are listed in extended data table 4

Authentication

Organoid lines were derived from samples collected straight from clinic.

Mycoplasma contamination

All lines tested negative for mycoplasma contamination.

Commonly misidentified lines
(See ICLAC register)

None were used in this study

nature portfolio | reporting summary

March 2021

