## [Peer Review File · Nature]

Manuscript Title: Acquisition of epithelial plasticity in human chronic liver disease

Reviewer Comments & Author Rebuttals

Reviewer Reports on the Initial Version:

Referees' comments:

The article by Gribben & Galanakis et al entitled 'Acquisition of epithelial plasticity in the human liver during chronic disease progression' describes the plasticity and differentiation of cells from biliary lineage towards hepatocytes in NAFLD by means of single nucleus sequencing. The study includes a large data of 47 liver samples. Validation was done by means of immunohistochemistry and immunofluorescence. Functionally, the authors explored the role of the PI3K-AKT-mTOR axis in cell fate by using human organoids.

The authors do not really specify how they define a cholangiocyte. The phenotype of a cholangiocyte differs enormously depending on the anatomical location: going from large mature mucus producing cells to small interconnecting cells located at the canals of Hering. Upon epithelial damage, the latter can proliferate, observed as ductular reaction. Ductular reaction is regarded as activation liver progenitor cells, which can lead differentiation into hepatocytes or mature cholangiocytes. The clustering of cholangiocytes should follow the consensus nomenclature on the branches of the biliary tree (PMID: 15185318 and PMID: 21983984). If one wants to look at plasticity, one needs to properly identify ductular reaction and intermediate lineage. This is currently lacking and makes some of the conclusions questionable.

Keratin 7 is also a marker for intermediate hepatocytes. One cannot claim that 'In addition these experiments revealed cells co-expressing ALB and KRT7, suggesting the presence of cells combining hepatocyte and cholangiocyte phenotypes'. This is fundamentally incorrect. Intermediate hepatocytes are negative for K19 or TROP2 but still show positivity for EPCAM or K7. As a consequence, the bi-phenotypic cells within the clusters mentioned in the manuscript are not well specified (e.g. first remark). How do the authors explain Krt19-positive hepatocytes (Fig3B) or Krt19-negative cholangiocytes (Fig3D)? On protein level, K19 is used to indicate cells of biliary lineage.

Gene expression will not be the same as protein level. This should be taken into account when defining a cell type.

Figure 2 'Major changes in hepatocyte zonation and biliary tree remodelling in end stage NAFLD' is not new. For example: PMID: 14507639, PMID: 21983984 and PMID: 24254368. This should be referenced properly and put into perspective. A lot of the findings in this manuscript refer to the presence of intermediate hepatocytes + ductular reaction, which is known to occur in advanced liver disease. How do cell phenotypic gene signatures of ductular reaction change in the different stages of NAFLD? How does plasticity change during chronic NAFLD? This has not really been answered. The

main focus is on end-stage NAFLD.

Extended data tables are not included. Difficult to assess which markers were used to identify the cell phenotypic markers. How do you define a cholangiocyte or hepatocyte? Ductular reaction, intermediate hepatocytes?

- For example, CYP3A4 expression has been reported to go down with disease progression. How does that influence the clustering of hepatocytes?
- CFTR is mainly expressed in mature mucus producing cholangiocytes? What about the rest of the biliary tree? Only based on KRT7? So intermediate hepatocytes as well?
- Fig1E MARCO and CD163 are Kupffer cell markers. What about the monocyte-derived macrophages? Where do those cluster?

Standard IHC on all samples would be beneficial. In what extent do you see ductular reaction and intermediate cells in the early stages? It is very likely that there is an association with disease activity. Is there a difference between at-risk NASH patients and end-stage liver? How does that translate to the observed gene profiles?

Is the resolution of snRNAsequencing high enough to identify progenitor cells in early stages? These will be a small minority of the total cells <1%.

The effect of genotype and T2DM should be explored.

Figure 1. Albumin gives background staining. Sinusoidal lining cells and portal endothelial cells show positivity. These findings are not really reliable and the Ab is not validated. I would suggest to use a better validated Ab and do a dye swap.

Central scoring is lacking.

Explant tissue will behave differently. A thorough comparison between explant and needle biopsy should be included. The histological stainings indicate substantial damage to the bile ducts, something that is not typical for NAFLD. Have the immunofluorescent stainings mainly been done on explant specimens?

There is a high % of NAFL with advanced fibrosis. Are these missed NASH samples? Central reading with IHC for identify ballooned hepatocytes might answer this question.

Healthy controls are not healthy. Rejected donor livers because of obstructions or alcohol use cannot be considered as healthy. At best, you can refer to them as non-NAFLD controls. Same goes for the screening biopsy. There is a reason why a patient gets a diagnostic biopsy (e.g. abnormal ALT-AST levels). These samples should be assessed by an expert liver pathologist.

Ref 45 should be put better in perspective as the article published similar findings on the IHC.

Signalling mechanisms are lacking. What triggers the activation and differentiation? How does this change in throughout the NAFLD spectrum?

Title is somewhat misleading. The authors only focus on NAFLD where hepatocellular damage occurs. The other spectrum has not been taken into account (PSC, PBC).

Minor

GEO data not accessible.

Extended data table 1 is not mentioned in the text and is missing.

NAFLD refers to the spectrum. NAFL (nonalcoholic fatty liver) is used to annotate the early stage.

Annotations: KRT is used for genes, proteins are annotated with K or CK. Genes should be annotated in Italic.

High number of reviews in the references. Relevance of Ref4 not clear.

Limitations of snRNAsequencing should be discussed and tested.

Referee #2 (Remarks to the Author):

The present manuscript by Gribben et al., addresses an important question in liver biology. How does the organ cope with prolonged injury in the course of NAFLD development. Using imaging and single-cell approaches, the authors analyse patient biopsy material on a trajectory from healthy over NAFLD to end stage diseases. Using these data, the authors identify some interesting new insights into the progression of NAFLD disease, such as the loss of zonation (marker expression) and the identification of transdifferentiation events in the ductal region. Particularly the resource nature of the data is important and might serve as a basis for future more functional and hypothesis-driven work. Overall, the study is well laid out and the manuscript is easy to follow. I only have very few reservations that should be addressed before the study is ready for publication.

The observation of an increase in bi-phenotypic cells with disease progression is quite intriguing. As this is one of the central and novel points in the manuscript, this part is still too underdeveloped and needs further confirmation in form of experiments and analysis as well as a much deeper discussion.

1. The authors should show how QC was performed to exclude doublets and potentially free-floating RNA as well, which is a particular problem in single-nucleus sequencing.

2. Based on pseudo time analysis, the authors chose two marker genes (Sox4 and Krt23) to identify bi-phenotypic cells in tissue. Both Sox4 and Krt23 are expressed in cells of ductal origin and it would be essential to add another bi-phenotypic gene to this analysis (such as FKBP5) to show co-expression in given cells. In case immunofluorescence does not work, FISH-based approaches would present a very good alternative.

3. A recent paper (31350390) identified a population of hepatobiliary hybrid progenitor of ductal origin that at least shares some markers with the bi-phenotypic cells identified here. What are the difference/similarities between these populations? The other dataset is based on SMART-seq2, so more sensitive, but a comparison might still give interesting insights. At least, this paper has to be discussed.

Finally, the authors used cholangiocyte-derived organoids from NAFLD end stage patients to address a potential role of the PI3K-AKT signalling pathway, which was enriched in GSEA pathway analysis in

the bi-phenotypic cell population. Although the organoids showed similarities to this population and might thus be a good ex vivo system to study the emergence of these cells, the current experimental setup is not conclusive as organoids from healthy donors need to be cultured and compared in parallel. The authors even point out that organoid differentiation can be very heterogeneous. Thus, addition of control organoids is an essential addition to the manuscript.

Organoids differentiated in the presence of PI3K-AKT pathway inhibitors did not show hepatic marker expression. How was differentiation efficiency assessed in this case? Could the inhibitors simply block or halt differentiation? This has to be ruled out. ^[1]_{SEP}

Minor points

- it would be good to add a plot/table to show representation of cell types / patient

- how were samples treated for snRNA-seq? was each patient one lane on the 10x Chromium (as Extended Data 3 suggests) or were samples pooled? Please clarify in Results and M&M for better clarity.

Referee #3 (Remarks to the Author):

Review for “Acquisition of epithelial plasticity in the human liver during chronic disease progression”

The authors address the important question of tissue regeneration in the context of non-alcoholic fatty liver disease (NAFLD) progression in humans. There are potentially three scenarios for regeneration: stem cell activation, de/re-differentiation, and transdifferentiation. Combining single-cell transcriptome analysis with imaging revealed reorganized zonation profiles and considerable changes in the biliary tree. Moreover, the results found indicate that transdifferentiation between hepatocytes and cholangiocytes, without activation of stem or progenitor cells, is at the heart of the plasticity underlying regenerative capacity. While overall the findings are potentially of great interest, some analyses leading to key hypotheses need to be solidified as some of the claims may not be sufficiently robust at this stage.

Major:

1. Cell types. Fig. 1E: it seems surprising that only 25% of cholangiocytes express KRT7. Why is Albumin not shown as a hepatocyte marker?

2. “We observed the existence of cells “bridging” hepatocyte and cholangiocyte clusters and co-expressing specific markers for both cell types (Extended data Figure 3E).” The argument about the biphenotypic cells seems crucial but it was not very convincing with the short description and ED Fig. 3E. It seems necessary to supplement it with more statistical analysis, and importantly provide a clearer definition of those cells. For example in ED Fig. 3E, it appears that ABCC2 and KRT7 are also expressed in the other (non hepatocyte and non cholangiocyte) clusters.

3. Loss of zonation, Figure 2B. The visual discussion of correlation matrices to show the loss of zonation should be supplemented with a statistical argument. Moreover, perturbed zonation in NAFLD is established in mouse models, and in humans it was shown at the level of the lipidome (DOI: 10.1002/hep.28953) proteogenomics (DOI: 10.1016/j.cell.2021.12.018).

4. Figure 3A-D. What makes hepatocyte cluster 9 be part of hepatocytes? For example, it appears that cluster 9, but also many other clusters, shows very low Albumin expression. These arguments in favor of the possible dual origin of bi-phenotypic cells needs to be made more convincing.

5. Are those biphenotypic cells related to hepatoblasts?

6. Fig. 3E. How are the p-values calculated? Please show all the proportions for the individual samples, for example using violin plots.

7. In a recent paper (DOI: 10.1126/scitranslmed.add3949), the Friedman lab also reported snRNA-seq of patients with NASH, albeit fewer in numbers and covering less disease stages. It would be of interest to assess whether biphenotypic cells are also found in those data.

8. Lines 166-184: Arguments based on co-detection of two genes are dangerous in scRNA-seq or snRNA-seq due to the low detection rate. At least this need to be assessed in comparison to a proper null model. Most genes will in fact not be co-detected even if they are both present in a cell. Moreover, the arguments made from snRNA-seq should be taken with care since the effective lifetime of nuclear RNAs is very short. Therefore absence of nuclear transcript may not mean absence of cytoplasmic transcript or even protein. Thus the suggestion that plasticity increases with time needs to be strengthened. It is also risky to base such an argument on few marker genes. A systematic multi-gene analysis would be more convincing.

9. RNA velocity is not expected to work with snRNA-seq due to the short life time of nuclear transcripts. Thus it should be demonstrated that it works here (which is not completely impossible but unlikely), for example by showing that intron and exon signals are shifted in function of pseudotime.

10. PI3K-AKT signaling in biphenotypic cells and ICOs. Is it possible that the activity of mTOR in ICOs is reflecting the high demand on protein synthesis in this system? It is unclear what the blot in Fig. 8C is showing. Is beta actin meant as control? Show the quantifications and replicas.

Minor:

1. Fig 1F: Caption was cut.

2. "Importantly, QCs were performed to confirm that these cells were not due to doublets or RNA contamination."

Please provide more details about this.

3. Fig 4A may be significantly reduced in size or removed.

Typos, etc:

33 - no comma after here

34 - "from across"

41 - comma before thereby

41 - where does insulin signaling come from all of a sudden?

52 - comma

57 - comma before (ii)

86 - biopsies in plural

95-100 - hepatocytes highest changes, what about stellate cells that actually produce fibrotic tissue?!

114 - hypoxia is not a pathway

119 - genes italics

150-152 - I don't believe you can define origin by clustering analysis or markers!

234 - do not capitalize Rapamycin

238 - no space after comma

figure legends - spaces around mathematical symbols and units

Referee #4 (Remarks to the Author):

The manuscript from Gribben, Galanakis and colleagues focuses on the progression of human liver disease to identify potential changes in cellular plasticity associated to liver regeneration and disease. The experimental methodology is robust and state-of-the art, including single-nuclei sequencing of human liver tissues at different stages of chronic liver disease, and human liver organoids. By using these experimental approaches, the authors detected a biphenotypic cell population expressing both hepatocytes and cholangiocyte markers at end stages of liver disease. This is suggestive of cellular plasticity of liver hepatocytes and cholangiocytes occurring in chronic liver disease. More in detail, the authors hypothesise that hepatocyte-into-cholangiocyte and cholangiocyte-into-hepatocyte trans-differentiation occurs in chronic liver disease. Cellular plasticity of hepatocytes and cholangiocytes (i.e. the capacity to both hepatocytes and cholangiocytes to give rise to each other) has been extensively detected in mouse and associated to regeneration of the liver epithelial compartment after prolonged liver injury (Yanger et al., *Genes&Dev*, 2013; Font-Burgada et al., *Cell*, 2015; Raven et al., *Nature*, 2017; Russell et al., *Hepatology*, 2018; Deng et al., *Cell Stem Cell*, 2018; Manco et al., *J. Hepatol*, 2019). Importantly a biphenotypic cell population has been previously identified in human liver disease (Yanger et al., *Genes&Dev*, 2013, PMID: 23520387). This particular reference should be added to the manuscript as it is highly relevant.

Altogether, the findings of this manuscript enable to robustly confirm the presence of a human biphenotypic cell population; at the same time evidence in literature questions the novelty of this manuscript since this biphenotypic cell population had been previously detected in the liver. In addition, only a limited characterisation of this population has been performed in this manuscript. Specifically, i) the role of this biphenotypic cell population in liver disease remains unclear; ii) data provided in this manuscript regarding a role of PI3K/AKT/mTOR in the specification/maintenance of biphenotypic cells, are inconclusive. Regarding the latter, the authors show that inhibition of

PI3K/AKT/mTOR impairs the expression of hepatocyte markers in differentiated human liver organoids. However, it remains unclear if PI3K/AKT/mTOR inhibition in organoids blocks the specification of hepatocyte cell identity or the generation of biphenotypic cells. In addition, levels of insulin in the serum of patients with chronic liver disease peaks at cirrhosis, whereas the authors observe biphenotypic cells mainly at the end stage of chronic liver disease, thus suggesting that additional or different signals drive cellular plasticity and the specification of liver biphenotypic cells.

In summary, the manuscript adopts a robust experimental approach to confirm the presence of a biphenotypic cell population in human liver; however, as it is, this manuscript shows limited novelty since this biphenotypic cell population had been previously identified in both mouse and human livers. Further characterisation of this biphenotypic cell population should be performed to determine the role of this cell population in the liver response to damage/regeneration and the mechanisms leading to their specification.

Major points:

1) Experimental methods:

- All immunostaining experiments throughout the manuscript should be quantified and n= cells analysed and n= biological/technical experimental replicates should be indicated. Statistical analyses should be applied and reported in the Figure Legends;
- The overwhelming majority of the tissues have been obtained by white British individuals; this should be clearly stated in the text and captured as a diversity statement.

2) The authors have confirmed the presence of ALB+ KRT7+ biphenotypic cells previously detected in human livers, and have found them prominently at the end stage of chronic liver disease. However, the role of these cells in liver disease remains unclear. The authors should perform experimental approaches aimed at determining the role of these cells. Since chronic liver disease can degenerate into cancer, one possibility would be to investigate the presence of biphenotypic cells in different types of human liver cancer and perform analyses to determine whether they are associated to cancer drivers and carcinogenic processes.

3) The authors have identified different clusters associated to biphenotypic cells. This seems particularly relevant in cholangiocytes since the analyses suggest that biphenotypic cells in some cholangiocyte clusters may originate at earlier stages of liver disease. Together this suggests that the biphenotypic cells form a heterogeneous cell population. The authors should determine if biphenotypic sub-populations have different features, including expression of stem-cell genes, proliferation potential and regenerative and carcinogenic capacity in vivo and/or in vitro using organoid systems.

4) The authors have found no evidence of a progenitor signature associated to biphenotypic cells by checking co-expression with stem-cell genes such as LGR5 and TROP2. Thus, they conclude that in human chronic liver disease trans-differentiation (hepatocyte-to-cholangiocyte and cholangiocyte-to-hepatocyte) occurs rather than de-differentiation of hepatocytes/cholangiocytes into bipotent progenitors capable of giving rise to both cell types. However, these analyses were conducted at the end stage of liver disease. To determine if liver progenitors are detectable and could give rise to the

biphenotypic cells detected at the end stage of chronic liver disease, the author should check if expression of LGR5 and TROP2 overlaps with subsets of makers of different clusters of biphenotypic cells at earlier stages of chronic liver disease.

5) Line 202: Proliferation of SOX4+ and KRT23+ is described as number of cells expressing mKi67 but not shown as a Figure.

6) The authors show significant biliary tree remodelling in chronic liver disease. Does this impact the generation of biphenotypic cells? To address this, the authors could take advantage of organoids grown with physical constrains resembling the remodelling of the biliary tree observed in the tissue.

7) The authors should perform experimental approaches aimed at determining a role of PI3K/AKT/mTOR in the specification of biphenotypic cells in chronic liver disease. Based on the results shown in organoids it remains unclear if PI3K/AKT/mTOR inhibition blocks the specification of hepatocyte cell identity or the generation of biphenotypic cells. In this Reviewer's opinion, experimental approaches in mouse models would be appropriate to determine the signalling determining the specification of biphenotypic cells.

Author Rebuttals to Initial Comments:

Point by point answer: “Acquisition of epithelial plasticity in the human liver during chronic disease progression”. Nature 2022-11-17484B

Gribben C. and Galanakis V et al.,

General answer:

We would like to thank the reviewers for their supportive and helpful comments. Reviewer 1 found our study interesting while asking for more information regarding the single cell analyses. Reviewer 2 believes that our manuscript “addresses an important question in liver biology” while our “study is well laid out and the manuscript is easy to follow”. Reviewer 3 finds our “findings potentially of great interest” and “ have only very few reservations before the study is ready for publication”. Reviewer 4 believes our “experimental methodology is robust and state-of-the art”. Based on these encouraging remarks, we have now addressed all the reviewers’ comment and we are grateful for their input. We believe the resulting data has strengthened our original conclusions while expanding the scope of our manuscript.

Answers to Referee 1 comments:

The article by Gribben & Galanakis et al entitled ‘Acquisition of epithelial plasticity in the human liver during chronic disease progression’ describes the plasticity and differentiation of cells from biliary lineage towards hepatocytes in NAFLD by means of single nucleus sequencing. The study includes a large data of 47 liver samples. Validation was done by means of immunohistochemistry and immunofluorescence. Functionally, the authors explored the role of the PI3K-AKT-mTOR axis in cell fate by using human organoids.

Comment 1: *The authors do not really specify how they define a cholangiocyte. The phenotype of a cholangiocyte differs enormously depending on the anatomical location: going from large mature mucus producing cells to small interconnecting cells located at the canals of Hering. Upon epithelial damage, the latter can proliferate, observed as ductular reaction. Ductular reaction is regarded as activation liver progenitor cells, which can lead differentiation into hepatocytes or mature cholangiocytes. The clustering of cholangiocytes should follow the consensus nomenclature on the branches of the biliary tree (PMID: 15185318 Nomenclature of the finer branches of the biliary tree: Canals, ductules, and ductular reactions in human livers and PMID: 21983984 Ductular reactions in human liver: diversity at the interface). If one wants to look at plasticity, one needs to properly identify ductular reaction and intermediate lineage. This is currently lacking and makes some of the conclusions questionable.*

First, we would like to reassure the reviewers that our analyses and the data generated have been extensively reviewed by collaborators with world-wide recognised expertise in liver diseases (Dr Michael Allison, Head of the NASH clinic, Addenbrooke’s hospital, Dr Sue Davies, Histopathologist, HPB subspecialty, Addenbrooke’s hospital, and Frank Tacke, Head of the hepatology and

gastroenterology department, Charite Berlin). Thus, we are confident that our study follows standards in the field and relies on up-to-date clinical evaluations. Moreover, we respectfully ask the reviewer to consider the importance of single cell transcriptomics in our conclusions. Most immunostaining were used to validate the conclusions of these analyses and thus histopathology results should not be considered in isolation.

To answer more precisely to the reviewer, the revised manuscript includes additional data showing the markers used to identify cholangiocytes in our single cell analyses. We used 10 markers which are commonly applied to identify cholangiocytes independently of their location in the biliary tree (**Extended Data Fig. 3C**). This combinatorial approach allows the robust identification of all the cholangiocytes captured by our single cell sampling.

We also feel the need to underline that our analyses do not show that ductal reaction (DR) and regenerative processes are linked. We simply observed that the biliary tree undergoes a major reorganisation during disease progression while cholangiocytes become more plastic in end stage livers. Moreover, we don't claim that the association between NAFLD/NASH progression and DR is novel. However, we believe that our innovative 3D imaging technology shows for the first time the importance of the reorganisation of the biliary system associated with this process. This aspect has been clarified in the new version of the manuscript.

More importantly, we agree that the cellular diversity in the biliary epithelium could be important and that this aspect of cholangiocyte biology can be very complex. While the organisation of the intrahepatic biliary tree is well described histologically (canal of Hering, intralobular and intraportal bile ductules, and terminal bile ducts), the transcriptomic profile of the corresponding cells is more challenging to define due to the lack of well-established markers for these different locations. In other words, there is only a limited correspondence between the transcriptomic profile of cholangiocytes, their location in the tree and their morphology. We and other have described this challenge in previous single cell studies showing that cellular diversity in the biliary epithelium is much more limited than initially suggested (Brevini et al., Nature 2022, Sampaziotis et al., Science 2021, Aizarani et al. 2019 Nature, Andrews et al., 2022 Hepatology communications)

Nonetheless, our single cell analyses reveal two main population of cholangiocytes as indicated by a clear separation in the UMAP space. One population shows high expression of mucins such as MUC1 and MUC5B, and the other population shows higher expression of genes such as BCL2 (**Extended data Fig. 6a-d**). The first population is likely composed of mucus producing cells located in the larger ducts while the second population is likely located in smaller ducts. Of note, defining the cells of the canals of Hering is difficult as they are not well described transcriptionally. Indeed, previous publications have suggested that NCAM1 and TWEAK receptor Fn14 (*TNFRSF12A*) can not only mark these cells but also ductal reaction structures (Gadd et al, 2014). Accordingly, our single cell analyse show that these genes are mainly expressed in end stage cholangiocytes which belong to the “smaller duct population” (**Extended Data Fig. 6e-i**). To further characterise the location of NCAM1 positive cells, we performed immunostaining of end stage NAFLD/NASH liver tissue sections and observed NCAM1 positive cells in structure related to ductular reaction region (**Extended Data Fig. 8a**). Finally, we observe that biphenotypic cells are more related at the transcriptomic level to cholangiocytes expressing

NCAM1/BCL2 (Extended Data Fig. 7a) thereby suggesting that cholangiocytes located in the small ducts are more likely to transdifferentiate into hepatocytes.

Considered together, these new analyses show that we are sampling cholangiocytes of small and larger ducts, and ductal reaction/ canal of Hering cells while confirming that cholangiocyte location in the biliary tree could play a role in plasticity acquisition. The manuscript has been modified to include these new conclusions.

Comment 2a. *Keratin 7 is also a marker for intermediate hepatocytes. One cannot claim that 'In addition these experiments revealed cells co-expressing ALB and KRT7, suggesting the presence of cells combining hepatocyte and cholangiocyte phenotypes'. This is fundamentally incorrect. Intermediate hepatocytes are negative for K19 or TROP2 but still show positivity for EPCAM or K7. As a consequence, the bi-phenotypic cells within the clusters mentioned in the manuscript are not well specified (e.g. first remark).*

We thank the reviewer for highlighting intermediate hepatocytes and appropriate references are now included in the results section referring to Figure 1 and in the discussion. However, we would like to refer to our answer to Reviewer 3 comment 2 concerning the identification of biphenotypic cells using single cell analyses. As indicated in our answer, the biphenotypic cells identified by our analyses transdifferentiating cells co-expressing high levels of several markers for hepatocytes and cholangiocytes. Moreover, these cells are only detected in end stage livers. Thus, they differ by definition from intermediate hepatocytes.

Importantly, we do observe cells expressing K7/ ALB by immunostaining (Fig. 1b). Thus, we agree that some of these K7/ALB cells could be intermediate hepatocytes and we have modified the manuscript accordingly. However, these immunostaining analyses must be considered in the context of our single cell analyses (see answer to Reviewer 3 comment 2). Indeed, these transcriptomic analyses are much more precise and efficient to identify cell type. Accordingly, our biphenotypic cells are defined by the co-expression of multiple markers specific for cholangiocytes and hepatocytes while they establish a clear transdifferentiation process (Fig. 3b and d) These biphenotypic cells are not hepatocyte expressing K7 only. These cells are hepatocytes acquiring a biliary identity or cholangiocytes acquiring an hepatocytes identity. Of note, we validate most of our analyses using K19/ALB double positive cells which are not intermediate hepatocytes based on the reference provided by the reviewer and we could not detect our biphenotypic cells before end stage disease while intermediate hepatocytes appear at early stage (Roskams et al., 2003). Finally, we have sought clarification regarding the nature of these cells histologically. Sections exhibiting biphenotypic cells have now been analysed by consultant histopathologist Dr Sue Davies (consultant histopathologist, liver subspecialty, Addenbrookes Hospital) who confirmed that biphenotypic cells identified in our data look different from intermediate cells. We believe that intermediate cells are hepatocytes expressing abnormal level of KRT7 while biphenotypic cells described in our study are truly transdifferentiating cells. Such population can only be characterised by transcriptomic analyses.

To conclude, the biphenotypic cells identified in our validations on tissue sections may include some intermediate hepatocytes which have been described histologically before, but our single cells transcriptomic analyses and our conclusions definitely rely on different cells.

Comment 2b: *How do the authors explain Krt19-positive hepatocytes (Fig3B) or Krt19-negative cholangiocytes (Fig3D)? On protein level, K19 is used to indicate cells of biliary lineage.*

Gene expression shown in Figure 3B and 3D are relative expression, so this is indicating higher or lower expression, not necessarily absence of expression. Figure 3B shows the KRT19+ hepatocytes which also express additional cholangiocyte markers. These cells are the biphenotypic cells which are the main focus of our study and which are definitely different from intermediate cells.

Comment 2c: *Gene expression will not be the same as protein level. This should be taken into account when defining a cell type.*

We agree that expression at transcript and protein level can vary. However, the method used to annotate cell type bypassed these limitations. Indeed, a cell type are initially defined by the expression of 10 markers and then by their entire transcriptome (See answer comment 1). Furthermore, our conclusions were systematically validated using immunostaining. So this aspect does not change our findings.

Comment 3. *Figure 2 'Major changes in hepatocyte zonation and biliary tree remodelling in end stage NAFLD' is not new. For example: PMID: 14507639 Oxidative Stress and Oval Cell Accumulation in Mice and Humans with Alcoholic and Nonalcoholic Fatty Liver Disease , PMID: 21983984 Ductular reactions in human liver: diversity at the interface and PMID: 24254368 The portal inflammatory infiltrate and ductular reaction in human nonalcoholic fatty liver disease. This should be referenced properly and put into perspective. A lot of the findings in this manuscript refer to the presence of intermediate hepatocytes + ductular reaction, which is known to occur in advanced liver disease.*

We agree with the reviewer that that abnormal zonation and ductular reaction in NAFLD/NASH have been show previously and we have now included the corresponding publications. However, our study is the first one to characterise the progressive change of zonation during disease progression at the transcriptional level. More importantly, our data show that disease affect the capacity of hepatocyte to zonate properly since they start to co-express markers of different zone which should not be impossible. Concerning the ductular reaction, we believe that our 3D imaging and the corresponding movies show for the first time the extent of biliary tree remodelling in end stage disease. We have shown these data to numerous clinicians who were all surprised not only by the extent of the process but also by the way that bile ducts surround hepatocyte nodules. This may explain how ductular reaction could be an aggravating cause. Finally, our answer to comments 2a and b explains that the biphenotypic cells identified in our study are different from the intermediate hepatocytes referred by the reviewer. Overall, the novelty of our manuscript lies in the demonstration that transdifferentiation is the most likely a disease process in chronic liver injury in patient and this acquisition of plasticity is in fact not a repair mechanism but a sign of disease progression (See Answer to Reviewer 4 comment 1)

Comment 4. *How do cell phenotypic gene signatures of ductular reaction change in the different stages of NAFLD? How does plasticity change during chronic NAFLD? This has not really been answered. The main focus is on end-stage NAFLD.*

Ductular reaction is characterised by morphological changes and by the increase in the number of cholangiocytes. However, markers for this process are relatively rare with the exception of NCAM1 and TNFRSF12A (Gouw et al, 2011). Accordingly, we observed an increase in the number of cholangiocytes expressing these markers with disease progression (**Extended Data Fig. 6h-i**). However, we did not observe the appearance of new type of cholangiocytes suggesting that DR is not associated with a fundamental change in cholangiocyte gene signature.

Concerning disease progression, our transcriptomic analyses clearly show an increase of cholangiocyte-like-hepatocytes (biphenotypic cells) in end stage livers. Thus, transdifferentiation process seems to occur mainly in end stage disease suggesting that acquisition of plasticity could occur progressively during disease progression.

Comment 5. *Extended data tables are not included. Difficult to assess which markers were used to identify the cell phenotypic markers. How do you define a cholangiocyte or hepatocyte? Ductular reaction, intermediate hepatocytes?*

We apologise for the absence of this table and for any confusion caused regarding cell type annotation. We have now added an supplementary data table (**Supplementary data table 3**) and a heatmap showing expression of various markers used for annotation across cell types (**Extended Data Fig. 3c**). Markers shown in Fig. 1e are just two examples of markers used per cell type to indicate the identity of the different clusters in a simplified figure. The revised version of the manuscript also refers to ductular reaction and intermediate hepatocytes but as discussed above our data are not focusing on this cell type.

Comment 6. *For example, CYP3A4 expression has been reported to go down with disease progression. How does that influence the clustering of hepatocytes?*

Our analyses confirm that *CYP3A4* expression changes during disease progression, while being zonated in control liver. However, this marker does not disappear (i.e. hepatocytes continue to express this marker even at low level). Furthermore, the clustering of the hepatocytes is performed in an unsupervised manner using the whole transcriptome and so change in one marker is unlikely to drive a change in clustering.

Comment 7. *CFTR is mainly expressed in mature mucus producing cholangiocytes? What about the rest of the biliary tree? Only based on KRT7? So intermediate hepatocytes as well?*

CFTR expression in cholangiocytes is shown in **Extended Data Fig. 6d**. These analyses show that CFTR transcripts are detected across most cholangiocytes suggesting that this marker is less specific than initially suggested by immunostaining analyses. This is not entirely surprising as CFTR protein is notoriously difficult to detect. Numerous other cholangiocyte markers were used to defined cholangiocytes (**Extended Data Fig. 3c**) and see answer to comment 1. *CFTR* and *KRT7* were just 2 examples of markers shown in Fig 1e.

Comment 8. *Fig1E MARCO and CD163 are Kupffer cell markers. What about the monocyte-derived macrophages? Where do those cluster?*

The UMAP representation in Figure 1 including all the cell type lacks the resolution to detect subpopulation for each cell type. To address this limitation, we have subclustered the macrophages and analysed the expression of specific markers for Kupffer and monocyte derived macrophages. As indicated by the reviewer, we can observe both populations based on the expression of specific markers

thereby confirming the quality of our data set (data shown below).

Comment 9. *Standard IHC on all samples would be beneficial. In what extend do you see ductular reaction and intermediate cells in the early stages? It is very likely that there is an association with disease activity. Is there a difference between at-risk NASH patients and end-stage liver? How does that translate to the observed gene profiles?*

The revised version of the manuscript includes additional K19 immunohistochemistry on biopsies from patients at different stage of the disease. These analyses show that ductal reaction starts in cirrhosis (NASH F4) and largely increase in end stage disease as suggested by our single cell analyses (see answer to comment 1). In this new figure, we highlight some examples of interesting events including cells with hepatocyte morphology which are K19 positive (**Extended Data Fig. 1d**). As discussed in comment 1, we observed the highest level of cells displaying ductular reaction markers in end stage disease (point 1 and 2) thereby confirming an association between this process and disease progression. Taken together, these data suggest that ductular reactions is progressive and culminate in patients with end stage disease.

Comment 10. *Is the resolution of snRNA sequencing high enough to identify progenitor cells in early stages? These will be a small minority of the total cells <1%.*

In our data we captured the following numbers of hepatocytes and cholangiocytes:

Disease stage	Healthy	NAFLD	NASH	NASH-C	End stage
Hepatocytes	3750	7073	31903	3603	23097
Cholangiocytes	245	330	1966	441	2430

As we have captured high numbers of cells it should be possible to resolve rare populations even if below 1% (around 40 cells). We also analysed the expression of genes marking potential progenitor and stem cells populations (**Extended Data Fig. 7**) and we did not find a positive population either co-expressing these markers and/or related to the production of bi-phenotypic cells. Thus, we can confidently exclude the involvement of stem cells/ progenitors in regenerative processes occurring in cirrhotic livers.

Comment 11. *The effect of genotype and T2DM should be explored.*

[Figure redacted]

We have analysed the frequency of 6 SNPs commonly associated with NAFLD in our patients cohort (**data shown below**). These SNPs are represented across all disease stages. Notably, some of these frequencies are much higher than the population carrier frequency, confirming previous studies describing the importance of these variants in disease progression. However, the relatively small number of patients included in our study limits the statistical power necessary to corroborate phenotype with genetic variants. So, it would be impossible to link plasticity acquisition with a specific genotype. Similarly, the proportion of patients with T2D increase with disease which

is to be expected and which also reinforce our hypothesis regarding the role of insulin signalling in plasticity acquisition (**Data shown below**).

[Text Redacted]

Comment 12. Figure 1. Albumin gives background staining. Sinusoidal lining cells and portal endothelial cells show positivity. These findings are not really reliable, and the Alb is not validated. I would suggest to use a better validated Ab and do a dye swap.

The albumin antibody used in our study has routinely been employed by others on tissue sections and on cells grown in vitro (Segal et al, 2019, Wesley et al, 2022). In addition, we have now performed immunostaining using an antibody against HepPar1 (Dako Clone OCH1E5) which is another marker commonly used to identify hepatocytes in human. We observe specific staining on hepatocytes confirm the presence of HepPar1/ Krt7 double positive cells in end stage livers (**Extended data Fig. 1c**).

Comment 13. Central scoring is lacking.

Patient ID	SAF score from Histopathologist 1	SAF score from Histopathologist 2
3	S2A3F1	S2A3F1
6	S1A1F1	S1A2F0
7	S3A4F2	S3A3F3
8	S2A4F3	S2A4F3
9	S2A2F1	S1A2F1
11	S2A4F3	S2A3F3
12	S2A3F4	S2A3F4
15	S2A4F3	S2A4F4
16	S1A4F3	S1A4F3
19	S1A4F3	S1A4F1
20	S2A3F3	S1A2F3
21	S2A3F2	S2A3F3
22	S1A3F4	S1A3F3
30	no nafld / no SAF score	no nafld / no SAF score
48	S1A2F2	S1A2F2
49	S1A3F2	S1A4F1
50	S3A3F2	S2A3F2
51	S1A1F1	S1A2F2
52	S2A2F1	S3A2F0
53	S2A4F3	S1A4F2
54	S3A3F2	S3A4F2
55	S1A1F2	S1A2F2
56	S1A1F1	S1A0F0
57	S2A3F2	S2A2F3
60	S1A1F0	S1A1F0
62	S2A3F3	S2A4F3
64	S3A3F1	S3A2F1
67	S2A2F3	S2A4F3
68	S1A3F4	S1A3F4
70	S2A3F3	S2A3F3
71	S2A4F2	S1A2F1
72	S2A3F3	S2A4F3
73	S2A3F1	S2A2F1
75	S1A3F4	S2A4F3
76	S1A3F2	S1A3F2
77	S1A3F2	S1A4F2
78	S1A1F0	S1A1F0
83	S2A4F2	S2A4F2
84	S1A0F3	S1A2F3
98	no nafld / no SAF score	S0A4F3 (not nafld)
cl103	end stage NAFLD	S0A2F4
cl104	end stage NAFLD	S0A2F4
cl16	end stage NAFLD	S0A1F4

Scoring of NAFLD/NASH vary between clinical centres and variation can be observed between intermediate stage. Thus, two consultants histopathologist (liver sub-speciality) have now reviewed all the biopsies included in our study using an alternative scoring system. In sum, we did not see major variation between the different scoring (see Table included here). More importantly this variation has no impact on our conclusions since they only concern intermediate stage which no influence on the bi-phenotypic cells identified by our transcriptomics analyses.

Comment 14. *Explant tissue will behave differently. A thorough comparison between explant and needle biopsy should be included. The histological staining indicate substantial damage to the bile ducts, something that is not typical for NAFLD. Have the immunofluorescent staining mainly been done on explant specimens?*

Immunostaining has been performed on sections from all disease stages (**Extended Data Fig. 1**). Sampling of end stage liver and healthy liver was optimised to closely resemble the procedure used with biopsies. All the tissues were collected and snap frozen or fixed immediately after surgery. Furthermore, we have included multiple imaging data either in 2D or 3D FLASH imaging (**Fig 2d and Extended Data Fig. 5d**) showing that the biliary tree is well preserved in healthy liver and while being massively different in end stage, the biliary structures are still intact and cells visible to a single cell resolution (**Extended Data Fig. 5d and Supplementary Videos 3 and 4**). Thus, we are confident our tissue preservation approaches are robust. Regarding the snRNAseq, we have used strict QCs to exclude dying, doublet and stressed cells from our analyses bioinformatically and during sample processing. Also, all analyses were performed using post-filtered data to ensure only high quality cells are analysed. Thus, comparisons performed in our study between these disease stages cannot be affected by the method of collection. Finally, we have now confirmed our results using an alternative data set thereby confirming that the existence of biphenotypic cells and the transdifferentiation process is not related to tissue processing (See answer to Reviewer 3 comment 7).

Comment 15. *There is a high % of NAFL with advanced fibrosis. Are these missed NASH samples? Central reading with IHC for identify ballooned hepatocytes might answer this question.*

See answer comment 13. Patient tissue was analysed and IHC were performed for each biopsy as standard practice for NAFLD/NASH diagnosis by histopathologists. However, these analyses are still subject to interobserver variability. Accordingly, each biopsy was scored twice by histopathologist experts and we did observe limited change in scores concerning intermediate stages as suggested by the reviewer. This is a well-known challenge which is due to the difficulties of interpreting histological slides. However, these changes were limited and did not affect our results which concerns variations

between end/ cirrhotic stages vs intermediate stages. Thus, this variability does not affect the conclusions of our study.

Comment 16. *Healthy controls are not healthy. Rejected donor livers because of obstructions or alcohol use cannot be considered as healthy. At best, you can refer to them as non-NAFLD controls. Same goes for the screening biopsy. There is a reason why a patient gets a diagnostic biopsy (e.g. abnormal ALT-AST levels). These samples should be assessed by an expert liver pathologist.*

All samples have been assessed by an expert liver pathologist including healthy samples. We only used liver with minimal steatosis and low LFTs (See below). Thus, we consider that these livers were functionally healthy and that the best samples that we could obtain the context of human pathophysiology. Importantly, we have provided this additional information about these controls liver in Supplementary table 2

Patient ID	Bilirubin (0 – 20 umol/L)	ALP (30 – 130 U/L)	Alanine Transaminase (10 – 49 U/L)	Aspartate Transaminase (<=34 U/L)
30	17	65	46	33
98	16	45	58	34
HL1	5	91	28	n/a
HL2	11	84	11	n/a

Comment 17. *Ref 45 should be put better in perspective as the article published similar findings on the IHC (Keratin 23 is a stress-inducible marker of mouse and human ductular reaction in liver disease)*

We have now added additional text to the discussion citing this manuscript since it reinforces our observations showing that K23 expression increases with fibrosis. However, this report did not examine samples of end stage NAFLD and mainly relies on mRNA and protein lysates. More importantly, they did not identify K23 as a marker of plasticity and transdifferentiation.

Comment 18. *Signalling mechanisms are lacking. What triggers the activation and differentiation? How does this change in throughout the NAFLD spectrum?*

The revised version of the manuscript includes new data providing additional information about the signalling pathways involved in transdifferentiation. We previously identified the PI3K-AKT-mTOR pathway as one regulator of the differentiation based on GSEA analyses on biphenotypic cells (**Extended Data Fig. 9a**). We also showed that inhibiting this pathway blocks differentiation in vitro (**Fig. 4c and d**). To further investigate the role of this pathway, we have now performed gain of function experiments and observed that activation of the PI3K-AKT-mTOR signalling increased differentiation of intrahepatic cholangiocyte organoids (ICOs) into cells expressing hepatocytes markers (**Fig. 4e and**

f and Extended Data Fig 8d). We also observed the generation of ALB+ cells in ICOs when treated with mTOR activator even in standard cholangiocyte media (**Fig. 4e and f**). This striking affect support the importance of this pathway. Furthermore, the revised manuscript also include data showing that inhibition of PI3K-AKT-mTOR signalling blocks the differentiation of cholangiocytes into biphenotypic cells without being necessary for their maintenance (See answer to reviewer 2 comment 6). Considered together, these results confirm that the PI3K-AKT-mTOR signalling is necessary for the transdifferentiation process and not for the survival of biphenotypic cells.

In addition, we hypothesised that plasticity acquisition and the subsequent transdifferentiation is driven by combination of different pathways over a prolong period of time. Thus, we decided to identify additional mechanism and other pathway potentially involved in this process. We first focus on FGF13 since this growth factor is significantly induce during transdifferentiation (**Fig. 3k**). Interestingly, addition of FGF13 did induce a limited increase in expression of hepatocyte markers during ICO differentiation (**Extended Data Fig. 10a**). Thus, FGF13 could play a role in transdifferentiation. We also treated cells with the pro-inflammatory cytokine TWEAK and fatty acids, both of which play a key role in NALFD progression. However, none of these factors appeared to increase transdifferentiation (**Extended Data Fig. 10b and c**), confirming that mechanisms involved in plasticity acquisition might occur after the original injury (i.e steatosis) and the pro-inflammatory phase. Finally, we also observed that components of the YAP/TAZ pathways were strongly up regulated during disease progression (**Extended Data Fig. 10e**). Interestingly, genetic studies in the mouse have shown that this pathway could be important for ductular reaction in vivo (Planas-Paz et al., 2019, Pepe-Mooney et al., 2019). Thus, we performed activation of this pathway using small molecule and observed a total inhibition of transdifferentiation (**Extended Data Fig. 10d**). Thus, activation of YAP/TAZ could promote ductular reaction while protecting cellular identity.

Considered together, these additional data confirm that a complex combination of several signalling pathways are driving the acquisition of plasticity. The PI3K-AKT-mTOR pathway plays a central role in the transdifferentiation processes and we propose that the increase of this signalling through augmentation of insulin during disease progression is major drive for the development of regenerative process in end stage human livers.

Comment 19. *Title is somewhat misleading. The authors only focus on NAFLD where hepatocellular damage occurs. The other spectrum has not been taken into account (PSC, PBC).*

We would prefer to keep the current title. PSCs and PBC affect the biliary tree while NAFLD is the most common chronic liver disease. Furthermore, we are convinced that our conclusions can be extended to other liver disease involving chronic injury of hepatocytes. Finally, the abstract quickly clarifies the focus of our study and we feel the title fits better for the broad appeal of the journal. Nonetheless, we will follow the recommendation of the editor on this point.

Minor

Comment 20. *GEO data not accessible*

We apologise for this; the correct link and token is provided below:
To review GEO accession GSE202379:

Go to:

<https://eur03.safelinks.protection.outlook.com/?url=https%3A%2F%2Fwww.ncbi.nlm.nih.gov%2Fgeo%2Fquery%2Facc.cgi%3Facc%3DGSE202379&data=05%7C01%7Cecw63%40universityofcambridgecloud.onmicrosoft.com%7Ca78fa38642dc4acc989908db890c9d1c%7C49a50445bdfa4b79ade3547b4f3986e9%7C1%7C0%7C638254459493241207%7Cunknown%7CTWFpbGZsb3d8eyJWlloiMC4wLjAwMDAiLCJQIjoiV2luMzliLCJBTiI6IklhaWwiLCJXVCI6Mn0%3D%7C3000%7C%7C%7C&sdata=y1HrbhByxvAQGfsTaAqMy6aGpwG91DOPpXRBGJ183ig%3D&reserved=0>

Enter token yponkeygvfmdtdwd into the box

Comment 21. *Extended data table 1 is not mentioned in the text and is missing.*

The table 1 is now included in the revised version of the manuscript (now as Supplementary table 3).

Comment 22. *NAFLD refers to the spectrum. NAFL (nonalcoholic fatty liver) is used to annotate the early stage.*

We will follow the nomenclature recently recommended by EASL once the manuscript is accepted for publication.

Comment 23. *Annotations: KRT is used for genes, proteins are annotated with K or CK. Genes should be annotated in Italic.*

The revised manuscript has been modified accordingly.

Comment 24. *High number of reviews in the references. Relevance of Ref4 (Identification of stem cells in small intestine and colon by marker gene Lgr5) not clear.*

The revised version of the manuscript has been modified to include more references of primary publications. The reference 4 highlights the intestine as an example of organs containing adult stem cells as part of the introduction.

Comment 25. *Limitations of snRNA sequencing should be discussed and tested.*

We have now added some discussion on this in our revised manuscript. Overall, we believe the snRNAseq approach is best suited for our experiments due to its increased efficiency at capturing hepatocytes and cholangiocytes compared to scRNAseq studies. Additionally, snRNAseq allows for the processing of frozen biopsies, which was of great advantage to our study logistically, ensuring high quality samples and consistency in processing. Importantly, our conclusions are based on a diversity of validation including immunostaining on tissue sections, 3D imaging and in vitro functional experiments. This approach compensates for the usual drawbacks associated with transcriptomic analyses such as mRNA vs Proteins and absence of functional validations.

Answers to Referee 2 comments:

The present manuscript by Gribben et al., addresses an important question in liver biology. How does the organ cope with prolonged injury in the course of NAFLD development. Using imaging and single-cell approaches, the authors analyse patient biopsy material on a trajectory from healthy over NAFLD to end stage diseases. Using these data, the authors identify some interesting new insights into the progression of NAFLD disease, such as the loss of zonation (marker expression) and the identification of transdifferentiation events in the ductal region. Particularly the resource nature of the data is important and might serve as a basis for future more functional and hypothesis-driven work. Overall, the study is well laid out and the manuscript is easy to follow. I only have very few reservations that should be addressed before the study is ready for publication.

The observation of an increase in bi-phenotypic cells with disease progression is quite intriguing. As this is one of the central and novel points in the manuscript, this part is still too underdeveloped and needs further confirmation in form of experiments and analysis as well as a much deeper discussion.

Comment 1. *The authors should show how QC was performed to exclude doublets and potentially free-floating RNA as well, which is a particular problem in single-nucleus sequencing.*

Doublets were removed both during the processing of the sample for snRNAseq and bioinformatically. Indeed, we performed nuclei sorting by flow cytometry before loading on the 10X machine. Our gating strategy and sorting settings were precise to remove debris and to only sort single nuclei. Additionally nuclei isolation was optimised during this project; QC steps were performed to ensure efficient sample lysis pre-sort and nuclei were examined post-sort to ensure a high quality single nuclei suspension (single nuclei with nuclear membrane intact with no blebbing) was loaded into the 10x. Additional details of this have now been added to the updated methods. Post-sequencing doublets were removed

by filtering using nCount and nFeature. Violin plots of the post-filtering nCount and nFeature distributions are shown in Extended data Figure 2. Of note, biphenotypic cells were mainly detected in end-stage livers and thus their existence can't originate from a technical problem (otherwise, they should be present in all the samples). Finally, we did perform validations using immunostaining showing that biphenotypic cells can be detected in vivo and they express markers identified by our in silico analyses such as K23 and SOX4. Thus, we are confident that our cell annotation is accurate.

However, we agree that the “soup” remains a major source of false positive expression in single cell analyses especially in liver tissues where some gene are expressed at a very high level. However, we optimised our snRNAseq protocol to decrease free-floating RNA using nuclei sorting and subsequent wash steps. Furthermore, cell type annotation including for the biphenotypic cells was performed using multiple genes expressed at relatively low levels and thus which are not particularly affected by free-floating RNAs. We avoid highly expressed genes such as *ALB* or *A1AT (SERPINA1)* (**Extended Data Fig. 3c**).

***Comment 2.** Based on pseudo time analysis, the authors chose two marker genes (Sox4 and Krt23) to identify bi-phenotypic cells in tissue. Both Sox4 and Krt23 are expressed in cells of ductal origin and it would be essential to add another bi-phenotypic gene to this analysis (such as FKBP5) to show co-expression in given cells. In case immunofluorescence does not work, FISH-based approaches would present a very good alternative.*

We have now performed additional staining for additional biphenotypic markers identified in Figure 3K (**Extended Data Fig. 8a**). Staining for FKBP5 did not work as we could not identify a specific antibody. However, the revised manuscript includes additional immunostaining for NCAM1 and KLF6. NCAM1 shows little staining in the healthy liver while marking ductal reaction structures and biphenotypic cells (positive for NCAM1, KRT19 and ALB) in end stage livers. A limited number of hepatocytes express KLF6 in the healthy liver while this transcription factor is commonly observed in cholangiocytes and hepatocytes in end stage livers. This includes biphenotypic cells (positive for KLF6, KRT19 and ALB). We believe that together these additional immunostainings address the reviewer comments and provide further validations of the genes identified by our single cell analysis.

***Comment 3.** A recent paper (31350390 Single cell analysis of human foetal liver captures the transcriptional profile of hepatobiliary hybrid progenitor) identified a population of hepatobiliary hybrid progenitor of ductal origin that at least shares some markers with the bi-phenotypic cells identified here. What are the difference/similarities between these populations? The other dataset is based on SMART-seq2, so more sensitive, but a comparison might still give interesting insights. At least, this paper has to be discussed.*

We thank the reviewer for this comment and this reference has now been added to the revised manuscript. This report focuses on healthy liver and on a limited number of cells. Thus, a direct parallel is difficult since our biphenotypic cells are mainly detected in end stage livers. However, this study did identify a population of progenitor cells which express a combination of markers (CD24+ CDH6+ CD133+ FGFR2+ SOX9+ GPRC5B+ TROP-2+ SFRP5+ ALB+ STAT1+ CLDN3+ CLDN10+) which

does not overlap with the factors identified by our pseudotime analyses. We suspect that these progenitors are either extremely rare or only present in healthy liver where they play a role in tissue homeostasis.

Comment 4. *Finally, the authors used cholangiocyte-derived organoids from NAFLD end stage patients to address a potential role of the PI3K-AKT signalling pathway, which was enriched in GSEA pathway analysis in the bi-phenotypic cell population. Although the organoids showed similarities to this population and might thus be a good ex vivo system to study the emergence of these cells, the current experimental setup is not conclusive as organoids from healthy donors need to be cultured and compared in parallel. The authors even point out that organoid differentiation can be very heterogeneous. Thus, addition of control organoids is an essential addition to the manuscript.*

We have derived cholangiocytes organoids from healthy and cirrhotic donors and performed differentiation in parallel. Organoids from healthy and cirrhotic donors display the same capacity of differentiation (**Extended data Fig. 19**). However, we consider that cells derived from a disease environment are more relevant to study plasticity mechanisms. Thus, we have decided to use organoids from end stage livers to validate our in silico results.

Comment 5. *Organoids differentiated in the presence of PI3K-AKT pathway inhibitors did not show hepatic marker expression. How was differentiation efficiency assessed in this case?*

We performed qPCR and immunofluorescent staining (**Fig. 4c-e**). Untreated organoids were used as control to ensure that the differentiation did work. Also note, that the revised manuscript includes additional data showing that activation of the same pathway increases transdifferentiation (**Fig. 4e and f**) and answer to Reviewer 1 comment 18.

Comment 6. *Could the inhibitors simply block or halt differentiation? This has to be ruled out.*

To address this comment, small molecule inhibitors the PI3K-AKT-mTOR pathway were applied to ICOs differentiating into biphenotypic cells either at the beginning of the protocol, half way through, or for the final 24h (acute block) (**Fig. 4e and Extended Data Fig 9d**). These experiments showed that inhibiting the PI3K-AKT-mTOR pathway at the end of the differentiation does not inhibit the process, thereby confirming that this pathway is not necessary to maintain biphenotypic cells. on the other hand, blocking the pathway at the earliest time point is most effective at inhibiting differentiation while the intermediate time point impacts the differentiation but to a lesser extent. Taken together, these results confirm that the PI3K-AKT-MTOR pathway signalling pathway is necessary for ICOs to differentiate into biphenotypic cells without being necessary for their survival.

Minor points

Comment 7: *it would be good to add a plot/table to show representation of cell types / patient*

The revised manuscript includes this useful information in **Extended Data 3a**, which shows the proportions of each cell type per patient.

Comment 8: *how were samples treated for snRNA-seq? was each patient one lane on the 10x Chromium (as Extended Data 3 suggests) or were samples pooled? Please clarify in Results and M&M for better clarity.*

One sample was run per lane. We have now added this information in the material and methods.

Answers to Referee 3 comments :

Review for “Acquisition of epithelial plasticity in the human liver during chronic disease progression”

The authors address the important question of tissue regeneration in the context of non-alcoholic fatty liver disease (NAFLD) progression in humans. There are potentially three scenarios for regeneration: stem cell activation, de/re-differentiation, and transdifferentiation. Combining single-cell transcriptome analysis with imaging revealed reorganized zonation profiles and considerable changes in the biliary tree. Moreover, the results found indicate that transdifferentiation between hepatocytes and cholangiocytes, without activation of stem or progenitor cells, is at the heart of the plasticity underlying regenerative capacity. While overall the findings are potentially of great interest, some analyses leading to key hypotheses need to be solidified as some of the claims may not be sufficiently robust at this stage.

Major:

Comment 1. *Cell types. Fig. 1E: it seems surprising that only 25% of cholangiocytes express KRT7. Why is Albumin not shown as a hepatocyte marker?*

KRT7 is expressed in a majority of cholangiocyte but at relatively low level which might not be systematically capture by single nuclei RNASeq. This is a common limitation of such analyses which can be compensated computationally by combining several markers to annotate cell type. We have used at least 10 markers to identify cholangiocytes and thus our annotation did capture the right cells. We have now included a comprehensive heatmap of makers used for annotation (**Extended Data Fig. 3c**).

Please see to our answer to reviewer 2 comment 1. *ALB* expression is not shown in UMAP due background expression associated with free floating RNAs. Indeed, it is well known that single cell preparation can be contaminated by mRNA of genes highly expressed. This is particularly problematic for liver tissues as hepatocytes express extremely high level of genes such *ALB*. Our single nuclei approach significantly decreases this problem, but we still observe some background expression. For all these reasons, we have not used *ALB* as primary marker to annotate hepatocyte in our transcriptomic analyses. However, *ALB* protein was used to identify hepatocytes in most of our immunostainings.

Comment 2. *“We observed the existence of cells “bridging” hepatocyte and cholangiocyte clusters and co-expressing specific markers for both cell types (Extended data Figure 3E).” The argument about the biphenotypic cells seems crucial but it was not very convincing with the short description and ED Fig. 3E. It seems necessary to supplement it with more statistical analysis, and importantly provide a clearer definition of those cells. For example in ED Fig. 3E, it appears that *ABCC2* and *KRT7* are also expressed in the other (non hepatocyte and non cholangiocyte) clusters.*

The previous Extended data Figure 3E only introduced the biphenotypic cells by showing two markers. However, Figure 3 is entirely focused on the characterisation of the cells. In sum, we observed a population of cells bridging hepatocytes and cholangiocytes mainly in end stage disease (**Fig.1 d and f**). This “bridge” suggested the presence of biphenotypic cells co-expressing varying level of hepatocyte and cholangiocyte markers. We then performed detailed subclustering analyses on hepatocytes and cholangiocytes to further define this subpopulation (**Fig. 3a-d**). Using this approach, we identified several subcluster of hepatocytes or cholangiocytes expressing markers specific for each other. This approach identified which subpopulation is transdifferentiating. Importantly, this is a dynamic process and thus biphenotypic cells can express different markers at different level. However, we have been very strict in selecting biphenotypic cells for subsequent analyses by considering only cells expressing high level of multiple markers. Our goal was to avoid cells that could “randomly” express unspecific markers or early intermediate cells (See Reviewer 1 comment 1) and to focus on cells undergoing transdifferentiation. The pertinence of our approach has been confirmed by immunostaining analyses showing that genes identified as differentially expressed (increased) in biphenotypic cells are indeed expressed at the protein level in cells co-expressing hepatocyte and cholangiocyte markers, including *SOX4*, *KRT23*, *NCAM1* and *KLF6* (**Extended Data Fig. 8a and b**).

Comment 3. *Loss of zonation, Figure 2B. The visual discussion of correlation matrices to show the loss of zonation should be supplemented with a statistical argument. Moreover, perturbed zonation in NAFLD is established in mouse models, and in humans it was shown at the level of the lipidome (DOI: 10.1002/hep.28953) proteogenomics (DOI: 10.1016/j.cell.2021.12.018).*

We have now added a supporting figure examining the correlations and provided statistical analyses (**Extended Data 5b**). This confirms that there is a significant difference in correlations of periportal and pericentral markers in the healthy liver and this difference decreases during disease, eventually no longer being significantly different in end stage disease. The suggested references have been included and discussed.

Comment 4. *Figure 3A-D. What makes hepatocyte cluster 9 be part of hepatocytes? For example, it*

appears that cluster 9, but also many other clusters, shows very low Albumin expression. These arguments in favor of the possible dual origin of bi-phenotypic cells needs to be made more convincing.

All cells in the UMAP have been previously assigned as hepatocytes based on the unsupervised clustering and the expression of various markers to guide cell type annotation (**Extended Data Fig. 3c**). The expression shown is relative expression and so it is not necessarily a low level. This expression is just the lowest of the hepatocyte cluster. The low level of ALB in this cluster may also indicate a downregulation of hepatocyte markers which could be due to the transdifferentiation process.

Comment 5. *Are those biphenotypic cells related to hepatoblasts?*

In Extended data Figure 7, only few cells express AFP and SPINK1 (hepatoblasts markers see Wesley et al, Nature Cell Biology 2022) and they are not related to the biphenotypic cells. Importantly, we have a broad experience in characterising hepatoblast. They typically don't express high level of KRT19 and KRT7. They also don't express SOX4, KRT23 or KLF6. Thus, we are confident that biphenotypic cells are not related to foetal liver cells.

Comment 6. *Fig. 3E. How are the p-values calculated? Please show all the proportions for the individual samples, for example using violin plots.*

We have now replaced this plot with one showing all data points. P values were calculated by performing Welch's t-test between the per-patient proportions in different disease stages.

Comment 7. *In a recent paper (DOI: 10.1126/scitranslmed.add3949), the Friedman lab also reported snRNA-seq of patients with NASH, albeit fewer in numbers and covering less disease stages. It would be of interest to assess whether biphenotypic cells are also found in those data.*

We thank the reviewer for this suggestion. The Friedman dataset includes control liver and liver with NASH (fibrosis stage 1-3) but no end stage liver. Nonetheless, we have downloaded and analysed their data using our computational pipeline. Interestingly, we identified a cluster of hepatocytes expressing cholangiocyte markers and the plasticity markers KRT23/SOX4 identified by our analyses (**Figure 3k and l**). Interestingly, the cells included in this cluster (cluster 19) are predominantly coming from livers with advanced fibrosis (fibrosis score 3 in their data) (**re-analysed data shown below**). Together, these analyses confirm the existence of biphenotypic cells in a separate dataset generated with different protocols and a different population of patients. They also confirm that biphenotypic cells mostly appear in the late stage of the disease.

Supporting data Figure – A-C) Cholangiocyte marker KRT7 and plasticity marker KRT23 and SOX4 expression in NASH patients UMAP. D) Subclustering of cells. E) The proportion of cells in cluster made up of the indicated fibrosis score.

Comment 8. Lines 166-184: Arguments based on co-detection of two genes are dangerous in scRNA-seq or snRNA-seq due to the low detection rate. At least this need to be assessed in comparison to a proper null model. Most genes will in fact not be co-detected even if they are both present in a cell. Moreover, the arguments made form snRNA-seq should be taken with care since the effective lifetime of nuclear RNAs is very short. Therefore absence of nuclear transcript may not mean absence of cytoplasmic transcript or even protein. Thus the suggestion that plasticity increases with time needs to be strengthened. It is also risky to base such an argument on few marker genes. A systematic multi-gene analysis would be more convincing.

We agree with the reviewer on the risk associated in using expression of only 2 genes to annotate cells and accordingly biphenotypic cells were defined by unsupervised clustering and the expression of various markers to guide cell type annotation. Using this approach, the number of biphenotypic cells remain extremely low in the early stage of the disease. Furthermore, we have validated the increase of biphenotypic cells by IF on tissue slides. Finally, factors associated with plasticity KRT23, SOX4, KLF6 and NCAM1 increase with disease progression. All these observations suggest that plasticity do increase over time.

Concerning progenitors, we have now analysed the expression of 4 markers ki67, TROP2, LGR5, AFP in combination (**Extended Data Fig. 7a-d**). This analysis confirms that cells co-expressing these markers are not related to transdifferentiating biphenotypic cells.

Comment 9. RNA velocity is not expected to work with snRNA-seq due to the short life time of nuclear transcripts. Thus it should be demonstrated that it works here (which is not completely impossible but unlikely), for example by showing that intron and exon signals are shifted in function of pseudotime.

We agree with the reviewers that snRNA-Seq can change the nature of the mRNA captured for sequencing. Accordingly, we see an increase in proportion of intron matching reads in our dataset (data shown below). However, RNA velocity has been used recently in publications on snRNAseq data (Kang et al 2023, Genome Medicine, Adewale et al 2022, MedRxiv) and the analysis of spliced/unspliced transcripts can be performed on similar assumptions as for single-cell data (Gorin et al 2022, PLoS Computational Biology). Thus, we are confident that our conclusion regarding the directionality of transdifferentiation is accurate.

Furthermore, we focus on the Monocle inferred order for our pseudotime analyses, which better underlines the characteristics of plasticity, in contrast to latent pseudotime, derived from the velocity analysis. Finally, we note that the validity of RNA velocity conclusions cannot be reached by summarising the dynamics of expression for individual genes. We agree that further analyses of regulatory interactions and networks, taking into account the types of alternative splicing observed within the system, are a natural next step, but these analyses are beyond the scope of the current manuscript.

Comment 10. *PI3K-AKT signaling in biphenotypic cells and ICOs. Is it possible that the activity of mTOR in ICOs is reflecting the high demand on protein synthesis in this system? It is unclear what the blot in Fig. 8C is showing. Is beta actin meant as control? Show the quantifications and replicas.*

The western blot shows the action of the inhibitors and beta actin was included as loading control. We agree the inclusion of this figure is not really necessary and thus it was removed in the new version of the manuscript. Importantly, we perform additional experiments showing that PI3K-AKT-mTOR is necessary for the differentiation of ICOs into biphenotypic cells (see answers to reviewer 1 comment 18 and to review 3 comment 6) while the inhibition of the signalling after differentiation has no effect.

Thus, the function of PI3K-AKT-mTOR is unlikely to be only related to protein synthesis, but we can't rule out that protein synthesis demand may increase during transdifferentiation and mTOR may play a role there.

Minor:

Comment 11. Fig 1F: Caption was cut.

The figure has been corrected.

Comment 12. "Importantly, QCs were performed to confirm that these cells were not due to doublets or RNA contamination." Please provide more details about this

See answer to reviewer 2 comment 1. The material and methods part has been modified to include more details.

Comment 13. Fig 4A may be significantly reduced in size or removed.

We have removed this panel to allow for the addition of new data.

Comment 14: Typos, etc:

33 no comma after here. Corrected

34 - "from across" Corrected

41 - comma before thereby Corrected

41 - where does insulin signaling come from all of a sudden?

Insulin is a major activator of PI3K/mTOR this pathway in the liver.

52 – comma Corrected

57 - comma before (ii) Corrected

86 - biopsies in plural Corrected

95-100 - hepatocytes highest changes, what about stellate cells that actually produce fibrotic tissue?!

Stellate cell do also show an end stage disease signature, but the transcriptomic profile of hepatocytes appear to be more affected. The stellate cells can be examined by the publicly available data (shiny cell app provided in the 'data availability' section. – TO DO – confirm with Irina the updated link

114 - hypoxia is not a pathway. Corrected

119 - genes italics Corrected

150-152 - I don't believe you can define origin by clustering analysis or markers! Corrected

234 - do not capitalize Rapamycin Corrected

238 - no space after comma Corrected

figure legends - spaces around mathematical symbols and units Corrected

Answers to Referee 4 comments :

The manuscript from Gribben, Galanakis and colleagues focuses on the progression of human liver disease to identify potential changes in cellular plasticity associated to liver regeneration and disease. The experimental methodology is robust and state-of-the art, including single-nuclei sequencing of human liver tissues at different stages of chronic liver disease, and human liver organoids. By using these experimental approaches, the authors detected a biphenotypic cell population expressing both hepatocytes and cholangiocyte markers at end stages of liver disease. This is suggestive of cellular plasticity of liver hepatocytes and cholangiocytes occurring in chronic liver disease. More in detail, the authors hypothesise that hepatocyte-into-cholangiocyte and cholangiocyte-into-hepatocyte trans-differentiation occurs in chronic liver disease.

Comment 1: *Cellular plasticity of hepatocytes and cholangiocytes (i.e. the capacity to both hepatocytes and cholangiocytes to give rise to each other) has been extensively detected in mouse and associated to regeneration of the liver epithelial compartment after prolonged liver injury (Yanger et al., Genes&Dev, 2013; Font-Burgada et al., Cell, 2015; Raven et al., Nature, 2017; Russell et al., Hepatology, 2018; Deng et al., Cell Stem Cell, 2018; Manco et al., J. Hepatol, 2019). Importantly a biphenotypic cell population has been previously identified in human liver disease (Yanger et al., Genes&Dev, 2013, PMID: 23520387 Robust cellular reprogramming occurs spontaneously during liver regeneration). This particular reference should be added to the manuscript as it is highly relevant.*

Altogether, the findings of this manuscript enable to robustly confirm the presence of a human biphenotypic cell population; at the same time evidence in literature questions the novelty of this manuscript since this biphenotypic cell population had been previously detected in the liver.

We agree with the reviewer that mouse models have been used extensively to study regenerative processes occurring during liver injuries. However, these studies have proposed several mechanisms including the activation of stem cells or progenitors, dedifferentiation into developmental progenitor and cholangiocyte/ hepatocyte transdifferentiation. All these mechanisms were mentioned in our introduction and referenced using reviews commonly used in the field. We did not reference original publications due to editorial restrictions imposed on the number of references. We will be happy to modify these aspects under editorial approval.

More importantly, all these mechanisms are true in the context of mice models. They simply rely on the nature of the injury and the genetic model used. However, our results demonstrate for the first time that transdifferentiation is the mechanism occurring during chronic liver disease in human and exclude a role for progenitors or dedifferentiation.

In addition, the reports mentioned above reinforce the broad interest of our study. Indeed, these studies rely on short term injury in mouse (months compared to years in human) using toxin or genetic manipulation which are not directly relevant for human diseases while targeting specific hepatic cell type (i.e cholangiocytes vs hepatocytes). In fact, few studies if any have analysed regenerative processes in mouse model commonly used for modelling NAFLD/NASH such as the western diet mice. Keeping these aspects in mind, the studies mentioned by reviewer 4 have observed transdifferentiation of hepatocytes/cholangiocytes after a few weeks and in a proportion up to 10-40%. This process actively repairs damaged tissues and thus have been rightly interpreted as regeneration. However, our observations in the context of human disease are fundamentally different. Our single cell analyses show that acquisition of plasticity is very progressive in human and only occurs after years of repetitive injuries once the liver environment has been significantly compromised. This transdifferentiation is induced by combination of stress signals and change in the microenvironment while being limited to a relatively few cells. Finally, key markers associated with transdifferentiating cells are also associated with liver cancer. Thus, transdifferentiation is not a regenerative process in human. This is a disease mechanism marking aggravation and not-tissue repair. We believe that this is a paradigm change in the field of organ regeneration.

Concerning the reference Yanger et al., Genes&Dev, 2013, PMID: 23520387, we agree that this manuscript is very relevant and thus this study is now cited in our revised manuscript. This manuscript used mouse models and included limited human data showing the presence of hepatocytes displaying the expression of Sox9 in liver of patients with Joubert Syndrome, Hepatitis C infection, and necrosis using only histology. These cells are likely similar to the intermediate hepatocytes mentioned by Reviewer 1 (See answer to reviewer 1 comments 1 and 2) and commonly observed by others. These cells are indeed not novel. However, it is impossible to define if these HNF4a/Sox9 are similar to the transdifferentiating cells identified by our study. On the contrary, Sox9 was not identified as plasticity factors in our study indicating that this factor is not specifically expressed in our transdifferentiating population. Overall, our study demonstrates formally for the first time that transdifferentiation does occur in human liver in the context of chronic disease.

Comment 2: *In addition, only a limited characterisation of this population has been performed in this manuscript. Specifically, i) the role of this biphenotypic cell population in liver disease remains unclear; ii) data provided in this manuscript regarding a role of PI3K/AKT/mTOR in the specification/maintenance of biphenotypic cells, are inconclusive. Regarding the latter, the authors show that inhibition of PI3K/AKT/mTOR impairs the expression of hepatocyte markers in differentiated human liver organoids. However, it remains unclear if PI3K/AKT/mTOR inhibition in organoids blocks the specification of hepatocyte cell identity or the generation of biphenotypic cells.*

In addition, levels of insulin in the serum of patients with chronic liver disease peaks at cirrhosis, whereas the authors observe biphenotypic cells mainly at the end stage of chronic liver disease, thus

suggesting that additional or different signals drive cellular plasticity and the specification of liver biphenotypic cells.

Our analyses show that transdifferentiating cells appear in end stage liver when the liver is not functional anymore and when cancer risk is extremely high. So, biphenotypic cells mark disease progression and are a consequence of injury. Our immunostaining analyses also suggest that they are not organised or located in “healthy” region suggesting that these cells are unlikely to have a role in tissue repair. These cells simply mark extremely damaged livers.

We have performed additional experiments showing that PI3K/AKT/mTOR is necessary for the expression of hepatocyte markers in cholangiocytes while not been required to maintain biphenotypic cells (**Fig. 4e and Extended Data Fig. 9d**) (See answers to reviewer 1 comment 18 and reviewer 2 comment 6). Also, we have now added new data showing activation of this pathway enhances differentiation (**Fig. 4e and f**), reinforcing the important role of this signalling pathway in regulating the transdifferentiation.

Furthermore, we fully agree that additional signalling pathways are likely to play a role in transdifferentiation. Accordingly, the revised manuscript includes data showing that FGF13 and YAP/TAZ may also being involved (**Extended data Figure 10**) (See answer to reviewer 1 comment 18) and we believe the transdifferentiation in vivo is likely induced by a complex signalling interplay of various pathways over an extensive period of time. The discussion has been modified to reinforce this point.

***Comment 3:** In summary, the manuscript adopts a robust experimental approach to confirm the presence of a biphenotypic cell population in human liver; however, as it is, this manuscript shows limited novelty since this biphenotypic cell population had been previously identified in both mouse and human livers. Further characterisation of this biphenotypic cell population should be performed to determine the role of this cell population in the liver response to damage/regeneration and the mechanisms leading to their specification.*

We have performed additional computational analyses to further characterised biphenotypic cells including showing that they are only very rarely found to be proliferative (**Extended data Fig. 8c**) and now show in greater detail that these do not express stem/ progenitor markers (**Extended data Fig. 7**). We have also compared these cells to small and big cholangiocytes populations, and this data indicates biphenotypic cells are more similar to the cholangiocyte of small ducts (**Extended Data Fig. 7a**) (see answer to Reviewer 1 comment 1). This, combined with our observation that we observed biphenotypic cells towards the ends of small ducts (such as in **Fig. 3e**) reinforces that they may be located more in that part of the biliary tree. We have also re-analysed published single cell data from NASH patients, and we found a very similar population of biphenotypic cells which are also predominantly found in the most advanced stage of disease they analyse (NASH Fibrosis 3) reinforcing that the cells we identify are present in late stage disease (see answer to reviewer 3 comment 7).

Additionally, we validated the expression of additional biphenotypic markers such as NCAM1 and KLF6 (**Extended data Fig 8a and b**). Again, none of the data generated in our analyses suggest a specific function for these cells. They simply mark disease progression.

Concerning novelty please refer to answer to reviewer 4 comment 1.

Comment 4: *All immunostaining experiments throughout the manuscript should be quantified and n= cells analysed and n= biological/technical experimental replicates should be indicated. Statistical analyses should be applied and reported in the Figure Legends.*

All our immunostaining has been performed at least on 4 different patients from the appropriate disease stage and across multiple tissue slides. The methods section has been updated accordingly. Importantly, we have not used immunostaining to quantify the number of cells expressing a specific marker. These analyses are used to (i) confirm the presence/ absence of a cell population or ii) demonstrate co-expression of key markers. Furthermore, our single nuclei approach allows precise cell quantification and is an efficient approach to capture all cell types including rare cells from tissue, compared to scRNAseq, where cell types can be lost disproportionality during tissue dissociation.

The material and method part has been modified to provide all the information concerning statistical analyses and this information has been included in figure legends when relevant.

Comment 5: *The overwhelming majority of the tissues have been obtained by white British individuals; this should be clearly stated in the text and captured as a diversity statement.*

This information has been added in the material and method part

Comment 6: *The authors have confirmed the presence of ALB+ KRT7+ biphenotypic cells previously detected in human livers, and have found them prominently at the end stage of chronic liver disease. However, the role of these cells in liver disease remains unclear. The authors should perform experimental approaches aimed at determining the role of these cells. Since chronic liver disease can degenerate into cancer, one possibility would be to investigate the presence of biphenotypic cells in different types of human liver cancer and perform analyses to determine whether they are associated to cancer drivers and carcinogenic processes.*

We agree that these experiments would be extremely interesting, and we have performed preliminary work showing liver tumours contain biphenotypic cells expressing plasticity factor such as KRT23 (figure below). Moreover, plasticity factors such as SOX4, KLF6 and NCAM1 are associated with liver cancer. These data reinforce our hypothesis that acquisition of plasticity could be linked to tumorigenesis. However, the functional demonstration that liver tumours could originate from biphenotypic cells is extremely challenging to establish. Indeed, it is unlikely to all biphenotypic cells can give rise to cancer cells. They are just too many of them. Thus, additional events such as somatic mutation will be required and modelling this aspect in the context of transdifferentiation will be

extremely time consuming if even possible. Thus, we believe that such experiments will go far beyond the scope of the current study.

[Figure redacted]

[Text redacted]

Comment 7. *The authors have identified different clusters associated to biphenotypic cells. This seems particularly relevant in cholangiocytes since the analyses suggest that biphenotypic cells in some cholangiocyte clusters may originate at earlier stages of liver disease. Together this suggests that the biphenotypic cells form an heterogenous cell population. The authors should determine if biphenotypic sub-populations have different features, including expression of stem-cell genes, proliferation potential and regenerative and carcinogenic capacity in vivo and/or in vitro using organoid systems*

Biphenotypic cells identified by our single cell analyses represent transdifferentiating cells transitioning between cellular identity. Thus, we expect a dynamic population expressing different level of different markers. In addition, the revised manuscript contains additional data reinforcing our previous results showing that biphenotypic cells do not express stem cell or foetal markers (See answers to Reviewer 1 comment 10, Reviewer 2 comment 8 and Reviewer 3 comment 3). Finally, generation of biphenotypic cells in vitro is associated with inhibition of proliferation. Furthermore, this process is very inefficient and less than 10% of cells express Alb, while FACS sorting ICOs is very challenging (lack of cell surface markers, cell viability, extraction from Matrigel etc.). Finally, carcinogenic capacity is also defined by complex combination of somatic mutation. For all these reasons, biphenotypic cells generated in vitro are unlikely to directly form tumours in vitro or in vivo.

Comment 8. *The authors have found no evidence of a progenitor signature associated to biphenotypic cells by checking co-expression with stem-cell genes such as LGR5 and TROP2. Thus, they conclude that in human chronic liver disease trans-differentiation (hepatocyte-to-cholangiocyte and cholangiocyte-to-hepatocyte) occurs rather than de-differentiation of hepatocytes/cholangiocytes into bipotent progenitors capable of giving rise to both cell types. However, these analyses were conducted at the end stage of liver disease. To determine if liver progenitors are detectable and could give rise to the biphenotypic cells detected at the end stage of chronic liver disease, the author should check if expression of LGR5 and TROP2 overlaps with subsets of makers of different clusters of biphenotypic cells at earlier stages of chronic liver disease.*

Our analyses were performed on cells from all the disease stage and we did not observed cells expressing stem cell or progenitor marker while being associated with transdifferentiating cells. In addition, we have reinforced these analyses by combining a different markers and again could not detect a consistent population of cells related to biphenotypic cells (See answers to Reviewer 1 comment 10, Reviewer 2 comment 8 and Reviewer 3 comment 3 and Extended Data Fig. 7).

Comment 9. Line 202: Proliferation of SOX4+ and KRT23+ is described as number of cells expressing mKi67 but not shown as a Figure.

We have generated the necessary figure (Extended Data Fig. 8c).

Comment 10. The authors show significant biliary tree remodelling in chronic liver disease. Does this impact the generation of biphenotypic cells? To address this, the authors could take advantage of organoids grown with physical constraints resembling the remodelling of the biliary tree observed in the tissue.

We believe that as part of the biliary remodelling the ends of terminal ducts change, from being single cells in the healthy, to bulkier blunt ends in the end stage (**Extended Data figure 5d**) and we have observed biphenotypic cells at the ends of ducts (**Fig. 2e**). So this reorganisation may impact the generation of biphenotypic cells. Additionally, we have found that the biphenotypic cells tend to resemble the cell of smaller ducts (**Extended Data Fig. 7a**) which is in line with the idea that this region may contain more biphenotypic cells

To define the importance of physical constraint, we have performed the ICO differentiation in the presence of increasing concentration of collagen. The addition of Collagen I to the Matrigel did not appear to increase the differentiation (**Extended data Fig. 10f**). However, it did cause the formation of tube-like structures and the emergence of ALB+ cells in clusters. So, increase stiffness and change in ECM composition is likely to play a role in the tubulogenesis observed in ductular reaction. However, the function of these factors in plasticity is not clear reinforcing our hypothesis that this process is induced by the combination of different signalling pathways.

Comment 11. The authors should perform experimental approaches aimed at determining a role of PI3K/AKT/mTOR in the specification of biphenotypic cells in chronic liver disease. Based on the results shown in organoids it remains unclear if PI3K/AKT/mTOR inhibition blocks the specification of hepatocyte cell identity or the generation of biphenotypic cells. In this Reviewer's opinion, experimental approaches in mouse models would be appropriate to determine the signalling determining the specification of biphenotypic cells.

See answers to reviewer 1 comment 18 and reviewer 2 comment 6. The revised manuscript includes additional data demonstrating that PI3K/AKT/mTOR signalling is necessary for the induction of hepatocytes markers in cholangiocytes. The same pathway is also sufficient to promote the expression of these markers. However, PI3K/AKT/mTOR is not necessary to maintain biphenotypic cells thereby excluding a simple role in cell survival. In addition, we performed additional experiments showing that FGF13 and YAP/TAZ signalling are also involved in the process of transdifferentiation. Taken together, these results reinforce our previous conclusions and suggests multiple pathways are likely involved in the acquisition of plasticity.

As mentioned by the reviewer, mice models have been extensively used to characterise regenerative process in the liver. However, most of these studies rely on models which are not entirely relevant to human disease (See answer to reviewer 4 comment 1). Thus, we decided to explore the possibility to use western diet mice which are commonly used to model NAFLD/NASH. This model is particularly interesting since it does not rely on a chemical toxin and has been shown to reproduce in part the metabolic syndrome observed in human. To validate the interest of this model, we performed immunostaining on western diet mice liver and we observed ductular reaction and the induction of plasticity factors (Sox4 or KRT23) in cholangiocytes (OPN+) (Figure shown above). This induction is even pronounced after prolonged period of western diet which is following diethylnitrosamine (DEN) treatment known to induce tumorigenesis. These data suggest that transdifferentiation could also take place in these mice and could mark disease progression. However, the first sign of liver injury takes up to 30 weeks to appear while the phenotype remains relatively mild. Thus, functional studies are very challenging to perform in such models. Furthermore, inhibition of PI3K/AKT/mTOR is likely to interfere with the metabolic syndrome rendering data interpretation extremely complicated. Thus, functional studies in mice would be extremely time and resource consuming while they will have limited impact on our conclusions.

[Figure redacted]

[Text redacted]

Reviewer Reports on the First Revision:

Referees' comments:

Referee #1 (Remarks to the Author):

Rebuttal comments revised manuscript

The authors have done a good effort to address the technical comments and this has improved the manuscript. Unfortunately, some important comments have been left unanswered or inadequately addressed. There still is a discrepancy between the conclusions that have been made based on the snRNAseq results and what the validation is showing. The snRNAseq data set has value but over-interpretation without acknowledging the limitations leads to inaccurate conclusions. Absence of certain gene signatures in one specific data set does not mean that these cells do not exist, especially when they are being visualised by other methods. Please see below for more detailed comments.

Rebuttal to comment 1.

The authors state the following: "our study uncovers transdifferentiation events occurring between hepatocytes and cholangiocytes without stem cell or progenitor activation.". Yet, all of the immunohistochemical stainings show ductular reaction and hence progenitor cell activation. It are the terminal branches of the biliary tree that become response to epithelial damage, start to proliferate and differentiate to the cell type that is damaged, depending on the underlying aetiology. It is not because the authors fail to find a robust signature of DR in their snRNAseq data, that it does not exist. They are proving the existence of progenitor cells in their immunostaining. One cannot just ignore 50 years of research in the field, thousands of papers and throw away accepted nomenclature.

It is very likely that the authors are not powered enough to identify the DR signature, especially in earlier stages where DR is less prominent (as indicated by the rebuttal on comment 10).

Additionally, the authors only focus on 1 aetiology. One cannot simply make general conclusions for all of the aetiologies. DR has been described in early NASH (Gadd V et al 2014 Hepatology). More granularity is needed within the different cell populations and this needs to be validated properly, either by immunohistochemistry or, ideally, spatial transcriptomics/multiplex.

Extended data Fig 3 F shows, assuming the labelling is wrong, BICC1-/KRT7- cholangiocytes and ABCC2- hepatocytes. What are these cells? Another population of 'transdifferentiation cells'? Cluster identification is based on overall high expression of markers in these clusters as compared to other clusters, but this does not mean the markers are unique! SPP1, for example, has been described as a marker of lipid-associated macrophages by the group of Charlotte Scott. This shows that one cannot just make conclusions on cluster annotations and projected velocity plots. Of note, the group of Neil Henderson found KRT19+ positive mesenchymal cells in their single cell data set (Nature 2019). They did not claim transdifferentiation of mesenchymal cells. Extended data figure 6, only focusses on the main cholangiocyte cluster. I would recommend to broaden the scope and be cautious with cell annotations and the interpretation. It is possible that the DR signature is not within the more mature cholangiocyte cluster.

ALB is a broad marker. Different online data sets indicate a high expression in cholangiocytes and even endothelial cells (<https://shiny.igc.ed.ac.uk/livercellatlas/> & <https://www.livercellatlas.org>). I would suggest to look for another marker to indicate hepatocyte lineage or phenotype. This makes the conclusions made on ALB expression questionable.

Rebuttal on comment 2

The term 'intermediate' indicates a cell state between a mature hepatocyte and a cholangiocyte (either differentiation or dedifferentiation), which in the past was usually visualised by the use of immunohistochemistry. How does this differ from biphenotypic cells? Why abandon general nomenclature accepted nomenclature? Furthermore, the authors restrict themselves to 1 aetiology and are probably not powered enough to identify the small population in pre-cirrhotic cases. Again, this does not mean it does not exist.

Rebuttal on comment 3

I fail to see the transcriptional changes in hepatocyte zonation. How does the transcriptome and phenotype from hepatocytes in zone1-3 change during NAFLD progression? The authors only show healthy versus end-stage.

Rebuttal on comment 9.

As mentioned earlier, DR can be observed in pre-cirrhotic cases, especially in at-risk NASH patients with a high NAS/SAF score.

Rebuttal on comment 14.

My comment was directed at the underlying biological differences, not the technical approach. There is a reason why these patients got transplanted. Decompensation is a completely different pathological process, meaning there are distinct differences between the NASH F4 needle biopsies and explant material.

Rebuttal on comment 19

The comment "PSCs and PBC affect the biliary tree while NAFLD is the most common chronic liver disease. Furthermore, we are convinced that our conclusions can be extended to other liver disease involving chronic injury of hepatocytes." is not fully correct. In PSC/PBC you also have damage to the hepatocytes (cholestasis, dedifferentiation, metaplasia); in ALD, the bile ducts are also chronically damaged; in ACLF you have a massive activation of the progenitor cells without differentiation. One cannot just generalise a conclusion without proof. Being convinced is not a proper scientific approach. I would suggest to talk with your experts, as clearly mentioned in the rebuttal to comment 1. They will be able to inform you better on the pathophysiology of different liver diseases.

Rebuttal on comment 22.

Some effort on using the right NAFLD nomenclature would be welcome (either the old or new consensus).

Referee #2 (Remarks to the Author):

The authors have addressed my concerns in the first round of revisions and I am happy to support publication.

Referee #3 (Remarks to the Author):

The authors have provided detailed answers to most of my points.

The arguments in support of Figure 2 (zonation) have been much improved, Extended Figure 5b looks convincing. Figure 2: Indicate on the plot which matrix corresponds to which stage.

“For all these reasons, we have not used ALB as primary marker to annotate hepatocyte in our transcriptomic analyses.” Yet, Alb is listed in Table S3 for hepatocytes.

Figures 3B-D: the analyses in Figure 3 in favor of hepatocyte-like cholangiocytes and vice versa remain relatively weak. It could be the choice of the representations, but if the signatures remain quite noisy. Is the effect strong for cholangiocyte-hepatocyte compared to endothelial-hepatocyte? That would be an important control.

Figure 3L: It is difficult to know how representative such fairly small images are, ideally such arguments (also ED Figure 1) would require quantifications.

Referee #1, comments upon responses to Referee #4 (Remarks to the Author):

Comments to the authors:

The authors have mainly addressed my comments using a computational approach. The characterisation of the biphenotypical population is still rather limited and does not provide any clarity on the origin or localisation of these cell types. The additional markers included in the immunostainings do not answer this, and do not really answer the question how this relates to disease stage and outcome. Additional functional models are still lacking.

Concerning the rebuttal for my first comment, the authors should have described the biphenotypical cells more in detail to increase the novelty of the paper. The authors state that they have shown transdifferentiation without the role for progenitors. In my opinion, the authors have not provided enough scientific evidence that progenitor cells do not exist. Extended data figure 1d suggests different ways of differentiation depending on the location and stage of the disease: ductular reaction (previously described as immature bipotential cells expressing markers of biliary lineage) and plasticity of mature cholangiocytes in end-stage explanted organs. A thorough characterisation in a spatial context would have provided clarity on this issue. The few markers selected for protein validation do not fully answer this, limiting the novelty of the manuscript.

In line with this, a functional approach using in vivo models (ref comments 3, 6 and 11) would have provided useful insights. The authors missed an opportunity here to fully answer the comments. Mouse models for advanced NAFLD (CDAA-HFD or HFD-CCL4) combined with an intervention or with

genetic KO background (ideally with lineage tracing) would have given clarity on the origin and function of the biphenotypic population in chronic disease and cancer. A Western diet mouse model is too mild to make conclusions on progenitor cells or trans-differentiation.

Previous work has shown that bipotential Lgr5+ cells can be differentiated towards mature cholangiocytes [Schneeberger, K. et al Hepatology 2020]. Where do these LGR5+ cells reside in the liver compared to the biphenotypical cells described in the current manuscript. For comment 8, evidence on protein level in a spatial setting should have been added to address this question. This also questions the resolution of the sequencing if you are not able to pick up well-described progenitor cell markers.

The biliary tree has different phenotypes of cholangiocytes. The used organoids seem to be rather immature, resembling small cells of the terminal branches of biliary tree. As mentioned in comment 10, what happens if one starts from mature mucin producing cholangiocytes? E.g. organoids grown with physical constrains to induce a more mature state and then push them towards hepatocytes. This has not fully been answered, which would have provided insight into potential origin of the biphenotypic cells. Of note, the authors state in the rebuttal 'So, increase stiffness and change in ECM composition is likely to play a role in the tubulogenesis observed in ductular reaction.' Ductular reaction is not the same as tubulogenesis. Tubulogenesis is the formation of ducts, observed in human biliary diseases. If the authors push their model towards more mature cholangiocytes and the formation of bile ducts, why do these cells express albumin? This raises further questions.

Author Rebuttals to First Revision:

We would like to thank the editor and reviewers for their comments and for reviewer 1 for additionally considering our response to reviewer 4's comments. Please see below a point-by-point response.

Referees' comments:

Referee #1 (Remarks to the Author):

Rebuttal comments revised manuscript. The authors have done a good effort to address the technical comments and this has improved the manuscript. Unfortunately, some important comments have been left unanswered or inadequately addressed. There still is a discrepancy between the conclusions that have been made based on the snRNAseq results and what the validation is showing. The snRNAseq data set has value but over-interpretation without acknowledging the limitations leads to inaccurate conclusions. Absence of certain gene signatures in one specific data set does not mean that these cells do not exist, especially when they are being visualised by other methods. Please see below for more detailed comments.

Comment 1: Rebuttal to comment 1. The authors state the following: "our study uncovers transdifferentiation events occurring between hepatocytes and cholangiocytes without stem cell or progenitor activation." Yet, all of the immunohistochemical stainings show ductular reaction and hence progenitor cell activation. It are the terminal branches of the biliary tree that become response to epithelial damage, start to proliferate and differentiate to the cell type that is damaged, depending on the underlying aetiology. It is not because the authors fail to find a robust signature of DR in their snRNAseq data, that it does not exist. They are proving the existence of progenitor cells in their immunostaining. One cannot just ignore 50 years of research in the field, thousands of papers and throw away accepted nomenclature.

It is very likely that the authors are not powered enough to identify the DR signature, especially in earlier stages where DR is less prominent (as indicated by the rebuttal on comment 10). Additionally, the authors only focus on 1 aetiology. One cannot simply make general conclusions for all of the aetiologies. DR has been described in early NASH (Gadd V et al 2014 Hepatology). More granularity is needed within the different cell populations and this needs to be validated properly, either by immunohistochemistry or, ideally, spatial transcriptomics/multiplex.

We would like to thank the reviewer for these additional comments which greatly helped to understand their perspective on our results. First, the choice of terminology might have caused some confusions and we apologise for this situation. Following more detailed reading of literature related to histology, we better understand the comments of the reviewer. In histology the terms ductal reaction and progenitor cell activation are often used together (as used in the reviewer comment above), almost interchangeably and are based on cell morphology and markers at the protein level (mainly KRT7). In contrast, our study is referring to progenitor cells in the context of stem cell and developmental biology (i.e. proliferative cells with the capacity to differentiate into hepatocytes). These foetal progenitors and adult stem cells are commonly identified by leading groups in the hepatology field using markers such as AFP and LGR5 (please refer to the following literature: (Huch et al., 2013; Deng et al., 2018; Wesley et al., 2022)). Thus, we have changed the wording in the manuscript to clarify that we are considering foetal progenitor and adult stem cells.

In addition, we fully agree that DR is observed during progression of NAFLD/NASH. Our 2D immunostaining analyses confirmed the previous reports mentioned by the reviewer (**Extended Data Fig. 1d**), while our 3D immunostaining further shows the extent of the biliary tree reorganisation (**Fig. 2d and supplementary videos 3 and 4**). Importantly, the 3D imaging

approach take advantage of the latest tissue clearing technology to visualise DR in 3D. These analyses show that the DR is an organised process leading to a profound remodelling of the biliary tree. So, we fully agree with the reviewer about the importance of DR and its link to disease progression.

However, all the analyses performed so far do not establish a functional link between DR induced by chronic injury, acquisition of plasticity and transdifferentiation. In fact, this has been extremely difficult to demonstrate in humans, especially in the context of chronic liver disease. Studies in animal models vary greatly in their conclusions (He *et al.*, 2014; Rodrigo-Torres *et al.*, 2014; Raven *et al.*, 2017) and systematically rely on acute injury induced by toxin (He *et al.*, 2014), toxic diet (Jörs *et al.*, 2015) and complex genetic alteration (Raven *et al.*, 2017) which are not relevant for human pathology. Thus, a formal demonstration that DR induced by chronic injury results in the production of hepatocytes has remained challenging to fully demonstrate in the context of human disease. This aspect is well described in the recent review published by key opinion leaders in the field (Gadd, Aleksieva and Forbes, 2020). This review accurately described the state of the field and its complexity, while underlining key questions that remain to be answered. One of their main conclusions is: *‘The extent to which BEC (cholangiocyte)-mediated hepatocellular regeneration occurs in different types of liver injury has long been debated, as it is unclear whether this spectrum represents biliary differentiation or hepatocyte transdifferentiation or a combination of both’* and they also note that *‘current data are unclear as to whether cholangiocytes undergo direct transdifferentiation or transit through an intermediate “progenitor” state following de-differentiation. The use of scRNA-seq to characterize the transcriptional profile and phenotypic state of regenerating cells could be crucial for resolving this’*. Thus, we believe that our study fits well in the current literature, and these quotes highlight that our data is not at odds with previous knowledge. On the contrary, our results provide novel information regarding liver regeneration in the context of chronic liver disease especially NAFLD/NASH.

In sum, we agree with the reviewer. DR is an important event which marks disease progression, and which could be involved in regenerative processes. However, our study does not establish a cause to consequence between DR and regeneration. In fact, demonstrating this point is beyond the scope of our study as it will require a different set of experiments. For example, we will need to identify factors inducing DR and then demonstrate that these factors also increase plasticity of cholangiocytes. Thus, we have modified the conclusions of our manuscript to put our results in the context of the broader literature on ductular reaction and to insist that we can't exclude DR as necessary step in the acquisition of plasticity.

Concerning the resolution of our analyses, we decided to re-analyze our snRNAseq data in more detail to specifically detect cells co-expressing DR markers. Using this approach, we were able to identify a cholangiocyte population co-expressing DR markers, which increases with disease progression (**Rebuttal Figure 1a**). Thus, the resolution of our snRNA-Seq allows for the capture of these cells, which rarely express plasticity factors present in transdifferentiating biphenotypic cells mainly observed in end stage liver (**Rebuttal Figure 1b**). Importantly, this observation does not imply that DR cholangiocytes cannot give rise to biphenotypic cells.

Rebuttal Figure 1 – Numbers of cell co-expressing the indicated number of listed markers for (a) Ductal reaction in all cholangiocytes, (b) plasticity markers in all cholangiocytes.

Similarly, we have examined the hepatocyte population and plotted the numbers of cells which co-express plasticity markers, cholangiocyte markers and intermediate hepatocyte markers. This analysis identifies cells with an intermediate hepatocyte phenotype at the NASH and cirrhosis stages of disease progression (**Rebuttal Figure 2a-c**) and biphenotypic cells were also detected at these stages (**Rebuttal Figure 2d**). However, the plasticity signature we describe in this paper is mostly evident at the end stage disease and does not overlap with intermediate hepatocytes at early stage (**Rebuttal Figure 2e**).

Rebuttal Figure 2 – Numbers of cell co-expressing the indicated number of listed markers for (a) intermediate hepatocytes in all hepatocytes, (b) KRT19 specifically in the KRT7/ TROP2 double positive cells (intermediate hepatocytes), (c) plasticity markers in the KRT7+/ TROP2+/ KRT19-ve cells, (d) cholangiocyte markers in all hepatocytes, (e) plasticity markers in all hepatocytes.

Thus, our data does capture events such as DR and intermediate hepatocytes. However, these cells appear at an earlier stage of the disease and are different from transdifferentiating biphenotypic cells observed in end stage. We have now added text to the manuscript to clarify this point and that there is a likely contribution of DR and intermediate hepatocytes to our snRNAseq data and immunostaining. We have also modified our conclusion to indicate that our results do not exclude that DR cholangiocytes or intermediate hepatocytes could be precursors of biphenotypic cells.

Regarding potential use of spatial transcriptomics, this approach has important resolution limitations. Indeed, methods currently available can either detect genes highly expressed (MERFISH) or can only capture group of cells (Visium). Thus, spatial transcriptomic will not address the reviewer comments, whereas snRNAseq and IHC have respectively greater resolution and allow the visualisation of protein.

To add more detail regarding marker expression at the single cell level on tissue sections We have performed more multiplex staining on end stage tissue sections, which allowed us to examine more combinations of DR, plasticity, cholangiocyte and hepatocyte markers together and added more spatial detail. We have now analysed plasticity markers SOX4 and KRT23 expression alongside established hepatocyte marker HepPar1 and cholangiocyte marker KRT7 (**Extended data Figure 8a-b**). We observe SOX4/HEPAR1, SOX4/ KRT7 and KRT23/HEPAR1 double positive cells, and KRT23/SOX4/KRT7 triple positive cells, proving that cells can co-express plasticity markers in end stage disease. Previously we only had shown one plasticity at a time (**Fig. 3i**). The additional inclusion of KRT23/HEPAR1 double positive cells provides examples of KRT23+ hepatocytes within the hepatocyte nodule (**Extended data Figure 8a**), in addition to the biliary structures shown in the original submitted paper (**Fig. 3i**). Also, we include examples of NCAM1/KRT7 double positive cells (**Extended data Figure 6j**) as well as NCAM1/HEPAR1+ double positive cells (**Extended data Figure 8d**) suggesting these are examples of ductal reaction. Finally, clearer examples of cells co-positive for KRT7 and HEPAR1 are now also included (**Extended data Figure 1c**), as representative of intermediate hepatocytes or biphenotypic cells. Due to lack of specific markers for DR and intermediate hepatocytes that do not overlap with cholangiocyte and hepatocyte markers it is difficult to clearly define if these cells are different from biphenotypic cells. We have modified our conclusions to indicate that we are likely analysing a mixture of these events in disease progression, with the plasticity signature we identify an end stage disease process, and the appearance of DR cholangiocytes and intermediate hepatocytes being an earlier event. We also added that these different populations might be functionally linked but that our analyses do not confirm or exclude this possibility.

Comment 2: Extended data Fig 3 F shows, assuming the labelling is wrong, BICC1-/KRT7- cholangiocytes and ABCC2- hepatocytes. What are these cells? Another population of ‘transdifferentiation cells’? Cluster identification is based on overall high expression of markers in these clusters as compared to other clusters, but this does not mean the markers are unique! SPP1, for example, has been described as a marker of lipid-associated macrophages by the group of Charlotte Scott. This shows that one cannot just make conclusions on cluster annotations and projected velocity plots. Of note, the group of Neil Henderson found KRT19+positive mesenchymal cells in their single cell data set (Nature 2019). They did not claim transdifferentiation of mesenchymal cells.

We thank the reviewer for this comment. Extended data Fig 3F was not very clear due to being a low resolution and due to the scaling/ colours used. We have now updated the scale and resolution of this figure. Cholangiocytes do express BICC1 and hepatocytes do express ABCC2 as expected. Here we are just indicating there is expression of both these markers on and around the connection in the umap between the cholangiocytes and the hepatocytes.

Importantly, the biphenotypic cells identified by our analyses are characterised by the expression of several markers. To clarify this point, we have now compared the gene expression levels for various hepatocyte and cholangiocyte markers in biphenotypic cells, to the main hepatocyte or cholangiocyte population. These analyses showed that biphenotypic cells express comparable levels of markers to the main cell populations (**Extended data Figure 7e**).

Comment 3: Extended data figure 6, only focusses on the main cholangiocyte cluster. I would recommend the broaden the scope and be cautious with cell annotations and the interpretation. It is possible that the DR signature is not within the more mature cholangiocyte cluster.

We have analysed the expression of DR signature across all cell types and find cholangiocytes only show a clear signature (**Rebuttal Figure 3**).

Rebuttal Figure 3 – Proportion of each cell type expressing both NCAM1 and TNFRSF12A (DR markers).

Additionally, we have now performed more detailed analysis of cells showing a DR signature (see response to comment 1), which suggested we do detect DR cells in End stage and earlier in disease.

Comment 4: ALB is a broad marker. Different online data sets indicate a high expression in cholangiocytes and even endothelial cells (<https://shiny.igc.ed.ac.uk/livercellatlas/> & <https://www.livercellatlas.org>). I would suggest to look for another marker to indicate hepatocyte lineage or phenotype. This makes the conclusions made on ALB expression questionable.

We agree that ALB should not be used in isolation when annotating cells based on gene expression. We have used numerous hepatocyte markers for cell type annotation (**extended data Fig. 3c**) Additionally, for IHC, we have now included HEPAR1 staining, which is a well-established antibody used in histology for recognising hepatocytes (**Extended data Figure 1c and 6a,b and d**).

Comment 5a: Rebuttal on comment. The term ‘intermediate’ indicates a cell state between a mature hepatocyte and a cholangiocyte (either differentiation or dedifferentiation), which in the past was usually visualised by the use of immunohistochemistry. How does this differ from biphenotypic cells? Why abandon general nomenclature accepted nomenclature?

Intermediate cells are defined by the reviewer as hepatocyte expressing KRT7+/ TROP2+/ but negative for KRT19. Using this definition, we have now included additional analyses showing that these cells gradually increase in number during disease progression (see answer to comment 1). These results confirm the relevance of these intermediate hepatocytes to monitor disease progression. Importantly, intermediate hepatocytes do not express plasticity markers before end stage disease. Furthermore, biphenotypic cells co-expressing KRT7+/ TROP2+/ remain extremely low. Finally, transdifferentiating hepatocytes are likely to express KRT7+/ TROP2+ since these markers are commonly found in cholangiocytes. Taken together, these analyses suggest that biphenotypic cells and intermediate are not the same population. However, we can't exclude that intermediate hepatocytes represent cells with an increase plasticity appearing at an early stage. Thus, these cells could be precursors of transdifferentiating cells appearing mainly in end stage liver. We have modified the conclusion of our manuscript to include this possibility.

Comment 5b: Furthermore, the authors restrict themselves to 1 aetiology and are probably not powered enough to identify the small population in pre-cirrhotic cases. Again, this does not mean it does not exist.

We have now added analysis and do detect these cells in earlier disease stages (see response to comment 1).

Comment 6: Rebuttal on comment 3. I fail to see the transcriptional changes in hepatocyte zonation. How does the transcriptome and phenotype from hepatocytes in zone1-3 change during NAFLD progression? The authors only show healthy versus end-stage.

Figure 2b shows change in zonation markers during the different stage of the disease progression. We have analysed zonation at the transcriptional level across all disease stages by performing correlation analysis and have provided statistical support for this in the previous revision round (**Extended Data Figure 5b**). This confirms that there is a significant difference in correlations of periportal and pericentral markers in the healthy liver and this difference decreases during disease, eventually no longer being significantly different in end stage disease.

Comment 7: Rebuttal on comment 9. As mentioned earlier, DR can be observed in pre-cirrhotic cases, especially in at-risk NASH patients with a high NAS/SAF score.

We have now added analysis showing that these cells can be found in our data set (see response to comment 1). We also provide additional immunostaining analyses confirming the presence of these cells in the biopsies used in our manuscript (**Extended data Fig. 1a and d**).

Comment 8: Rebuttal on comment 14. My comment was directed at the underlying biological differences, not the technical approach. There is a reason why these patients got transplanted. Decompensation is a completely different pathological process, meaning there are distinct differences between the NASH F4 needle biopsies and explant material.

We fully agree that decompensated cirrhosis is a different pathology to compensated NASH F4. However, decompensated cirrhosis still represents the ultimate outcome of NASH progression. In addition, we do detect a small number of transdifferentiating/bi-phenotypic cells in NASH F4 cells. Thus, this comment does not affect our conclusions that regenerative processes are mainly occurring during end stage disease and that transdifferentiation event occur more rarely in the earlier stage of the disease.

Comment 9: Rebuttal on comment 19. The comment “PSCs and PBC affect the biliary tree while NAFLD is the most common chronic liver disease. Furthermore, we are convinced that our conclusions can be extended to other liver disease involving chronic injury of hepatocytes.” is not fully correct. In PSC/PBC you also have damage to the hepatocytes (cholestasis, dedifferentiation, metaplasia); in ALD, the bile ducts are also chronically damaged; in ACLF you have a massive activation of the progenitor cells without differentiation. One cannot just generalise a conclusion without proof. Being convinced is not a proper scientific approach. I would suggest to talk with your experts, as clearly mentioned in the rebuttal to comment 1. They will be able to inform you better on the pathophysiology of different liver diseases.

We agree that chronic liver diseases diverge in many aspects, but they also share many features. Accordingly, we have modified our conclusion to decrease our claim that the mechanisms uncovered by our study can be directly applied to other liver diseases. NAFLD represents the largest chronic liver disease, and we quickly indicate the disease we are studying in the abstract, the text and figures. Overall, we believe the topic of the paper is clear so we would prefer to no change the title of our manuscript.

Comment 10: Rebuttal on comment 22. Some effort on using the right NAFLD nomenclature would be welcome (either the old or new consensus).

We are aware of this new nomenclature, and we will incorporate these changes in the final version of the manuscript.

Referee #2 (Remarks to the Author):

The authors have addressed my concerns in the first round of revision and I am happy to support publication.

Referee #3 (Remarks to the Author):

The authors have provided detailed answers to most of my points.

The arguments in support of Figure 2 (zonation) have been much improved, Extended Figure 5b looks convincing. Figure 2: Indicate on the plot which matrix corresponds to which stage.

We thank the reviewer for pointing this out, this has now been updated.

Comment 1: “For all these reasons, we have not used ALB as primary marker to annotate hepatocyte in our transcriptomic analyses.” Yet, Alb is listed in Table S3 for hepatocytes.

This table has been updated in the new version of the manuscript.

Comment 2: Figures 3B-D: the analyses in Figure 3 in favor of hepatocyte-like cholangiocytes and vice versa remain relatively weak. It could be the choice of the representations, but it the signatures remain quite noisy. Is the effect strong for cholangiocyte-hepatocyte compared to endothelial-hepatocyte? That would be an important control.

We thank the reviewer for this useful comment. We have re-examined endothelial marker expression in hepatocytes and found a population which expresses relatively high levels of endothelial markers (**Rebuttal Figure 4**). These same cells express low levels of hepatocyte markers, and thus, we conclude that this cluster has been incorrectly annotated. Importantly, the bi-phenotypic cells transdifferentiating between cholangiocytes and hepatocytes described in our manuscript show co-expression of high levels of both cholangiocyte and hepatocyte markers (**Extended data Figure 7e**). Furthermore, the endothelial cells cluster away from biphenotypic cells. Thus, this population of cells do not represent endothelial cells transdifferentiating into hepatocytes and therefore, does not affect our conclusions in any way.

Rebuttal Figure 4 – (a) UMAP of all hepatocytes with subclustering shown. (b) relative expression heatmap across the clusters for hepatocyte and endothelial markers.

Comment 3: Figure 3L: It is difficult to know how representative such fairly small images are, ideally such arguments (also ED Figure 1) would require quantifications.

We have included lower magnification images of these images (Extended data Figure X) to show that the regions shown are indeed representative.

Referee #1, comments upon responses to Referee #4 (Remarks to the Author):

Comment 1: Comments to the authors: The authors have mainly addressed my comments using a computational approach. The characterisation of the biphenotypical population is still rather limited and does not provide any clarity on the origin or localisation of these cell types. The additional markers included in the immunostainings do not answer this, and do not really answer the question how this relates to disease stage and outcome. Additional functional models are still lacking.

Concerning the rebuttal for my first comment, the authors should have described the biphenotypical cells more in detail to increase the novelty of the paper. The authors state that they have shown transdifferentiation without the role for progenitors. In my opinion, the authors have not provided enough scientific evidence that progenitor cells do not exist. Extended data figure 1d suggests different ways of differentiation depending on the location and stage of the disease: ductular reaction (previously described as immature bipotential cells expressing markers of biliary lineage) and plasticity of mature cholangiocytes in end-stage explanted organs. A thorough characterisation in a spatial context would have provided clarity on this issue. The few markers selected for protein validation do not fully answer this, limiting the novelty of the manuscript.

These points have been answered in detailed in our answer to reviewer 1's comments.

Comment 2: In line with this, a functional approach using in vivo models (ref comments 3, 6 and 11) would have provided useful insights. The authors missed an opportunity here to fully answer the comments. Mouse models for advanced NAFLD (CDAA-HFD or HFD-CCL4) combined with an intervention or with genetic KO background (ideally with lineage tracing) would have given clarity on the origin and function of the biphenotypic population in chronic disease and cancer. A Western diet mouse model is too mild to make conclusions on progenitor cells or trans-differentiation.

First, we would like to underline that a previous reports have already shown that direct transdifferentiation of cholangiocytes into hepatocytes (for recent example see: (Deng *et al.*, 2018)) or hepatocytes into cholangiocytes (for recent example see: (Schaub *et al.*, 2018)) can occur in mouse models of liver injury. These studies all include lineage tracing suggested by the reviewer. We fully agree that these experiments all rely on acute injury induced by toxins and thus are not directly relevant for human pathology. However, they do demonstrate that transdifferentiation is possible. Importantly, the CDAA-HFD and HFD-CCL4 models present the same limitations. These models do not mimic the metabolic phenotype associated with NAFLD/NASH (for detailed review see: (Flessa *et al.*, 2022)). Once again, these mouse model rely on toxins to accelerate the liver injury and this approach is not directly relevant for human pathology. Thus, these models are likely to simply confirm what has already been shown previously with acute injury. Overall, the WD mice remains the only model which includes metabolic phenotype, slow disease progression (including DR, fibrosis and cancer) (for detailed study see (Ghallab *et al.*, 2021)). As indicated, this model is extremely challenging to use especially with lineage tracing since disease progression can take 18-48 weeks. The use of such model would require several years and might not provide more information.

Comment 3: Previous work has shown that bipotential Lgr5⁺ cells can be differentiated towards mature cholangiocytes [Schneeberger, K. et al Hepatology 2020]. Where do these LGR5⁺ cells reside in the liver compared to the biphenotypical cells described in the current manuscript.

The existence of a putative bipotential LGR5⁺ cells in the biliary epithelium has been initially shown in the mouse (Huch et al. 2013). Importantly, the existence of such a population of LGR5 expressing cells has not been confirmed in humans (Aizarani *et al.*, 2019). Furthermore, the importance of these cells in regeneration has been challenged by recent publications based on mouse models (Planas-Paz *et al.*, 2019). LGR5 is a known target gene of the Wnt pathway (de Lau *et al.*, 2014). Thus, LGR5 mainly mark cells which are exposed to Wnt. As an example, LGR5 is a good zonation marker (Paris and Henderson, 2022) as shown in our data (**Fig.2b**) and thus cannot be use in isolation to identify putative stem cells in the liver. For this reason, we have used combination of different markers (LGR5/TROP2) to identify putative stem cell in the liver and these analyses could not identify a clear population link to biphenotypic cells.

Comment 6: For comment 8, evidence on protein level in a spatial setting should have been added to address this question. This also questions the resolution of the sequencing if you are not able to pick up well-described progenitor cell markers.

We have been able to detect LGR5 expression in hepatocytes (see above) and TROP2 in end stage cholangiocytes. This is also shown in extended data figure 7. Please also see the answer to comment 3 discussing that LGR5 is a pericentral zonation marker. Finally, LGR5 is notoriously difficult to detect at the protein level as there is no good antibody available.

Comment 7: The biliary tree has different phenotypes of cholangiocytes. The used organoids seem to be rather immature, resembling small cells of the terminal branches of biliary tree.

We politely disagree with this comment which is not substantiated by reference to published reports. Indeed, we and others have performed extensive characterisations of intra-hepatic and extra-hepatic cholangiocyte organoids (Sampaziotis et al., 2017) (Rimland *et al.*, 2021). We have also been involved in a recent review which brought key leaders in the field to reach a consensus on nomenclature and the nature of the cells grown as liver organoids (Marsee *et al.*, 2021). In a few words, ICOs are made of cholangiocytes which display the functional characteristics of mature cells including the capacity to respond to diverse stimuli such as bile acids (Sampaziotis *et al.*, 2021; Brevini *et al.*, 2022), to form duct like structure in specific conditions (Roos *et al.*, 2022) and to transdifferentiate into hepatocytes (Huch *et al.*, 2015). Again, terminology might be confusing; “mature” cholangiocytes might not have the same definition by histology vs by molecular analyses. ICOs grown in vitro are fully functional cells but they lack the regional identity of their in vivo counterpart (Roskams *et al.*, 2004). This identity can be re-established by using environmental factors such as bile acid (Sampaziotis *et al.*, 2021; Brevini *et al.*, 2022).

To further convince the reviewer, we provide below a link to navigate unpublished single cell analyses showing that ICOs grown in vitro do express markers such as KRT7, KRT19 and mucins such as MUC5B and MUC1 (**Rebuttal Figure 5**). Of note, these data complement extensive functional characterisations already performed in the literature mentioned above.

[Text redacted]

[Figure Redacted]

[Text Redacted]

Comment 8: As mentioned in comment 10, what happens if one starts from mature mucin producing cholangiocytes? E.g. organoids grown with physical constrains to induce a more mature state and then push them towards hepatocytes. This has not fully been answered, which would have provided insight into potential origin of the biphenotypic cells.

Of note, the authors state in the rebuttal ‘So, increase stiffness and change in ECM composition is likely to play a role in the tubulogenesis observed in ductular reaction.’ Ductular reaction is not the same as tubulogenesis. Tubulogenesis is the formation of ducts, observed in human biliary diseases. If the authors push their model towards more mature cholangiocytes and the formation of bile ducts, why do these cells express albumin? This raises further questions.

First, we would like to underline that these comments focus on a very minor part of the manuscript added in response to a reviewer comment, and we would be happy to withdraw these data if required. The main validation performed on ICOs concern the importance of the Insulin/mTOR pathway. The collagen/stiffness is secondary.

As indicated above, ICOs express mucins such as MUC5b and remain able to generate hepatocyte like cells in vitro. ICOs display all the functional characteristics of mature cholangiocytes.

Interestingly, our 3D imaging analyses clearly show that DR in NAFLD/NASH is associated with remodelling of the biliary tree and the formation of complex network of tubes (**Fig 3d and supplementary video 3 and 4**). Thus, it seems that DR in NAFLD/NASH also results in tubulogenesis. This observation underlines the interest to use 3D imaging and these data should be carefully considered when considering our study. Importantly, we do observe cells expressing ALB after differentiation (**Fig. 4 and extended data figure 10f**) in low or high collagen concentration. Thus, our data suggest that increase in collagen concentration (and the associated augmentation in stiffness) could increase transdifferentiation. However, previous studies have shown that tubulogenesis in vitro is not sufficient to increase the number of Alb expressing cells (Roos *et al.*, 2022).

References

Aizarani, N. *et al.* (2019) ‘A human liver cell atlas reveals heterogeneity and epithelial progenitors’, *Nature*, 572(7768), pp. 199–204. Available at: <https://doi.org/10.1038/S41586-019-1373-2>.

Brevini, T. *et al.* (2022) ‘FXR inhibition may protect from SARS-CoV-2 infection by reducing ACE2’, *Nature* 2022 615:7950, 615(7950), pp. 134–142. Available at: <https://doi.org/10.1038/s41586-022-05594-0>.

Deng, X. *et al.* (2018) ‘Chronic Liver Injury Induces Conversion of Biliary Epithelial Cells into Hepatocytes’, *Cell Stem Cell* [Preprint]. Available at: <https://doi.org/10.1016/j.stem.2018.05.022>.

Flessa, C.M. *et al.* (2022) ‘Genetic and Diet-Induced Animal Models for Non-Alcoholic Fatty Liver Disease (NAFLD) Research’, *International journal of molecular sciences*, 23(24). Available at: <https://doi.org/10.3390/IJMS232415791>.

Gadd, V.L., Aleksieva, N. and Forbes, S.J. (2020) ‘Epithelial Plasticity during Liver Injury and Regeneration’, *Cell stem cell*, 27(4), pp. 557–573. Available at: <https://doi.org/10.1016/J.STEM.2020.08.016>.

Ghallab, A. *et al.* (2021) ‘Spatio-Temporal Multiscale Analysis of Western Diet-Fed Mice Reveals a Translationally Relevant Sequence of Events during NAFLD Progression’, *Cells*, 10(10). Available at: <https://doi.org/10.3390/CELLS10102516>.

He, J. *et al.* (2014) ‘Regeneration of liver after extreme hepatocyte loss occurs mainly via biliary transdifferentiation in zebrafish’, *Gastroenterology*, 146(3). Available at: <https://doi.org/10.1053/J.GASTRO.2013.11.045>.

Huch, M. *et al.* (2013) ‘In vitro expansion of single Lgr5 + liver stem cells induced by Wnt-driven regeneration’, *Nature* [Preprint]. Available at: <https://doi.org/10.1038/nature11826>.

Huch, M. *et al.* (2015) ‘Long-term culture of genome-stable bipotent stem cells from adult human liver’, *Cell*, 160(1–2), pp. 299–312. Available at: <https://doi.org/10.1016/j.cell.2014.11.050>.

Jörs, S. *et al.* (2015) ‘Lineage fate of ductular reactions in liver injury and carcinogenesis’, *The Journal of Clinical Investigation*, 125(6), pp. 2445–2457. Available at: <https://doi.org/10.1172/JCI78585>.

de Lau, W. *et al.* (2014) ‘The R-spondin/Lgr5/Rnf43 module: regulator of Wnt signal strength’, *Genes & development*, 28(4), pp. 305–316. Available at: <https://doi.org/10.1101/GAD.235473.113>.

Marsee, A. *et al.* (2021) ‘Building consensus on definition and nomenclature of hepatic, pancreatic, and biliary organoids’, *Cell stem cell*, 28(5), pp. 816–832. Available at: <https://doi.org/10.1016/J.STEM.2021.04.005>.

Paris, J. and Henderson, N.C. (2022) ‘Liver zonation, revisited’, *Hepatology*, 76(4), pp. 1219–1230. Available at: <https://doi.org/10.1002/HEP.32408>.

Planas-Paz, L. *et al.* (2019) ‘YAP, but Not RSPO-LGR4/5, Signaling in Biliary Epithelial Cells Promotes a Ductular Reaction in Response to Liver Injury’, *Cell Stem Cell*, 25(1), pp. 39–53.e10. Available at: <https://doi.org/10.1016/j.stem.2019.04.005>.

Raven, A. *et al.* (2017) ‘Cholangiocytes act as facultative liver stem cells during impaired hepatocyte regeneration’, *Nature* [Preprint]. Available at: <https://doi.org/10.1038/nature23015>.

Rimland, C.A. *et al.* (2021) ‘Regional Differences in Human Biliary Tissues and Corresponding In Vitro-Derived Organoids’, *Hepatology*, 73(1), pp. 247–267. Available at: <https://doi.org/10.1002/HEP.31252>.

Rodrigo-Torres, D. *et al.* (2014) ‘The biliary epithelium gives rise to liver progenitor cells’, *Hepatology (Baltimore, Md.)*, 60(4), pp. 1367–1377. Available at: <https://doi.org/10.1002/HEP.27078>.

- Roos, F.J.M. *et al.* (2022) ‘Human branching cholangiocyte organoids recapitulate functional bile duct formation’, *Cell Stem Cell*, 29(5), pp. 776-794.e13. Available at: <https://doi.org/10.1016/J.STEM.2022.04.011/ATTACHMENT/63042A78-4279-4A2B-BC26-22B0055D4A75/MMC8>.
- Roskams, T.A. *et al.* (2004) ‘Nomenclature of the finer branches of the biliary tree: canals, ductules, and ductular reactions in human livers’, *Hepatology (Baltimore, Md.)*, 39(6), pp. 1739–1745. Available at: <https://doi.org/10.1002/HEP.20130>.
- Sampaziotis, F. *et al.* (2017) ‘Reconstruction of the mouse extrahepatic biliary tree using primary human extrahepatic cholangiocyte organoids’, *Nature Medicine 2017 23:8*, 23(8), pp. 954–963. Available at: <https://doi.org/10.1038/nm.4360>.
- Sampaziotis, F. *et al.* (2021) ‘Cholangiocyte organoids can repair bile ducts after transplantation in the human liver’, *Science*, 371(6531), pp. 839–846. Available at: https://doi.org/10.1126/SCIENCE.AAZ6964/SUPPL_FILE/AAZ6964_SAMPAZIOTIS_SM.PDF.
- Schaub, J.R. *et al.* (2018) ‘De novo formation of the biliary system by TGF β -mediated hepatocyte transdifferentiation’, *Nature*, 557(7704), pp. 247–251. Available at: <https://doi.org/10.1038/S41586-018-0075-5>.
- Wesley, B.T. *et al.* (2022) ‘Single-cell atlas of human liver development reveals pathways directing hepatic cell fates’, *Nature cell biology*, 24(10), pp. 1487–1498. Available at: <https://doi.org/10.1038/S41556-022-00989-7>.

Reviewer Reports on the Second Revision:

Referees' comments:

Referee #1 (Remarks to the Author):

I extend my congratulations to the authors for their commendable efforts in enhancing the current manuscript. The incorporation of additional cell annotations, the integration of immunohistochemistry/immunofluorescence visualisations, and the refined nomenclature have markedly elevated the manuscript's impact. The authors' delineation between DR and the plasticity of mature cells, substantiated by scientific evidence and thoroughly discussed, effectively addresses numerous concerns raised by Reviewer 4 and myself. I wholeheartedly concur with the authors regarding the uncertainty surrounding the contribution of a genuine adult stem cell or fetal hepatoblast/progenitor in regeneration process during human liver disease.

I am happy with the rebuttal and the focus on human samples/models.

Only a few minor comments remain:

- Please use the new nomenclature for NAFLD
- Gene annotation should be in *Italic* to distinguish with protein identifiers
- For protein cytokeratin annotation, please use CK or K (KRT is the reference to the gene)
- Extended Figure 6j: NCAM1 staining is broader than the K7 staining. Is this correct or have the labels been swapped?
- Maybe consider to add Rebuttal Fig1 and 2 to Ex Fig 6 as it helps with the interpretation of the data.

Referee #3 (Remarks to the Author):

In the latest version, the authors have fixed the points raised. In particular comment 2 was an important one, which has now been addressed.

Author Rebuttals to Second Revision:

Point-by-point manuscript 2022-11-17484D

28th March 2023

We would like to thank the editor and reviewers for their time and comments. Please see below a point-by-point response to the remaining points raised by reviewer 1.

Referee #1 (Remarks to the Author):

Only a few minor comments remain:

-Please use the new nomenclature for NAFLD

The nomenclature has now been updated throughout the text and figures using MASLD/MASH.

-Gene annotation should be in Italic to distinguish with protein identifiers

This has now been updated throughout the text and figures.

-For protein cytokeratin annotation, please use CK or K (KRT is the reference to the gene) This has now been updated throughout the text and figures.

-Extended Figure 6j: NCAM1 staining is broader than the K7 staining. Is this correct or have the labels been swapped?

This has been double-checked and the labels are correct.

-Maybe consider to add Rebuttal Fig1 and 2 to Ex Fig 6 as it helps with the interpretation of the data.

We thank the reviewer for the suggestion, but we would like to keep the figure as it is for simplicity. The decision to include this figure for the reviewers was in part due to an editorial advice when the rebuttal plan was discussed.